# Multiple runoff processes and multiple thresholds control agricultural runoff generation

Shabnam Saffarpour[1], Andrew W. Western[1], Russell Adams[1], and Jeffrey J. McDonnell[2,3]

[1]Department of Infrastructure Engineering, The University of Melbourne, Parkville, 3010, Australia

5  [2]Natl Hydrol Res Ctr, Global Inst Water Secur, University of Saskatchewan, Saskatoon, SK S7N 3H5, Canada}

[3]School of Geosciences, University of Aberdeen, Aberdeen UK}

*Correspondence to*: Andrew Western (a.western@unimelb.edu.au)

**Abstract.** Thresholds and hydrologic connectivity associated with runoff processes is a critical concept for understanding catchment hydrologic response at the event timescale. To date, most attention has focused on single runoff response types and the role of multiple thresholds and flow path connectivities has not been made explicitly. Here we first summarise existing knowledge on the interplay between thresholds, connectivity and runoff processes at the hillslope-small catchment scale into a single figure and use it in examining how runoff response and the catchment threshold response to rainfall affect a suite of runoff generation mechanisms in a small agricultural catchment. A 1.37 ha catchment in the Lang Lang River catchment, Victoria, Australia was instrumented and hourly data of rainfall, runoff, shallow groundwater level and isotope water samples were collected. The rainfall, runoff and antecedent soil moisture data together with water levels at several shallow piezometers are used to identify runoff processes in the study site. We use isotope and major ion results to further support the findings of the hydrometric data. We analyse 60 rainfall events that produced 38 runoff events over two runoff seasons. Our results show that the catchment hydrologic response was typically controlled by the antecedent soil moisture index and rainfall characteristics. There was a strong seasonal effect in the antecedent moisture conditions that led to marked seasonal scale changes in runoff response. Analysis of shallow well data revealed that streamflows early in the runoff season were dominated primarily by saturation excess overland flow from the riparian area. As the runoff season progressed, the catchment soil water storage increased and the hillslopes connected to the riparian area. The hillslopes transferred a significant amount of water to the riparian zone during and following events. Then, during a particularly wet period, this connectivity to the riparian zone, and ultimately to the stream, persisted between events for a period of one month. These findings are supported by isotope results which showed the dominance of pre-event water, together with significant contributions of event water early (rising limb and peak) in the event hydrograph. Based on a combination of various hydrometric analyses and some isotope and major ion data, we conclude that event runoff at this site is typically a combination of subsurface event flow and saturation excess overland flow. However, during high intensity rainfall events, flashy catchment flow was observed even though the soil moisture threshold for activation of subsurface flow was not exceeded. We hypothesize that this was due to the activation of infiltration excess overland flow and/or fast lateral flow through preferential pathways on the hillslope and saturation overland flow from the riparian zone.

# 1 Introduction

Thresholds have been an integral part of overland flow theory since the early infiltration excess work of Horton (1933) and saturation excess studies of Dunne and Black (1970a, b). Thresholds in runoff response have also been observed in subsurface stormflow dominated systems (Hewlett and Hibbert, 1967). More recent work has shown these to be a function of catchment wetness status for saturation excess overland flow (Western and Grayson, 1998;Western et al., 2005) and subsurface stormflow (Freer et al., 2002;Tromp-van Meerveld et al., 2007). Hydrological connectivity is now a useful generic concept that links reservoirs to their downstream conduits (Tetzlaff et al., 2010) and a connectivity framework can provide a powerful explanator of catchment flow and transport response (Ali et al., 2013;Detty and McGuire, 2010;Lehmann et al., 2007;McGuire and McDonnell, 2010;Western et al., 1998, 2001).

In this paper we are interested in connectivity in terms of the movement of water from hillslopes to streams at the timescale of events and longer. We say there is connectivity along a flow pathway when water is moving along that pathway and contributing to stream flow from the catchment. Connectivity and thresholds are intimately related; typically a threshold in some catchment state controls the transition between connected and disconnected states; for example, the observation that subsurface flow becomes connected above some soil water storage and rainfall threshold (Detty and McGuire, 2010;Tromp-van Meerveld and McDonnell, 2006a;Fu et al., 2013;Penna et al., 2015). In this study we use the threshold concept to examine runoff generation mechanisms and to discuss how various mechanisms produce runoff and change in importance during the runoff season.

Despite significant progress in understanding the non-linear behaviour of catchments related to soil moisture thresholds, watertable dynamics, connectivity of surface and subsurface pathways and their influence on runoff generation mechanisms, it is not explicitly understood how the non-linear properties of catchments (connectivity and thresholds) work to convert rainfall to runoff nor how such behaviours vary between different types of catchments. It has been argued that interactions between the various processes and thresholds leads to complex non-linear rainfall-runoff behaviour in catchments (Hopp and McDonnell, 2009;Kirchner, 2006;Tetzlaff et al., 2010;Uchida et al., 2005) including: thresholds for initiation of hillslope-to-stream connectivity (Ali et al., 2013;Detty and McGuire, 2010;Fujimoto et al., 2008;Lehmann et al., 2007;McGuire and McDonnell, 2010;Tromp van Meerveld and McDonnell, 2005;Tromp-van Meerveld and McDonnell, 2006a); variable flow hysteresis patterns depending on rainfall amount and antecedent soil moisture conditions (Bowes et al., 2009;Holz, 2010;McGuire and McDonnell, 2010); and flushing of nutrients in agricultural catchments (Bracken and Croke, 2007;Ocampo et al., 2006;Tockner et al., 1999;Withers and Lord, 2002). Moreover, the explicit linkage of runoff mechanisms and flow pathways has received less attention in agricultural landscapes compared with forested basins. An exception is the study of Ocampo et al. (2006). Ocampo et al. (2006) investigated hydrologic connectivity, the threshold dependency of connectivity and its influence on seasonal and event based runoff mechanisms in an agricultural catchment in Western Australia.

While the concept of connectivity has been useful in many of these studies, most studies have concentrated on individual mechanisms. It is less clear how catchments behave when subject to a mixture of runoff mechanisms including infiltration excess and saturation excess overland flow, and subsurface stormflow. Only a small number studies have tried to tease apart the influence of multiple processes in catchments where infiltration excess runoff, saturation excess runoff and subsurface

stormflow are all important (e,g, Lana-Renault et al., 2007;Lana-Renault et al., 2014;Latron and Gallart, 2008). These catchments are in Mediterranean environments and have typically been studied by field inspection of the catchment surface to identify infiltration excess runoff generation areas, field inspection of saturated areas to identify saturation excess source areas and also by examination of hydrographs in combination with groundwater wells. These studies associate certain controls with specific processes, such as past cultivation being associated with infiltration excess runoff (Lana-Renault et al.,

2014). These studies have generally not considered rainfall intensity information at scales finer than daily. Despite the considerable improvement in our understanding of basic catchment functional mechanisms, today it is still challenging to apply our understanding of these specific case studies to other areas due to the problems of catchment heterogeneity, complexity, non-linearity of behaviour, scale, etc. (Beven, 2002).

To aid systematic consideration of the variation in runoff processes between catchments and over time, we develop a

summary of the status quo in terms of the combined effects of thresholds and connectivity on runoff processes at the hillslope to zero-order catchment scale (Figure 1). Often we think of dominant runoff processes and, as Figure 1 is easiest to interpret in that context, the following discussion takes that view initially. However, there is often a mix of runoff processes either spatially due to heterogeneity or for different events and Figure 1 can also be used to interpret such a mixture, which we will return to later. This paper aims to tease apart the influence of different processes by considering 60 rainfall events

(resulting in 38 runoff events) in a small agricultural catchment. We show the shifting importance of different processes over time associated with changes in catchment wetness and rainfall intensity and apply the runoff process framework. We consider the role of thresholds in different catchment states and fluxes as well as the role of thresholds in certain timescales in controlling different modes of hydrologic connectivity and associated rainfall-runoff response. The variety of potential thresholds leads to a variety of runoff processes, as illustrated in the following discussion.

Figure 1 shows the importance of various timescales, durations, fluxes and states, and how these relate to variation in rainfall-runoff processes over time (and between catchments with different physiographic characteristics). Of course, questions of instantaneous flux and also of the relative timescales of various processes are often important in determining the existance of connectivity (Tromp van Meerveld and McDonnell, 2005, 2006b;Western et al., 2005). It would be attractive to think of the problem of runoff response purely in terms of timescales of competing processes following Oldham et al.

(2013), who used a generalised Damköhler number to represent the competing effects of transport and reaction timescales on the loss of material along a flow path. When the reaction timescale is small compared with the transport timescale, the reactant is consumed before reaching the exit of the flow path it is moving along. A complication here is that, both flux and time thresholds are important. This arises because there is finite capacity for flow in various parts of the catchment system.

Figure 1 is divided into three parts, the lefthand area provides a series of catchment thresholds that depend on hydroclimatic and landscape characteristics and influence the type of runoff process and the connectivity between the hillslope and stream, depending on whether they are exceeded or not. The middle area points to the outcome in terms of dominant runoff generation processes and the righthand area provides example catchments from the literature that exhibit those processes. In Figure 1 we define the specific runoff processes as follows.

- Infiltration excess runoff: runoff that occurs due to the rainfall (or throughfall) rate exceeding the infiltration capacity of the surface and that results in flows of water to the catchment outlet by surface flow pathways.

- Subsurface stormflow: runoff due to infiltration that generates rapid lateral subsurface flow (i.e. a quickflow response) that flows to the catchment outlet through subsurface flow paths for at least part the distance to the outlet. This water may exfiltrate in low convergent parts of the catchment or directly into the stream before reaching the catchment outlet. Often impeading layers and preferential flow paths are implicated.

- Saturation excess runoff: runoff due to rainfall on saturated areas that flows to the catchment outlet by surface flow pathways. The saturated area may be generated by either lateral flow in excess of the capacity of the hillslope to transmit the lateral flow, a drainage impediment at depth coupled with a sufficient excess of infiltration over evapotranspiration and drainage, or a combination of these.

Some of the thresholds are posed in terms of flux rate compared with a flow capacity (e.g. box 2) and some in terms of a state threshold (box 5). The flux and state thresholds are considered in the context of process timescales and durations. This is because the threshold needs to be exceeded for a sufficient time for the action of the process to lead to a significant impact. That impact typically involves lateral flow either on the surface or through the subsurface and hence also interactions up and downslope. Specific examples are given below.

Consider box 1. Rainfall rates vary across a very wide range to timescales. If the rainfall (or throughfall) intensity exceeds the infiltration threshold for only a very short time, the water that ponds on the surface will continue to infiltrate as it flows down the hillslope toward the stream (runon infiltration) after the intensity reduces and very little or no runoff will reach the stream (surface connectivity did not become established). However if average intensities exceed the infiltration capacity for long enough for ponded water to flow to the catchment outlet, the hillslope will connect to the catchment outlet via surface pathways and produce runoff. Thus the duration of high intensity rainfall compared with the overland flow timescale (flow distance divided by wave celerity) is important. The remaining boxes consider thresholds in the context of subsurface flow times. Box 3 considers situations where subsurface saturation exists, allowing water to flow along lateral subsurface flow paths. If any of deep infiltration through the impeding layer (Jackson et al., 2014), unfilled bedrock storage (Janzen and McDonnell, 2015) or evaptranspiration (including between events) causes the saturation and/or lateral flow to cease before water can move a significant distance downstream, the water will not be effectively redistributed downslope, subsurface connection will not be established. If the saturation persists for long enough for lateral subsurface flow to move down the hillslope and into the stream, connectivity between the hillslope and the stream will develop. Thus the ratio of the lateral flow timescale for the hillslope and the evaporative drying (or other loss) timescale is important here. At the other extreme

(box 7), if lateral flow is persistently exceeding the subsurface flow capacity, surface saturation will exist leading to saturation excess runoff because saturated areas will exist antecedent to the event in this case. Provided continuous saturation exists to the sream, this will lead to a surface flow path connecting the hillslope and stream. In this case the lateral flow timescale is long enough to lead to memory in the spatial patterns between events (non-local control of Grayson et al (1997)).

Figure 1 goes about here

While Figure 1 suggests catchment rainfall-runoff response is dominated by specific processes (e.g. saturation excess runoff) it needs to be recognised that many catchment conditions vary over time and space. For example in our study catchment summer rainfall is often more intense than winter rainfall and this can lead to differences in runoff processes between events. Soil water conditions vary seasonally in response to both rainfall and potential evapotranspiration, sometimes leading to switching between characteristic spatial patterns of soil moisture and prevailing responses to rainfall (Grayson et al., 1997;Western et al., 1999). Topographic, soil and vegetation conditions can also vary across a catchment. This all suggests that catchments could exhibit a mix of processes.

Having introduced the framework above, we use it to understand the behaviour of a catchment in Australia that does indeed exhibit a mix of runoff processes. We examine how soil water storage and shallow water table response influence surface and subsurface connectivity and rainfall-runoff response at seasonal and event based time scales. We also examine the relative role of saturation excess and subsurface flow in generating peak runoff rates and event volumes. Finally we examine circumstances under which rainfall intensity plays a role in runoff generation responses. The field site is a small agricultural catchment in the Lang Lang River catchment, Victoria, Australia, which we examine through the lens of hydrometric and isotope and geochmistry measurements. In the context of Figure 1, these results are used to examine various runoff generation mechanisms and flow pathways that are important in the study catchment and to determine how they contribute to produce the catchment runoff as the catchment wetness and rainfall intensity vary. Thus the contributions of this paper revolve around the introduction of the framework in Figure 1 and demonstration of its application using a catchment where multiple runoff processes are important. The paper also contributes to further understanding of the role of thresholds and connectivity in determining flow pathways and runoff processes in agricultural catchments. Most connectivity studies in the past have focussed on forested catchments.

## 2 Methods

### 2.1 Study location

The study site is a 1.37 ha catchment (named RBF) located on a dairy farm at Poowong East, in the Lang Lang River Catchment, Victoria, Australia, 130 km south-east of Melbourne (Figure 2). The study area has a humid climate and rainfall is uniformly distributed across the year with an annual mean (1961-1990) of 1100 mm (Bureau of Meteorology, 2009). Annual areal potential evapotranspiration (1961-1990) is 1040mm (Bureau of Meteorology, 2005).

A general description of the study catchment can be found in Adams et al. (2014). The study period was between September 2009 and December 2011. Elevation ranges from 160 to 210 mAHD and the slope varies from 2% to 50%. Based on field observations, the topography, range of slopes and the groundwater behaviour, the catchment was divided into four different zones: 1) the riparian area located on the relatively flat convergent lower part of the catchment (outlined in red on Figure 2)

included sites 1, 2, 32 and 3; 2) the lower slope (low slope) area; 3) the mid slope area with sites 4, 5, 6 and 7 ; and 4) the upper slope (upslope) area with sites 10, 11 and 15 (Figure 2).

Figure 2 goes about here

The catchment geology comprises of sandstones and mudstones of the Cretaceous Strezlecki Group (VRO, 2013). Outcrops on the lower stream banks of the catchment (just downstream of the monitored hillsope) show weathered sandstone and

mudstone bedrock. Hand augering revealed a soil depth of between 1 and 1.6 m, and the lower parts of the profile included mottled clay and weathered bedrock particles. The soils are acidic and mesotrophic brown dermosols (Isbell, 2002) that grade from a fine sandy clay loam to a medium clay with mottles and weathered bedrock. Soil profile depth decreases moving downslope. These soils typically have a moderate hydraulic conductivity surface horizon (0-40 cm, $K_s \approx 5*10^{-6}$ m s$^{-1}$, about 20 mm hr$^{-1}$). The dominant land use is grazing by dairy cows.

**2.2 Site instrumentation and hydrometric data monitoring**

Stream discharge was measured at the catchment outlet (Figure 2) using an RBC flume (Clemens et al., 1984) and an Odyssey (Dataflow Systems inc. Christchurch, NZ) pressure transducer (PT) recorded stream water levels every 10 minutes, which were used to compute instantaneous discharge rates. After August 2011, the PT was replaced with an ISCO (Teledyne ISCO , Lincoln,NE,USA), model 730 bubbler. The following rating curve was used to calculate $Q$ from water level:

$Q = 0.001H^2 + 0.0168H$                                                     (3)

where $Q$ is stream flow discharge and $H$ is water head.

Rainfall data were recorded using a tipping-bucket raingauge at an automatic weather station (AWS) which was installed in 2010 on the upper boundary of RBF. Weather variables (temperature, humidity, wind, rainfall, global radiation) were measured by the AWS. Areal potential evapotranspiration (APET) was also computed using the Morton (1983) wet

environment method on a daily basis. APET was strongly seasonal resulting in strongly seasonal soil moisture contents and intermittent streamflow at RBF. Soil moisture storage was calculated from the volumetric soil water content which was measured for the 0-30 cm and 30-60 cm layers and recorded hourly by the AWS logger using two vertically installed 30 cm long Campbell Scientific (CS625) soil moisture probes (Campbell Scientific, 2006). The 0-30cm probe was inserted from the surface.  To install the 30-60cm probe, a 30cm deep hole was dug, with the excavated soil set aside.  The probe was inserted

vertically and then the excavated soil was repacked so that the soil was replaced at a similar depth to that from which it had been removed.

The CS625 produces a pulse signal and the pulse period was temperature corrected using measured soil temperature and the manufacturers recommended temperature correction, as follows (Campbell Scientific, 2006).

$$\tau_{corrected}(T_{soil}) = \tau_{uncorrected}$$
$$+ (20 - T_{soil}) * (0.526 - 0.052 * \tau_{uncorrected} + 0.00136 * \tau_{uncorrected}^2) \tag{1}$$

where $\tau_{uncorrected}$ is the probe output period, $T_{soil}$ is the soil temperature and $\tau_{corrected}$ is the corrected probe output. The *VWC* was then computed from $\tau_{corrected}$ using (Campbell Scientific, 2006):

$$VWC = -0.0663 - 0.0063 * period + 0.0007 * period^2 \tag{2}$$

Soil water storage over the top 60 cm soil depth was computed by adding the VWC from the 0-30 cm and 30-60 cm layers and then multiplying by the 300 mm soil depth. We use this soil water storage at the start of each rainfall event as an index of antecedent soil water (ASI) and assume it represents the catchment wetness condition.

To capture the nature of hydrologic connectivity, runoff mechanisms and flow pathways, shallow (1.5-1.6 m) groundwater wells were installed at 12 sites across the RBF catchment using 40 mm PVC pipes and backfilled with sand, bentonite, the topsoil and grass. Figure 2 shows these sites of which 1, 2, 3, 16 and 32 were in the riparian zone; 4, 5, 6 and 7 were on the mid slope; and 10, 11 and 15 were on the upper slope. Sites 4, 5 and 6 were equipped with water level loggers from July 2010 until the end of study period in December 2011. Sites 3, 7, 16 and 32 were logged from winter 2011 until the end of study period in December 2011. Water level loggers were not installed at sites 1 and 2 since they were nearly always saturated. At sites 1 and 2 groundwater levels were measured manually. Water levels were logged using Odyssey PT loggers.

### 2.3 Water sampling and analysis

A rainfall sampler (Kennedy et al., 1979) collected up to ten sequential rainfall samples per event, each being equivalent to 6.6 mm of rainfall. The sampler was initially installed close to the AWS, however, due to instances of damage by animals, it was relocated near to the flume in August 2010 until the end of the study period (December 2011).

An auto sampler (Teledyne ISCO 6712) was installed at the flume and streamflow sampling was triggered based on the rising stage. Following triggering, the sampler was programmed to collect up to 24 samples of streamflow at hourly intervals. Samples were removed from the auto sampler within 48 hours. To reduce the laboratory analysis workload, we plotted recorded water levels from the RBF catchment in the field prior to removing the event sample from the auto sampler and selected certain samples for analysis. All samples during the rising limb and the peak were selected and samples were typically selected at an interval of 4 hours during the falling limb. Routine grab sampling was undertaken at weekly intervals during the main runoff season when water was flowing through the RBF flume. This was supplemented by additional grab sampling during visits to collect event samples from the auto sampler. Routine grab sampling was also undertaken at weekly intervals during the main runoff season when water was flowing through the RBF flume.

Sub-samples for isotopic analysis were taken of stream water from both manual and auto sampler samples, and from all full rainfall sample bottles; these were collected in glass bottles for isotope analysis. Bottles were completely filled. The samples were refrigerated (+ 4°C) until analysis for $\delta^{18}O$ and $\delta^{2}H$. Stable isotope ratios were measured either at Monash University using a Finnigan MAT 252 and ThermoFinnigan Delta Advantage Plus mass spectrometers (2010 samples) or at the

University of Melbourne, where a Picarro L2120i cavity ring-down isotope analyser was used to determine isotope ratios (2011 samples). The instrumental precision was $\delta18O = \pm0.15‰$ and $\delta2H = \pm1‰$ for the isotope samples analysed at the Monash University and it was $\delta^{18}O=0.1‰$ and $\delta^{2}H =0.4‰$ for samples analysed at the Melbourne University. The results from the isotope analysers were checked for systematic differences, by analysing duplicate samples in both laboratories. We developed and applied a correction between the two laboratories based on these samples. With the exception of determining

pre-event end member uncertainty (described later), the analyses presented here only use samples analysed at the Monash laboratory and hence this difference between laboratories only affects our hydrograph separation uncertainty estimates. For isotope analysis, in total we collected 115 samples of rainfall, 28 samples during low flows and multiple stream water samples of isotopes from five events.

For the measurement of major ions, subsamples of grab and event streamflow samples were collected using 50 ml plastic

bottles. Sub-samples were also taken from each water sample for selected major ion ($Na^+$, $K^+$, $Ca^{2+}$, $Mg^{2+}$ and $Cl^-$) analyses, which were analysed in the NATA-certified, analytical chemistry laboratory of the Water Studies Centre at the Monash University using standard methods. Major ions were determined in non-filtered samples as follows (Cm is ion concentration):

- $Mg^{2+}$ was determined by atomic absorption spectrometry (APHA, 2005). The issue of interference with Si, Al,
and P was solved using a combination of Lanthanum and Caesium. The determined uncertainty for $Mg^{2+}$ was Cm* 0.0596, with 95% confidence interval (CI).

- $Ca^{2+}$ was analysed using atomic absorption spectrometry utilising an air/acetylene flame based on the American Public Health Association procedure (APHA, 2005). The issue of interferences with Si, Al, and P was solved using Lanthanum releasing agents. The determined uncertainty for $Ca^{2+}$ was Cm* 0.0386 (95%
CI).

- Following APHA (2005) approach, "flame atomic absorption spectrometry" was used to determine $K^+$ concentration and Caesium was used to solve the issue of ionization. Uncertainty for $K^+$ was Cm* 0.0372 (95% CI).

- $Na^+$ which is "ionized in the air/acetylene flame" was analysed by adding Caesium to overcome this problem
following the APHA (2005) method. Uncertainty in $Na^+$ results was Cm * 0.0432 (95% CI).

- $Cl^-$ was analysed by colourimetrically method "using flow injection analysis (FIA)" (APHA, 2005). Uncertainty in $Cl^-$ analysis laboratory results was Cm * 0.05 (95% CI).

## 2.4 Rainfall and runoff events

In order to analyse event behaviour, it was necessary to identify rainfall and runoff events. Based on an examination of the time series of hourly rainfall in the catchment (in the study period which was between April 2010 and December 2011), rain events were defined as having $\geq$ 5 mm total rainfall, and peak hourly rainfall intensity, $I_{peak} \geq$ 1.5 mm hr$^{-1}$. Distinct events were separated by > 12 hours without rainfall.

The runoff hydrograph was also divided into events. Runoff events began when the stream discharge hydrograph started to rise from its initial low flow value or moved above a threshold of 0.05 mm hr$^{-1}$ following the commencement of a rainfall event. Events ended either when: 1) the discharge returned to its initial value; 2) a new rainfall event started; or, 3) 96 hours after the end of the rainfall event in unusually wet situations where elevated flow continued. For each event, a number of characteristics were determined as shown in Tables 1 and 2.

The antecedent soil moisture index (*ASI*) was represented as the amount of the soil water storage in the top 60 cm of the profile at the AWS at the start of each rainfall event. The topography of the catchment was surveyed using a differential Global Positioning System (dGPS) and a 1 m horizontal resolution DEM (digital elevation model) was developed by interpolation methods. Instrument and well locations were also determined using dGPS. Then these data were used to produce maps of the study area and including sampling sites using Arc GIS. The lateral boundary of the saturated area was topographically constrained and field inspections suggested it was stable over time, while the upstream boundary moved up and down the riparian zone. The saturated area was estimated by locating the upper boundary through field inspection and then measuring the distance from either well 2 or 3 (The saturated area extended towards site 3 during very wet conditions). Figure 2 demonstrates the approximate maximum boundary location of the saturated area at site 3 and the main stream located close to the outlet of the catchment. The saturated area was then estimated from this information combined with the mapping of the riparian zone boundary in ArcGIS. These measurements were made between events. The proportion of saturated area was estimated using these data and then used to estimate saturation excess runoff generation for the different events. Consistent with our definitions in Figure 1, here we conceptually separate return flow and flow resulting from direct precipitation on the saturated area and use Saturation Excess Runoff to refer to the latter. The event runoff depth (mm) and event runoff coefficient (RC %) were calculated by separating the event hydrograph using the method of (Hewlett and Hibbert, 1967), which has been widely applied (Buttle et al., 2004;Fujimoto et al., 2008;McGuire and McDonnell, 2010). The method assumes that baseflow increases at the rate of 0.55 l s$^{-1}$km$^{-2}$h$^{-1}$ (0.002 mm hr$^{-1}$) from the start of the rising limb. While this method, like other hydrograph based baseflow separateion methods, is arbitrary, the results are insensitive to the assumed rate of rise.

## 2.5 Isotopic Hydrograph Separation

Isotope samples are used for hydrograph separation into pre-event and event contributions. We applied the well- known one tracer, two component model of hydrograph separation approach (Pinder and Jones, 1969; Sklash and Farvolden, 1979). We

determined uncertainties following Genereux (1998). We undertook the analysis using both the $^2$H and $^{18}$O data. We used the mean of the rainfall samples during the event to estimate the event water end member and low flow samples from the few days to the event for the pre-event end member. In the uncertainty analysis, the standard deviations of the event rainfall samples and of all the low flow samples across the study period were used. Half the analytic uncertainty was used to represent the standard deviation of the streamflow sample.

## 3 Results

The following results first provide an overview of the seasonal behaviour and rainfall-runoff events. They then examine whether thresholds in the antecedent conditions and/or event rainfalls exist. Next, links between the catchment condition and the event runoff are examined using the piezometer and soil moisture data. After that, the recession behaviour of events is examined and linked to catchment wetness conditions. Finally isotope and major ion data are presented for selected events.

### 3.1 Overview of runoff behaviour and rainfall-runoff event characteristics

Figure 3 shows time series of weekly rainfall, APET, soil water storage and runoff. The rainfall, although variable from week-to-week, exhibited seasonality, while there was strong seasonality in PET. This drove a strong seasonality in soil water storage. An examination of the weekly runoff data shows that there was generally no flow from about October to May due to the seasonal nature of this catchment; however, an exception was that persistent low flow occurred from 26 November 2011 to the end of the event on 10 December 2011. During this period ASI was often relatively low but there was frequent and substantial rainfall (>200mm in 30 days). While a strong link between runoff and soil water storage is evident at the seasonal scale in Figure 3, there are exceptions at the event scale. For example in February 2011, there was runoff response despite the catchment being near to the lowest soil water storage for the study period.

Figure 3 goes about here

Moving to the event timescale, Table 1 summarises 38 rainfall-runoff events and Table 2 shows a summary of 22 rainfall events that did not produce a runoff response. A further 16 rainfall events occurred over the study period which are not included in the analysis due to missing stream discharge data. For the 38 runoff events, total event rainfall varied from 7 to 72 mm, $I_{peak}$ ranged from 2 to 31 mm hr$^{-1}$, $ASI$ ranged from 130 to 286 mm and total event runoff varied between 0.23 and 41 mm. For the no-flow events (Table 2), total rainfall varied from 5 to 28 mm, $I_{peak}$ ranged from 2 to 10 mm hr$^{-1}$ and ASI ranged from 146 to 238 mm. Figure 4 shows rainfall-runoff responses for selected events at RBF. These graphs are ordered from lowest (27/11/2010) to highest $ASI$ (7/6/2011) for the selected events. All events (except events on 26/11/2011 and 10/12/2011(Figure 4c)) presented in Figure 4 had zero or very low initial flow. For the events on the 26/11/2011 and 10/12/2011, the initial discharge was 0.07 and 0.13 mm hr$^{-1}$, respectively.

Table 1 goes about here

Table 2 goes about here

Figure 4 goes about here

In Figure 4 most events showed rapid response to rainfall, except for the event on 8/12/2010 (Figure 4b), which did not produce any significant runoff, and the event on 7/6/2011. The events on 27/11/2010 and 10/12/2011 in particular showed a very flashy response. These events had the highest peak hourly rainfall intensity (30.4mm/ and 31mm/h respectively) during the study period and they occurred at the end of the flow season with low $ASI$ ($ASI$ was 161 and 192 mm respectively). The highest peak runoff rates for the study period were for the events on 27/11/2010 and 10/12/2011, which were 2.4 and 5.6 mm hr$^{-1}$, respectively. In contrast to most events, the runoff response for the event on 27/11/2010 was transient with very rapid recession. For the event on 10/12/2011, a second peak of moderate rainfall intensity (about 10 mm hr$^{-1}$) produced a second runoff peak and there was a more significant recession flow following the rainfall bursts. This was also true for the other events (26/11/11, 11/5/11, 8/11/11, 1/8/10, 7/6/11) shown in Figure 4, which were typical of responses to lower intensity rainfall during wetter (in terms of soil water) periods.

For events with $I_{peak} < 10$ mm hr$^{-1}$ there was a general increase in response as the $ASI$ increased. The event on 12/11/2010 had 184 mm $ASI$ and total rainfall was 28 mm and it did not produce any runoff. This was a typical example of no flow events. Coming into the runoff season, as $ASI$ increased (e.g. 220 mm on 11/5/2011), RBF started to respond gradually, producing small amounts of runoff (e.g. for events on 11/5/2011 and 14/5/2011). When the $ASI$ was > 250 mm for the event on 7/6/2011, it can be clearly seen that RBF responded to this low intensity, small size rainfall event with a delayed and smooth discharge hydrograph with continued flow following the event. This also occurred for the next event on 1/8/2010.

## 3.2 Runoff thresholds

In Figure 1 we set out a number of thresholds that are important in runoff production mechanisms. We now explore the event data from the perspective of thresholds, concentrating on two key ones: catchment wetness and rainfall intensity. Figure 5 builds on approaches by Detty and McGuire (2010), who considered thresholds in $ASI$ and $ASI$+Rain (i.e. Figure 5c and d), and Janzen and McDonnell (2015), who considered the impact of event rainfall and rainfall intensity on event runoff (i.e. Figure 5a). As we move from Figure 5a to 5d, the various influences on rainfall-runoff behaviour and thresholds become clearer. Figure 5a shows event runoff as a function of event rainfall, with the highest hourly rainfall intensity ($I_{peak}$) indicated in colour. This illustrates that there is little relationship between event rainfall and runoff or between rainfall intensity and runoff. For some quite large events (up to ~25mm), zero runoff can occur. There was also a wide variation in runoff coefficients (indicated by the scatter). It is also clear that the events with high peak hourly intensity also had relatively large total rainfall accumulations.  Overall, Figure 5a shows that event rainfall and intensity do not effectively differentiate rainfall-runoff behaviour when considered by themselves.

Figure 5 goes about here

Figure 5b begins to separate out different effects by showing the impact of five factors together. The cumulative curve shows the distribution of soil water storage as observed throughout the study period. Specific events are shown with the *ASI* identified (left hand end of the grey lines) and the *ASI*+Rain depth (filled markers at the right end of the horizontal grey lines). The length of the lines is the rainfall depth for the event. The colour of each bubble shows the peak hourly rainfall intensity ($I_{peak}$) and the size of the bubbles shows the event runoff coefficient. Squares indicate events that did not produce any runoff or where the peak runoff rate was less than 0.05 mm hr$^{-1}$. The vertical grey dashed line shows *ASI* (left hand end of horizontal grey line) or *ASI*+Rain=250mm (right hand end of horizontal grey line).

There are several trends that can be discerned from Figure 5b. First the rainfall events analysed here occurred across the full range of catchment wetness and were relatively evenly spread, showing that the rainfall events occur over a representative range of catchment antecedent conditions. The larger rainfall events generally occurred in summer when *ASI*<250 mm and a mix of low and high intensity events occurred for these conditions, also in summer. All the events on a wet catchment (*ASI*>250 mm) had low $I_{peak}$ (≤6.2 mm hr$^{-1}$). Events where the *ASI*+Rain was less than 250mm usually did not generate any runoff, although there were some high intensity rainfall events that were exceptions and a small number of events with very low runoff coefficients (1-4%) where the *ASI*+Rain was generally between 230 and 250mm. These low runoff coefficient events were at the end of the runoff season.

Figures 5c and 5d examine the impact of catchment wetness, quantified as *ASI* and *ASI*+Rain respectively, at the start (5c) and end (5d) of the event on event runoff response, combined with the impact of rainfall intensity, $I_{peak}$. Catchment wetness is plotted on the x-axis and rainfall intensity on the y-axis. Values *ASI*=250mm (Fig 5c), *ASI*+Rain =250mm (Fig 5d) and $I_{peak}$=15mm/h are shown by grey dashed lines. The bubble size shows the event runoff coefficient, as before, and crosses indicate rainfall events that did not generate any runoff. Colour indicates the runoff volume. The runoff coefficient behaviour is separated into groups more clearly in Figure 5d than in 5c. In Figure 5d, three different groups of events can be identified. Looking along the x-axis, there is a threshold at *ASI*+Rain=250mm that separates most events with a significant runoff response from those without. Looking along the y-axis direction, it can be seen that the *ASI*+Rain threshold is not successful at distinguishing runoff when $I_{peak}$ is high and there is also a threshold in runoff response at $I_{peak}$ =15mm/h. Thus the three groups are: 1) events without runoff where *ASI*+Rain<250mm and $I_{peak}$<10 mm hr$^{-1}$; 2) events that produce runoff when *ASI*+Rain > 250 mm; and 3) events with *ASI*+Rain<250 mm and $I_{peak}$>15 mmhr$^{-1}$ that did produce runoff (Tables 1 and 2). Where *ASI*+Rain exceeded 250 mm (group 2), some runoff was always produced.

Both of the first and third groups had *ASI*+Rain less than 250 mm but they behaved differently in that some produced runoff and others did not. In the first group low intensity rainfalls mostly happened in drier periods when *ASI* varied between 146 and 227 mm. It is assumed that rainfall completely infiltrated into the soil and these events did not produce runoff (see Table 2 for event characteristics).

The third group (Table 1) occurred during dry periods at the end of the flow season when the *ASI* was < 200 mm. The runoff coefficients for the four events with peak hourly intensity of 15 mm hr$^{-1}$ and higher are 3, 12, 20 and 68% for peak hourly intensities of 16, 30, 15 and 31 mm hr$^{-1}$ and *ASI*+Rain of 202, 215, 257 and 245 mm respectively. Note that one of these

events exceeds both the wetness and intensity thresholds. These events were distinguished by having maximum hourly rainfall intensities above ~15 mm hr$^{-1}$ and they did produce runoff. In particular, two of these events on 27/11/2010 and 10/12/2011 had the highest rainfall intensities observed ($I_{peak}$> 30 mm hr$^{-1}$) and they produced the highest peak runoff rates (8.1 mm hr$^{-1}$and 9.1 mm hr$^{-1}$) and hourly runoff totals (2.4 and 5.6 mm) observed during the study period (Figure 3). These runoff peaks happened at the same time as the highest recorded rainfall intensities. Antecedent stream discharge for the events on the 27/11/2010 and 4/2/2011 was zero and the hydrograph rose and recessed quickly. For the event on the 10/12/2011, the *ASI* was 192 mm, the total rainfall was 53 mm, with an initial discharge of 0.13 mm hr$^{-1}$ (Table 1). It produced the highest observed peak hourly runoff of 5.6 mm, the runoff duration was 32 hours and total runoff was 41 mm. The highest intensity was observed in the first two hours of the event and the runoff coefficient calculated for the first 2 hours of the rainfall event was 18%. . The RC calculated for the duration this event was 68%. These are large compared with the maximum surface saturated extent throughout the study period of about 6% of the catchment area, indicating processes other than saturation excess runoff are important. The event on 10/12/2011 marked the end of a particularly rainy period, with more than 200 mm over 30 days. Note that due to flow measurement equipment being removed after this event, it was the last recorded at RBF.

Table 3 provides a summary of the average conditions for each group of events. In terms of rainfall characteristics, the events that produced runoff (groups 2 and 3) tended to have higher total rainfall, with the highest intensity events also having the largest total rainfall. Of course the grouping criteria means larger groups are more likely to fall into group 2 compared with group 1. Peak rainfall intensities were both low and almost identical for groups 1 and 2. The runoff behaviour is quite different with almost no flow and very low runoff coefficients on average for group 1 and average runoff coefficients of 17.9 and 25.8 for groups 2 and 3 respectively. These runoff coefficients are clearly much higher than the observed maximum saturated area proportion (6%).

We also undertook a bootstrap analysis to test for differences between the group means. This showed that groups 2 and 3 are statistically different from group 1 at the 1% significant level for total event runoff, quick flow and the quickflow runoff coefficient. Groups 1 and 2 (which are distinguished by *ASI*+Rain) are statistically similar in terms of $I_{peak}$, suggesting rainfall intensity does not explain the runoff response differences between these two groups. Groups 1 and 3 (which are distinguished by $I_{peak}$) are statistically similar in terms of *ASI*+Rain, suggesting that catchment wetness does not explain the runoff response differences between these two groups.

Overall these results demonstrate a range of different rainfall runoff responses. The responses depended on both the catchment wetness as quantified by *ASI*+Rain and on the peak hourly rainfall intensity, $I_{peak}$ , with thresholds of 250mm and 15mm/h being identified. Runoff was produced whenever thresholds in either of these were exceeded.

### 3.3 Runoff processes and thresholds

The above presentation of results from Figure 5d identifies a threshold catchment wetness expressed as antecedent soil water storage plus event rainfall depth of 250 mm above which runoff always occurred and another threshold of hourly rainfall

depth exceeding 15 mm which also led to runoff production. It should be noted that there is some uncertainty in the $I_{peak}$ threshold as the most intense event (for $ASI$+Rain<250mm) that failed to produce runoff was 10mm/h, while the least intense event that did produce runoff was 14.8mm/h. Looking at events in the lower right quarter of Figure 5d also shows that the event runoff coefficient tends to increase as either catchment wetness or peak hourly intensity increases. In fact, runoff and

non-runoff producing events are very well separated by the relationship $I_{peak} = 3/11*(ASI-260)$. These results suggest that there are both wetness dependent and intensity dependent runoff production mechanisms operating. This section examines the evidence for different runoff mechanisms contributing to event runoff.

3.3.1 Catchment wetness-flow response relationhips

Figure 6 shows the runoff time series together with water level time series at several shallow piezometers; sites 4, 5, 3, 32, 2

and 1 (Figure 2).  All sites except 4 were located in drainage lines. In Figure 6, the sites are ordered by elevation from highest to lowest in the catchment. Manually read sites are shown with dashed lines. Figure 6 clearly shows that the water table became shallower and that the occurrence of profile saturation (water table at the surface) was restricted to the riparian zone.  Within the riparian zone, sites 1, 2 were saturated much of the time and site 32 quickly became saturated during events. The water table remained below the surface at sites 3 (at the upstream end of the riparian area), 4 and 5 (in Mid slope

positions). Comparing the runoff time series with the piezometer record for sites 1, 2 and 32, it is clear that the water table rose to the surface in the upper parts of the riparian zone during runoff events. Furthermore, the lower half of the riparian zone remained saturated to the surface for long periods during the runoff season. The data recorded at site 3 indicates that the water table at this site did not rise to the surface, even during events.

Figure 6 goes about here

Looking at the discharge record in Figure 6a, there were periods where significant baseflow persisted between events. These correspond to periods where the water table at site 5 was above about 120 cm and at site 4 was above about 140 cm deep. Flow became more strongly persistent between rainfall events as the water table at sites 4 and 5 rose further. The water table recessions at sites 4 and 5 correspond with flow recessions when the water table was above 120 cm and 140 cm at sites 4 and 5, respectively (Figure 6b).

So far we have examined thresholds in catchment wetness ($ASI$+Rain) associated with a change in runoff behaviour and the linkage between shallow water tables and catchment response.  It is probable that the linkage between catchment wetness and runoff is through the intermediary of shallow subsurface flow controlled by the existence of saturated conditions within a permeable part of the soil horizon. Figure 7 shows the relationship between water table levels in the catchment (sites 4 and 5) and soil water storage at the weather station. These sites represent planar and convergent mid slope positions respectively.

These sites were chosen because they show significant dynamics and we had logged records available. Other sites with loggers had limited dynamics or short records. Site 5 in particular shows a rapid change in behaviour for soil water storage around 250 mm, which corresponds with the $ASI$+Rain threshold identified above. As soil water storage moves above this level, much higher water tables develop and those water tables showed relatively rapid recession when shallower than 120

cm. Similar observations were seen at site 4 but the corresponding depth was 140 cm. Figure 7 thus explains the linkage between the 250mm *ASI*+Rain threshold and runoff. When soil water storage exceeded this level, water tables rose and lateral subsurface drainage occurred, as evidenced by the recessions. The recessions in particular suggest that the water table moved into a more permeable zone on these occasions. This is consistent with soil profiles being characterised by mottled

clay at depth which is likely to have lower hydraulic conductivity. This behaviour at these two wells in combination with the soil water and flow data, indicate that the hillslopes were becoming connected to the catchment outlet via subsurface flow from the hillslope to the riparian zone. Our analysis of isotope data below will add to the evidence for this. There were a few occasions where the water table responded strongly for soil water storage less than 250 mm (Figure 7). As indicated by the red colour, these corresponded to high intensity ($I_{peak}$>15mm/h) rainfall events.

Figure 7 goes about here

Flow recessions provide information on the drainage characteristics of catchments. Figure 6 shows that the catchment flow usually ceased between events during the wet period, with no flow during dry periods. However, in August and early

September 2010, continuous catchment flow endured for a month (Figure 6). There was also a marked variation in the recession behaviour during August/September 2010 and at other times during the study period. To explore this, we calculated the recession constant, $k$ (as in $Q = Q_0 e^{-kt}$, where $Q_t$ is discharge at time $t$ during the recession period, $Q_0$ is the discharge at the beginning of the recession and $k$ is the recession constant). In fitting recessions, we target the period after the recession of the event flow, which typically ceased around 24 hours after the end of the rainfall event. This was judged

visually by a marked change in slope of the recession hydrograph plotted in semi-log space. Recessions were only fitted where a long enough period of reliable flow data (>=24 hours) were available. Data availability was sometimes limited by the commencement of another event or flow falling to very low (<0.05mm/h). $k$ is plotted against soil water storage at the start of the catchment flow recession for individual events (Figure 8). In total we could estimate values of $k$ for 20 events. The three events in Figure 8 coloured blue were from November 2011 from a relatively dry period in late spring when there

was a drying trend but where flow from the riparian area was just persisting. These were excluded from the fitting because, in our judgement, they appear to represent different behaviour. $k$ for each event and the soil water storage at the start of the recession limb of was negatively correlated ($R^2$=0.64) (Figure 8). $k$ decreased as soil water storage increased at a rate of about 0.1d$^{-1}$ for each 10mm increase in soil water storage. Considering this and the transient nature of flow during dry periods, it is clear that the wetter the catchment is, the slower the recessions are. By inference, this suggests greater (perhaps

more spatially extensive) subsurface connectivity is providing flows from the hillslope and maintaining catchment flow during wetter conditions.

Figure 8 goes about here

### 3.3.2 Isotope and major ion results

The hydrometric results presented in Figures 6, 7 and 8 suggest that subsurface flow is important in this catchment. Given this, we would expect the hydrograph to be dominated by "old" or pre-event water; however, the saturated area in the lowest parts of the catchment would also be expected to produce direct flows of "new" or event water. This is addressed here using isotope data. We faced some constraints in obtaining meaningful hydrograph separation results. Of the five events sampled, two events were missing rainfall samples, which prevented their analysis. For another two events the rainfall isotope signature was quite close to the typical low flow signature. Uncertainty estimates showed standard deviations for pre-event and event fractions being over 50% for these two events, therefore these events were also excluded. For the event on 12 August 2010, the standard deviations were approximately 12% and 9% for $^{18}O$ and $^{2}H$ respectively, which we judged to be acceptable.

Figure 9 shows the event from 12 August 2010 during the wettest part of the study period. The antecedent soil water storage at the beginning of this event was 274 mm and total rainfall was 17 mm. We used a pre-event low-flow sample from ~2days before the analysed event as the pre-event end member. We estimated the contribution at the time of each stream water sample. Over the study period $\delta^{2}H$ ($\delta^{18}O$) for rainfall varied between -7‰ and -83‰ (-1.4‰ and 12.6‰) and isotopic concentrations for low flows were highly damped. Low flow samples from the RBF flume before and after the event showed a $\delta^{2}H$ ($\delta^{18}O$) of -27‰ (-4.1‰) and rainfall for this event was strongly depleted (3 samples prior to and during the event $\delta^{2}H$ = -42, -67 and -57‰, $\delta^{18}O$ = -6.5, -9.8 and -8.2‰), compared with low flow. The runoff samples showed a very different isotopic concentration during the rising limb and the peak of the hydrograph ($\delta^{2}H$ =-43‰, $\delta^{18}O$ = -6.8‰) in comparison to antecedent low flow. This shows that the isotopic concentration moves significantly towards the rainfall sample concentrations. To estimate the overall event water contribution we first separated the hydrograph at each sampling time and then interpolated the fraction of event water between stream water sampling times and combined this with the discharge hydrograph to calculate the overall volume of event water. This analysis suggests that the percentage of rain becoming runoff based on $\delta^{2}H$ and $\delta^{18}O$ is 4.4% (3.4-5.4%) and 3.6% (3.0-4.2%), respectively. The figures in the brackets are 95% confidence intervals for uncertainty based on the hydrograph separation uncertainty only. These results suggest that precipitation on the saturated area generates direct runoff in amounts that are close to what would be expected (i.e. 100% runoff) given that the saturated area is around 5-6% of the catchment area.

Figure 9 goes about here

Another interesting event is the higher intensity ($I_{peak}$ = 15 mm hr$^{-1}$) event on 8/11/11. Major ion geochemistry data were available for this event. Figure 10a shows the typical relationship between discharge and chloride concentration, with samples from this event identified by red. Figure 10b shows the time series of chloride concentration along with the hydrograph. The first and second chloride samples respectively plot above and within the typical scatter of data on Figure 10a, while the remaining samples plot well below the typical variation in chloride concentration with flow (Figure 10a). Similar plots are shown for Sodium, Magnesium, Calcium and Potassium in the supplementary material Figure S1. Sodium

and Magnesium show similar behaviour to Chloride. Potassium also shows somewhat anomalous behaviour but with anomalously high concentrations. The relationship between discharge and $K^+$ is also different to $Cl^-$, $Na^+$ and $Mg^{2+}$ in that concentrations increase slightly as discharge increase above about 0.2mm/mm, rather than a decline. Calcium does not show anomalous behaviour.

Given the late spring timing of this event, the first sample probably reflects some evapoconcentration of solutes in the riparian area. The flow shows a rapid peak in response to the main rainfall burst followed by a sustained relatively low flow and a recession over the second half of the day suggestive of subsurface flow. Up until the end of the first flow peak (i.e. 0600), there had been 23.4mm of rainfall and 1 mm of runoff. This runoff volume is 4.3% of the rainfall volume, which is similar to the proportion of event water that became runoff for the 12 August 2010 event discussed above.

Figure 10 goes about here

## 4 Discussion

### 4.1 Runoff mechanisms

The hydrometric data enables us to identify the important runoff mechanisms under different circumstances. The isotope and major ion geochemistry data provide further supporting evidence. The rainfall plus antecedent soil water threshold of 250

mm that needs to be exceeded for runoff in most circumstances shows that wetness dependent runoff processes are important, that is either saturation excess or subsurface stormflow. Shallow groundwater data combined with field mapping of surface saturated areas shows that complete profile saturation is limited to about 5% of the catchment area and this saturation is persistent with only a small variation (see table 1) over the winter-spring season but reduces over the dry summer-autumn period. The extent of the saturated area at the riparian zone varied seasonally and between events. Field

observations of surface saturation extending to the flume and well hydrograph measurements of the water table at the surface showed that the saturated area was highly connected to the catchment outlet and suggest that it would be expected to produce saturation excess runoff. This surface connection disappeared during the dry summer season when the water level in all wells fell below the surface (see Figure 6), so that the saturated area in the riparian zone disappeared. The isotope results for 12 August 2010 enabled the event runoff to be separated into event water and pre-event water contributions. Four to five

percent of the rainfall volume on the catchment appeared in the event runoff, which corresponds well to the proportion of surface saturated area in the catchment (5.5%, Table 1), supporting the identification of significant saturation excess runoff from this part of the catchment, as observed elsewhere (McGlynn and McDonnell, 2003;Penna et al., 2016).

While saturation excess runoff undoubtedly occurs, many of the event runoff coefficients were well in excess of 5% and they approach 100% under very wet conditions (Figure 5). The event on 12 August showed a substantial pre-event water

contribution; logged shallow wells show that the water table did not reach the surface in the steeper areas of the catchment, even within the convergent drainage lines under very wet conditions (e.g. sites 5, 7). The recession behaviour of wells in the catchment suggests subsurface flow moves down the catchment under wet conditions and the recession constant analysis

shows that this connection becomes stronger as the catchment wets beyond 250 mm of stored water. This is all consistent with a substantial contribution of subsurface flow to event runoff once the catchment is sufficiently wet to establish subsurface connection with the riparian area, as has been inferred in other studies (Buttle et al., 2004;Detty and McGuire, 2010;Hewlett and Hibbert, 1967;Jencso et al., 2009;Penna et al., 2011).

Perhaps more surprisingly there was a group of events that produced runoff under conditions of relatively low soil water storage (*ASI*+Rain < 250 mm) but high rainfall intensity. This suggests that an intensity dependent runoff process is being triggered when rainfall exceeds some threshold for sufficient time, in this case about 15 mm of rainfall in an hour. It is worth noting that the 15mm/h threshold will be dependent on the time step used in calculating the intensity. The choice of time step needs to consider the travel time from hillslope to stream as runon infiltration processes occurring on the catchment surface

will impact on the connectivity to the stream. Thus the most appropriate time to use for averaging intensity would be the average travel time to the stream because it is this time period which is available to infiltrate rainfall that has become runoff at the point scale. Here the choice of hourly rainfall was pragmatic as that was the recording time step of the AWS but it is also likely that it reasonably approximates the flow times.

It is tempting to assume that this evidence suggested surface runoff occurred due to infiltration excess runoff, but it is also

possible that the high rainfall intensities are efficiently activating macropore networks (Beven and Germann, 1982) and that the flow could be following subsurface pathways. For these events there were also rapid responses in water levels in wells on the hillslope and it is not clear exactly how the water moved rapidly into the wells in these cases but it could be due to preferential flow through macropores. Furthermore for the high intensity event on 8 November 2011, the flow after the main peak had a surprisingly low concentration of chloride (the cluster of low concentration red points on Fig 10a), which may

suggest that the higher intensity activated either overland or preferential flow paths, limiting soil contact time and leading to this low concentration.

The hydrograph from the event on 8/11/11 (the event that exceeded both thresholds, Figure 10) shows both a rapid and a delayed runoff response. The concentration of chloride was unusually low during the delayed runoff component compared with all other events (with major ion data available). This may suggest limited contact with the catchment soils which could

occur if macropore flow was important but this explanation is not definitive. Of the two events with peak hourly intensities around 15 mm hr$^{-1}$, one also exceeded the wetness (*ASI*+Rain) threshold of 250 mm and the other had a very low runoff coefficient (only 3%) and hence these two events are somewhat equivocal in terms of the importance of intensity. However, the two events with peak hourly rainfall intensities around 30 mm hr$^{-1}$ both produced rapid runoff responses without a significant delayed component (Figure 4) and had runoff coefficients (12 and 68%) well in excess of the surface saturated

area (5%) in the catchment, showing clear evidence of the role of rainfall intensity of 30 mm hr$^{-1}$. Unfortunately isotope and major ion data were not available for those events to attempt to determine whether surface or subsurface pathways are important. Overall there is clear evidence for intensity dependent runoff mechanisms, especially for the largest 30 mm hr$^{-1}$ events.

In summary, the process evidence relating runoff behaviour with catchment wetness thresholds, together with the data from shallow wells suggests a catchment where subsurface flow leads to a seasonally saturated riparian area that produces saturation excess runoff in immediate response to rainfall. This saturation excess is augmented by subsurface stormflow when the catchment wetness (*ASI*+Rain) exceeds a 250 mm threshold. This subsurface flow exfiltrates in the riparian area.

Volume considerations provide evidence that, for many of the monitored events, water must be coming from outside the riparian area. This area is only about 5% of the catchment but quickflow runoff coefficients very regularly exceed this, often (about 1/3 of runoff events) exceeded 20%, and on occasion exceeded 50%. The role of both saturation excess runoff and subsurface stormflow is corroborated to some extend by isotopic data from one event where the hydrograph separation shows a volume of event water similar to the volume of rainfall on the saturated area and by the highly damped isotopic

signal in the stream in general. When hourly rainfalls exceed a threshold of 15-30 mm hr$^{-1}$, an intensity-dependent runoff process is activated that also contributes flow from the hillslope area outside the riparian zone. It is not clear whether this is a purely surface runoff process or not. One of the key contributions of this work is clear hydrometric evidence of the interplay of both wetness and intensity dependent runoff processes in the one catchment. These responses include two wetness dependent processes: saturation excess runoff and subsurface stormflow, and an intensity dependent processes for which we

can't distinguish between surface and subsurface flow pathways with our data.

## 4.2 Thresholds and connectivity in runoff production

We identified two important thresholds in the catchment response. The first is a wetness threshold of *ASI*+Rain exceeding 250 mm. Under these conditions the water table approaches the surface in the riparian area and water tables rise on the hillslope into what is inferred from relatively rapid hillslope water table recessions to be a more transmissive part of the soil

profile (within ~120-140 cm of the surface). Based on the volume of event runoff (Table 1), the lack of surface saturation on the hillslope and the behaviour of shallow wells (Figure 6), we infer that the hillslope becomes connected to the riparian zone under these conditions. The existence of a large volume of pre-event water in the event on 12 August 2010 (Figure 8) adds further corroboration to this. Similar catchment wetness thresholds for connectivity and runoff generation have been reported elsewhere (Detty and McGuire, 2010;Penna et al., 2011;Tromp-van Meerveld and McDonnell, 2006b). These have

been expressed either in terms of rainfall depth, antecedent soil water storage conditions, or a combination of these. Similar to our results, in a study of a forested subcatchment with highly permeable soils and a small riparian area, Detty and McGuire (2010) found a relationship between a threshold of 316 mm for *ASI*+Rain and the start of the event streamflow response. They showed that the *ASI*+Rain threshold corresponded with a water table height threshold. Based on this, they suggested that subsurface flow, transmissivity feedback and preferential flow from hillslope to stream could be used to

explain runoff mechanisms in their catchment. They did not observe either Hortonian overland flow or SOF even during the largest events.

A second threshold to initiate event streamflow response associated with high rainfall intensities was also evident. A similar role of intensity has also been observed by Janzen and McDonnell (2015) who found that the Panola hillslope can produce

significant runoff from dry antecedent conditions when high intensity rainfall occurs. In general event runoff from Panola is controlled by catchment wetness, similar to our catchment. Mixed wetness and intensity dependent runoff processes have also been reported in Mediterranean catchments (e.g. Lana-Renault et. al 2014). Nevertheless there are only a few studies we know of that have reported intensity thresholds in catchments where runoff is normally dominated by wetness thresholds.

This may be a consequence of such events being relatively rare in any given catchment (roughly 10% of runoff producing events in our case).

### 4.3 Runoff processes framework

We now consider the three runoff processes occurring in the catchment in relation to the framework proposed in Figure 1 and the various flux and timescale thresholds identified therein. Essentially Figure 1 is posing a series of questions that

allow us to systematically think through the runoff processes. Above we have identified three groups of rainfall events: those that do not produce runoff; those that produce runoff by saturation excess and subsurface stormflow from a wet catchment and those that produce runoff from higher intensity events.

The rainfall events that do not produce runoff are not exceeding infiltration capacity for sufficient time for runoff to flow from the catchment (box 1, "No"). They may or may not produce significant percolation (box 2) but if any of these events do

produce percolation to a perched water table, this only results in an ephemeral water table that dissipates before lateral flow can move water down the hillslope (box 3, "No") and they do not saturate the full profile (box 4, "No"). As a consequence no event runoff is produced. These conditions correspond to the local control state of Grayson et al. (1997).

Some high intensity rainfall events on a dryer catchment do exceed the infiltration capacity for sufficiently long periods of time (box 1, "Yes"; hourly intensity of 15-30 mm hr$^{-1}$) and these events produce runoff. The increase in runoff coefficient as

intensity increases for a given *ASI*+Rain in Figure 5 suggests that infiltration thresholds may also be playing a role for wetter conditions (i.e. box 1 "Yes) but that infiltrated water (the dashed link in Figure 1) also contributes through other mechanisms.

The final group of events are those that produce runoff from a wet catchment at low intensity rainfall. These follow the path box 1 "No", box 2 "Yes" and box 3 "Yes" in Figure 1. On the upper hillslope segments of the catchment, the subsurface

flow capacity is sufficient so that the water table does not reach the surface and water drains either during or shortly after the storm (box 6, "Yes"). Subsurface connectivity develops during the event and subsurface flow dominates. The shallow well data (Figure 6, wells 1 and 2, to some extent 32) shows that the riparian area in the lower part of the catchment drains very slowly. There is a substantial reduction in slopes (and probably lower hydraulic conductivity associated with poorly structured, poorly drained soils) that suggests a substantial reduction in lateral subsurface flow capacity. The surface

topography also suggests subsurface flow would converge in this area. From the perspective of Figure 1, this leads to a situation where water is taking longer than the typical time between events in the wet season to drain (box 7, "Yes"), resulting in persistent surface saturation and saturation excess runoff generation from the lower catchment. Hence under wet

conditions this catchment produces a mix of saturation excess and subsurface storm flow, but from geographically distinct parts of the catchment.

The above illustrates how the framework in Figure1 can be used to understand the role of different thresholds regarding fluxes and timescales in determining runoff mechanisms. Such a framework is likely to be particularly valuable where there is a mix of runoff mechanisms operating for different events or in different parts of the catchment. Our study catchment nicely illustrates such a mixture.

## 5 Conclusion

This study has examined the role of intensity and wetness thresholds in determining runoff responses for an agricultural catchment in the Lang Lang River catchment, Victoria, Australia. Both intensity dependent and wetness dependent thresholds were identified in the runoff response. During wet conditions, hydrological connectivity has a strong influence on water delivery to the riparian area. Saturation excess runoff from the riparian zone was also important. The results of this study demonstrated that:

1) Runoff generation in most events is dependent on the catchment connectivity and soil moisture conditions. When the sum of the antecedent soil water storage and event rainfall exceeded 250 mm, runoff was typically produced by a mix of saturation excess and subsurface storm flow. Under these conditions, a water table forms in the soil and a saturated area develops in the riparian zone. When the water level rises to within about 1 m of the surface at mid slope sites, rapid subsurface flow pathways are activated which connected the mid slope and riparian area, contributing event flow to the flume at the catchment outlet.

2) When the catchment became very wet, high water levels persisted at the mid slope sites which remained hydrologically connected to the riparian area and baseflow became persistent between events.

3) High rainfall intensity events produced runoff even when the antecedent soil water storage (ASI) plus event rainfall depth was below the 250 mm threshold. This could be due to either Hortonian overland flow or fast subsurface preferential flow paths being activated.

We have also advanced a set of threshold conditions or questions (Figure 1) that allow a logical examination of which runoff mechanism are likely to be important in a catchment given, thresholds regarding fluxes and timescales. This framework provides a useful way of thinking through the controls on rainfall-runoff response as conditions change either between events or between different parts of the catchment. It is illustrated using the behaviour of this catchment. Our study catchment demonstrates a mix of intensity dependent and wetness dependent processes, something which has been rarely reported for humid catchments.

## Acknowledgements

The project was funded by the Australian Research Council (grant DP0987738). Saffarpour was supported by an International Postgraduate Research Scholarship (IPRS) from the Australian Government and an International Research Scholarship from the University of Melbourne (MIRS). Peter and Wilma Mackay provided access to their farm for the study. Many people helped in the field, especially Rodger Young, Lucas Dowell, Olaf Klimczak and Jacqui Lloyd. The water samples were analysed for $^{18}$O and $\delta^2$H, either by Ian Carwright's Earth Sciences laboratory, Monash University or by Russell Drysdale's isotope laboratory at the University of Melbourne. The chloride data were analysed by Mike Grace's Water Studies Laboratory, Monash University. Tony Weatherly assisted with soil interpretation at the site.

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

**Table 1. Rainfall-runoff events summary at RBF.** *ASI* is the Antecedent Soil moisture Index. RC is the quickflow runoff coefficient. Runoff event grouping is discussed in section 3.2.

| Group No. | Date | Rain duration (hr) | Total rainfall depth (mm) | Peak hourly rainfall intensity (mm/hr) | *ASI* (mm) | *ASI*+ rainfall depth (mm) | Runoff duration (hr) | Total runoff depth (mm) | Event runoff depth (mm) | RC (%) | Saturated area (%) |
|---|---|---|---|---|---|---|---|---|---|---|---|
| 2 | 25/06/2010 | 49 | 22.4 | 1.8 | 237 | 259 | 74 | 7.6 | 1 | 4 | 4.5 |
| 2 | 13/07/2010 | 33 | 23.8 | 5.4 | 237 | 261 | 41 | 3.5 | 0.54 | 2 | |
| 2 | 20/07/2010 | 15 | 9.2 | 4.2 | 272 | 281 | 32 | 3.1 | 0.8 | 9 | 5.3 |
| **2** | **1/08/2010** | **30** | **31.4** | **6.2** | **253** | **284** | **67** | **19.5** | **13.7** | **44** | **5.5** |
| 2 | 5/08/2010 | 52 | 23.6 | 2 | 275 | 299 | 96 | 21.6 | 7 | 30 | 5.5 |
| 2 | 12/08/2010 | 34 | 17.2 | 5 | 274 | 291 | 94 | 12.7 | 4.3 | 25 | 5.5 |
| 2 | 16/08/2010 | 50 | 19.2 | 3.4 | 275 | 294 | 73 | 16.9 | 7.7 | 40 | 5.5 |
| 2 | 18/08/2010 | 34 | 14.6 | 4 | 286 | 301 | 57 | 13.8 | 4.6 | 32 | 5.5 |
| 2 | 24/08/2010 | 4 | 9.6 | 5.2 | 273 | 283 | 24 | 4.1 | 1.6 | 17 | 5.5 |
| 2 | 25/08/2010 | 59 | 12.6 | 2 | 285 | 298 | 73 | 10 | 1.4 | 11 | 5.5 |
| 2 | 31/08/2010 | 22 | 12.4 | 2 | 271 | 283 | 80 | 9.5 | 0.8 | 6 | 5.5 |
| 2 | 5/09/2010 | 47 | 25.2 | 3.4 | 272 | 297 | 74 | 23.1 | 13.5 | 54 | 5.5 |
| 2 | 9/09/2010 | 17 | 8.0 | 2.4 | 264 | 272 | 16 | 1.1 | 0.2 | 3 | 5.3 |
| 2 | 6/10/2010 | 18 | 15.2 | 6.6 | 231 | 246 | 3 | 0.37 | 0.27 | 2 | na |
| 2 | 15/10/2010 | 59 | 34.2 | 2.8 | 228 | 262 | 86 | 11.3 | 3.9 | 11 | na |
| 2 | 23/10/2010 | 11 | 12.8 | 5.2 | 227 | 240 | 8 | 1 | 0.57 | 4 | na |
| 2 | 30/10/2010 | 29 | 32.8 | 8.8 | 202 | 235 | 16 | 1.2 | 0.83 | 3 | na |
| **3** | **27/11/2010** | **19** | **54.4** | **30.4** | **161** | **215** | **31** | **7.6** | **6.5** | **12** | **na** |
| 2 | 19/12/2010 | 49 | 25.6 | 3.8 | 191 | 217 | 10 | 0.83 | 0.18 | 1 | na |

Group 2 has *ASI*+Rain>250mm, Group 3 has $I_{peak}$>15mm/h. "na" stands for "not available".

**Table 1. cont Rainfall-runoff events summary at RBF. ASI is the Antecedent Soil moisture Index. RC is the quickflow runoff coefficient. Runoff event grouping is discussed in section 3.2.**

| Group No. | Date | Rain duration (hr) | Total rainfall depth (mm) | Peak hourly rainfall intensity | $ASI$ (mm) | $ASI$+ rainfall depth (mm) | Runoff duration (hr) | Total runoff depth (mm) | Event runoff depth (mm) | Runoff Coefficient (%) | Saturated area (%) |
|---|---|---|---|---|---|---|---|---|---|---|---|
| 3 | 4/02/2011 | 33 | 71.2 | 16.4 | 130 | 201 | 10 | 2.5 | 2 | 3 | na |
| **2** | **11/05/2011** | **122** | **63.6** | **6.6** | **220** | **284** | **47** | **5.6** | **2.2** | **3** | **na** |
| 2 | 22/05/2011 | 50 | 21.6 | 4.4 | 231 | 253 | 3 | 0.23 | 0.1 | 0.46 | na |
| 2 | 26/05/2011 | 37 | 13.6 | 1.6 | 244 | 258 | 33 | 2.66 | 1 | 7 | na |
| **2** | **7/06/2011** | **25** | **17.6** | **2.6** | **254** | **272** | **43** | **6.7** | **2.9** | **16** | **na** |
| 2 | 17/06/2011 | 31 | 15.6 | 2.2 | 248 | 264 | 45 | 3.2 | 0.2 | 1 | 4.6 |
| 2 | 21/06/2011 | 61 | 38.8 | 3.6 | 255 | 294 | 80 | 21.8 | 12.1 | 31 | 4.6 |
| 2 | 5/07/2011 | 10 | 9.8 | 2.4 | 253 | 263 | 26 | 2.9 | 0.8 | 8 | 4.7 |
| 2 | 6/07/2011 | 12 | 16.6 | 6.2 | 267 | 284 | 29 | 8.7 | 4.8 | 29 | 4.7 |
| 2 | 10/07/2011 | 20 | 7.4 | 2.2 | 262 | 269 | 27 | 3.7 | 1.4 | 19 | 4.7 |
| 2 | 28/09/2011 | 74 | 72.0 | 8.8 | 204 | 276 | 96 | 20.1 | 6.1 | 8 | 5.4 |
| 2 | 9/10/2011 | 54 | 23.2 | 2.4 | 232 | 255 | 5 | 0.35 | 0.14 | 1 | na |
| 2 | 28/10/2011 | 19 | 30.2 | 7.8 | 213 | 243 | 6 | 0.92 | 0.6 | 2 | na |
| **3** | **8/11/2011** | **12** | **35.2** | **14.8** | **222** | **257** | **31** | **10** | **7.1** | **20** | **5.2** |
| 2 | 9/11/2011 | 14 | 23.8 | 6.8 | 239 | 263 | 44 | 15.7 | 8.9 | 37 | 5.2 |
| 2 | 18/11/2011 | 30 | 45.5 | 10.2 | 219 | 265 | 82 | 17.7 | 8.9 | 20 | 5.5 |
| **2** | **26/11/2011** | **29** | **36.8** | **8.4** | **216** | **253** | **85** | **27.6** | **16.3** | **44** | **5.5** |
| 2 | 29/11/2011 | 47 | 19.4 | 4 | 233 | 252 | 70 | 11.9 | 0.9 | 5 | 5.5 |
| **3** | **10/12/2011** | **16** | **52.6** | **31** | **192** | **245** | **32** | **41.1** | **35.6** | **68** | **5.5** |

5    Group 2 has $ASI$+Rain>250mm, Group 3 has $I_{peak}$>15mm/h. "na" stands for "not available".

**Table 2. Rainfall events summary at RBF with no runoff (Group 1 events). *ASI* is the Antecedent Soil moisture Index.**

| Date | Rain duration (hr) | Total rainfall depth (mm) | Peak hourly rainfall intensity (mm/hr) | *ASI* (mm) | *ASI*+ rainfall depth (mm) | Runoff duration (hr) | Total runoff depth (mm) |
|---|---|---|---|---|---|---|---|
| 12/10/10 | 5 | 5 | 3 | 227 | 232 | 0 | 0 |
| 12/10/2010 | 5 | 5 | 2.6 | 227 | 232 | 0 | 0 |
| 12/11/2010 | 32 | 28.2 | 4.0 | 184 | 212 | 0 | 0 |
| 25/11/2010 | 17 | 13 | 3.6 | 157 | 170 | 0 | 0 |
| 2/12/2010 | 8 | 7.2 | 2.0 | 191 | 198 | 0 | 0 |
| **8/12/2010** | **11** | **19.2** | **7.8** | **175** | **194** | **0** | **0** |
| 9/12/2010 | 6 | 5.2 | 2.4 | 193 | 198 | 0 | 0 |
| 17/12/2010 | 21 | 15 | 2.4 | 181 | 196 | 0 | 0 |
| 10/01/2011 | 18 | 10 | 3.2 | 146 | 156 | 0 | 0 |
| 11/01/2011 | 26 | 10.4 | 2.4 | 152 | 162 | 0 | 0 |
| 13/01/2011 | 26 | 22.2 | 9.0 | 151 | 173 | 0 | 0 |
| 25/01/2011 | 9 | 5.4 | 1.6 | 147 | 152 | 0 | 0 |
| 5/02/2011 | 12 | 8.6 | 1.6 | 182 | 191 | 0 | 0 |
| 16/02/2011 | 20 | 14.8 | 9.6 | 167 | 182 | 0 | 0 |
| 26/02/2011 | 13 | 13.4 | 4.2 | 174 | 187 | 0 | 0 |
| 21/04/2011 | 28 | 8.4 | 2.6 | 213 | 221 | 0 | 0 |
| 1/05/2011 | 8 | 10.8 | 2.6 | 207 | 218 | 0 | 0 |
| 8/05/2011 | 14 | 7.2 | 4.8 | 215 | 222 | 0 | 0 |
| 5/06/2011 | 18 | 9.4 | 3.0 | 238 | 247 | 0 | 0 |
| 9/09/2011 | 22 | 12.8 | 3.0 | 221 | 234 | 0 | 0 |
| 19/09/2011 | 18 | 12.2 | 4.0 | 218 | 230 | 0 | 0 |
| 24/10/2011 | 12 | 15.6 | 4.2 | 211 | 227 | 0 | 0 |

**Table 2. Average characteristics for each event group. *ASI* is Antecedent Soil moisture Index.**

| Group and Criteria | Event rainfall (mm) | $I_{peak}$ (mm/h) | ASI (mm) | ASI+Rain (mm) | Total runoff (mm) | Quickflow (mm) | Quickflow Runoff Coefficient (%) |
|---|---|---|---|---|---|---|---|
| 1: ASI+Rain<=250 and $I_{peak}$<15 | 14.0 | 4.3 | 194.1 | 208.0 | 0.2 | 0.1 | 0.4 |
| 2: ASI+Rain>250 | 24.1 | 4.6 | 250.0 | 274.2 | 10.6 | 4.5 | 17.9 |
| 3: $I_{peak}$>=15 | 53.4 | 23.2 | 176.3 | 229.5 | 15.3 | 12.8 | 25.8 |

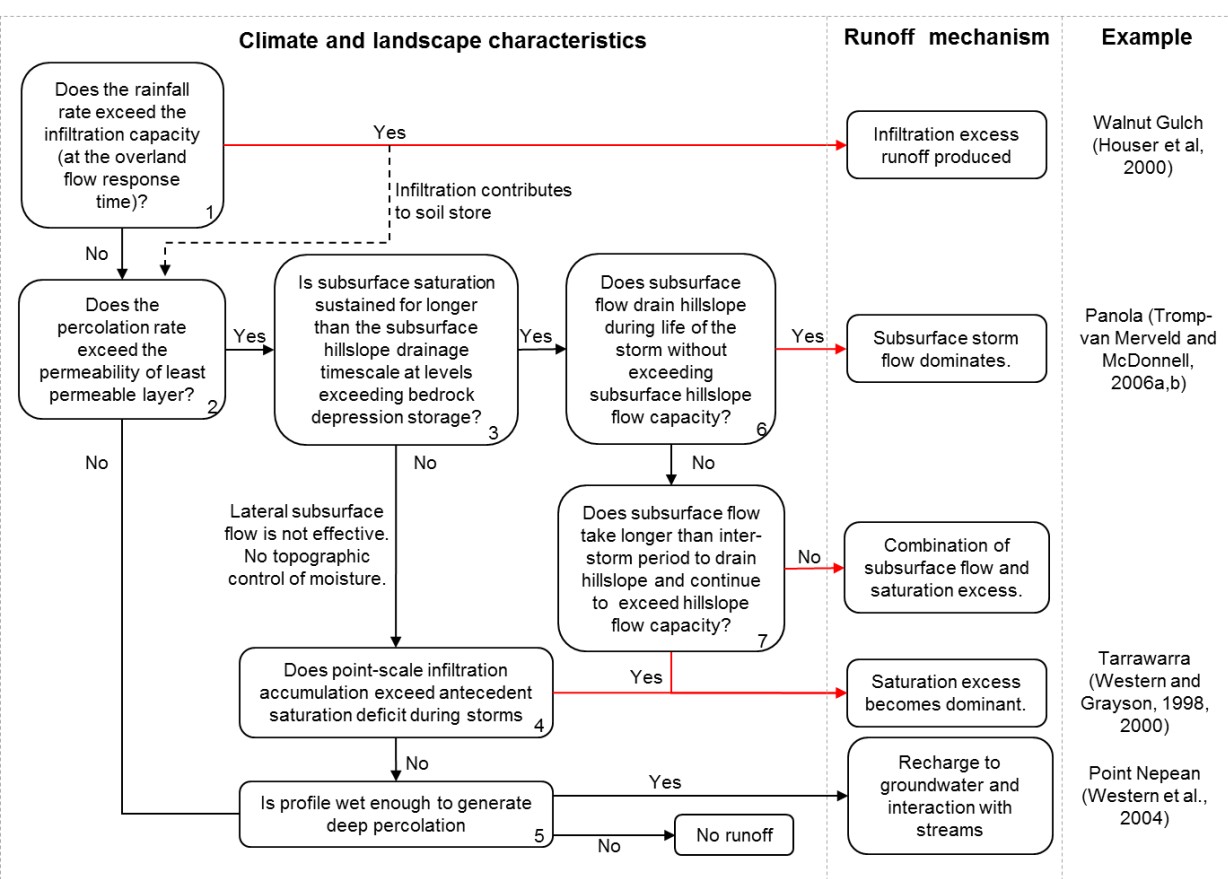

**Figure 1: The role of flux and timescale thresholds in determining runoff processes. The red lines indicate cases where there is surface or subsurface hillslope connectivity to the stream.**

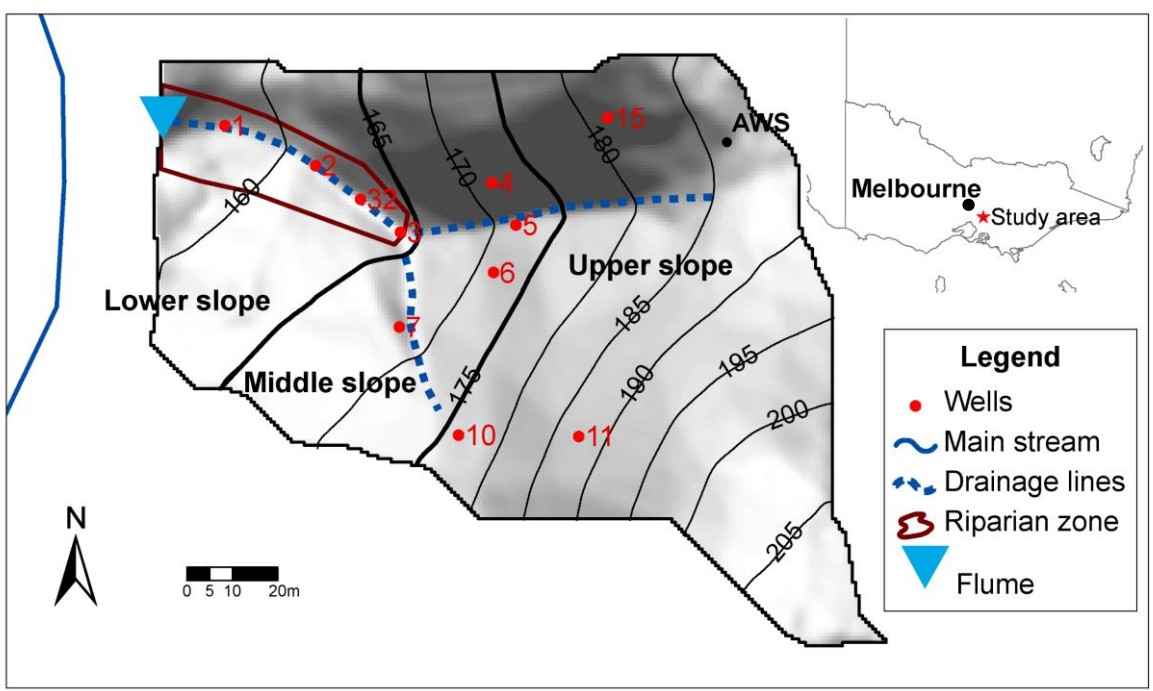

**Figure 2. The study site location within Australia and a hill shaded DEM, topography and sampling site locations at RBF. In this figure the black circles show shallow groundwater sites, the red circle shows soil moisture site and the blue triangle demonstrates the catchment flume.**

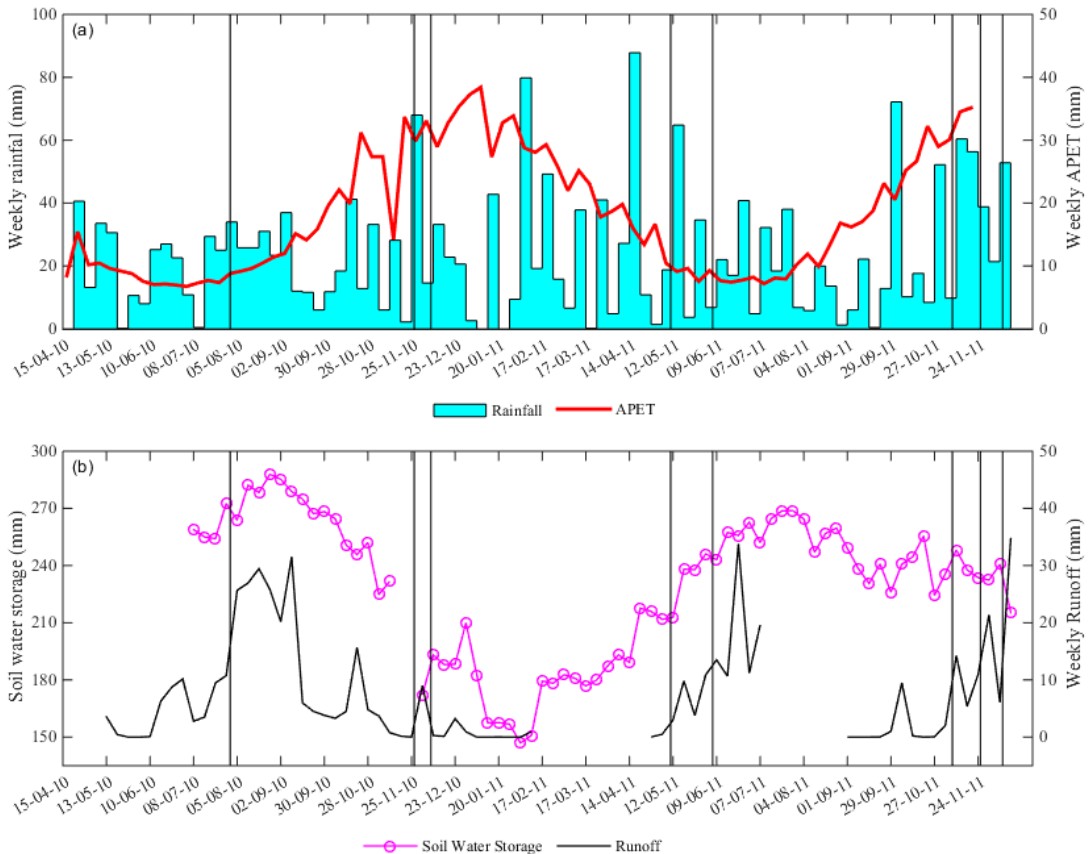

**Figure 3.** **(a) Weekly rainfall and APET (areal potential evapotranspiration) time series, (b) soil water storage in top 60 cm of the soil profile and weekly runoff time series for the flume at the catchment outlet. The rainfall, areal potential evapotranspiration and volumetric soil moisture, used to calculate soil moisture storage were all recorded at the AWS (automatic weather station) (Figure 1). The vertical bars show the timing of events in Figure 4.**

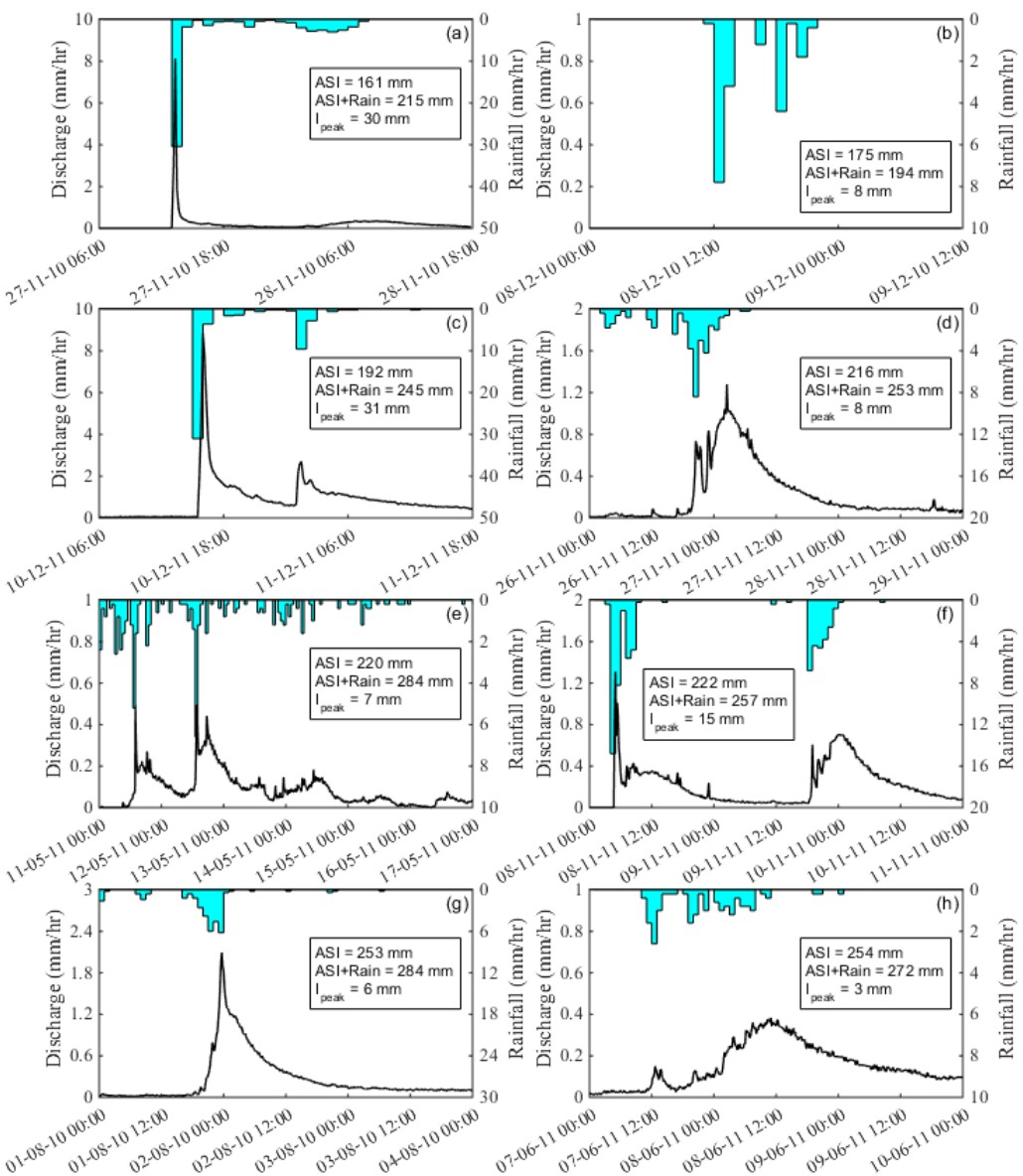

**Figure 4. Characteristics of 8 selected rainfall-runoff events sorted by their antecedent soil moisture index (*ASI*). Rainfall is shown in blue. It should be noted that the axis scales vary between events. All events (except those on 26/11/2011 and 10/12/2011) had zero or very low initial discharge. For the events on 26/11/2011 and 10/12/2011, the initial discharges were 0.07 and 0.13 mm hr⁻¹, respectively. Note that the axes vary between events.**

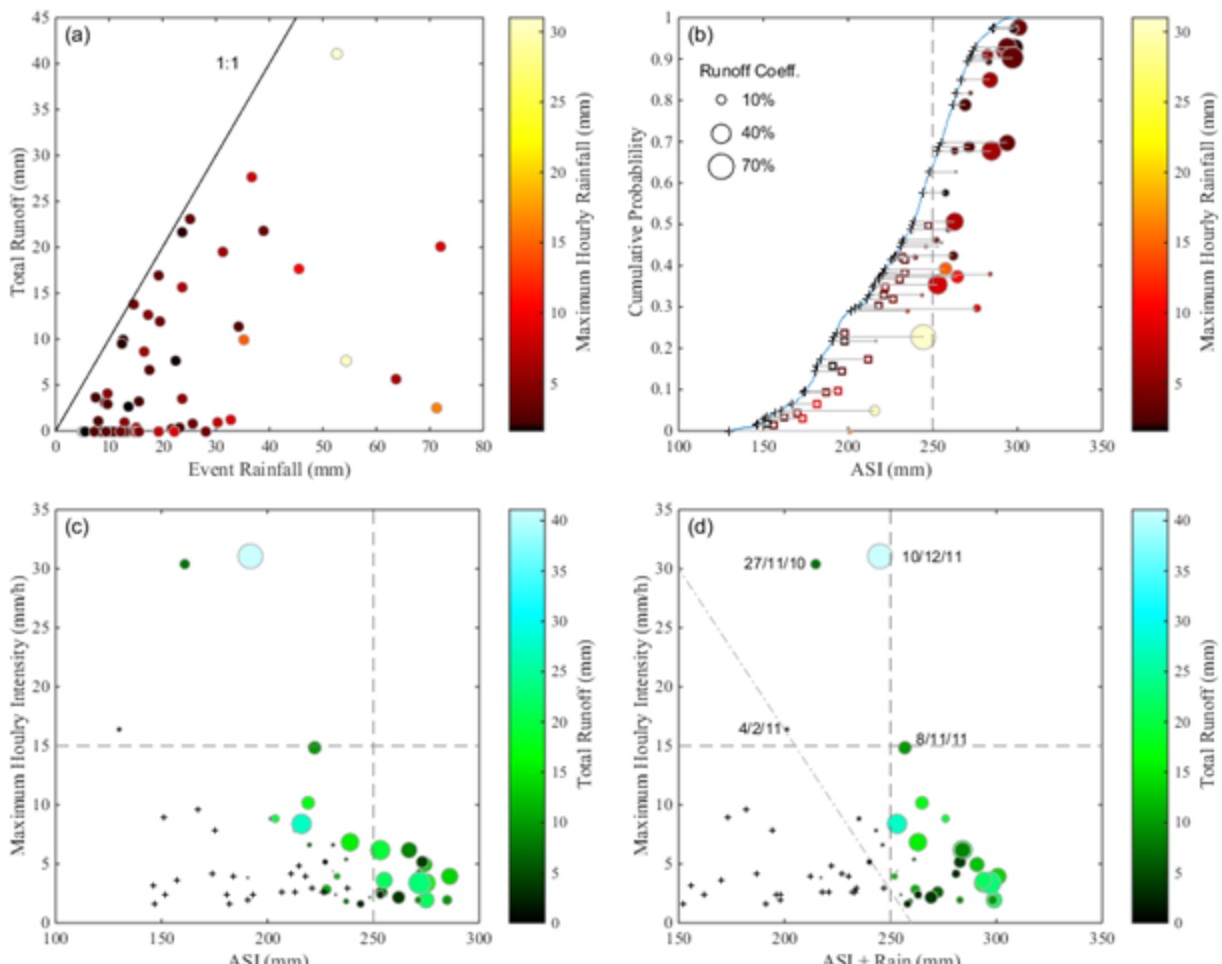

**Figure 5. Thresholds of runoff mechanisms at RBF, a) event rainfall versus total event runoff, colours indicate the highest hourly rainfall intensity, b) the impact of five factors together including: cumulative curve of the distribution of soil water storage as observed through the study period, *ASI*, *ASI*+Rain, colour shows the peak hourly rainfall intensity (*I*<sub>peak</sub>) and the size of the bubbles shows the event runoff coefficient, c) *ASI* versus the peak hourly rainfall intensity (*I*<sub>peak</sub>) and the size of the bubbles shows the event runoff coefficient and colour shows event total runoff, and d) *ASI*+Rain versus the peak hourly rainfall intensity (*I*<sub>peak</sub>) and the size of the bubbles shows the event runoff coefficient and colour shows event total runoff. Note *ASI* is the antecedent soil water storage (mm) in top 60cm of soil at the AWS at the beginning of the event.**

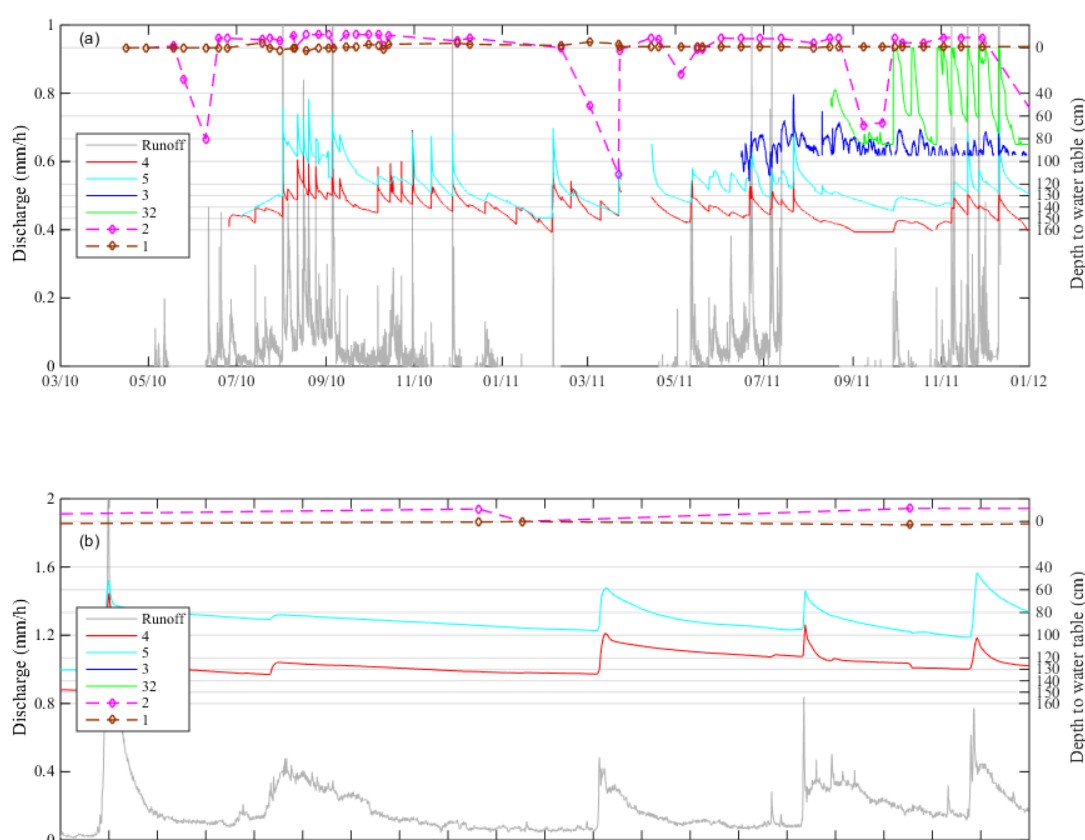

**Figure 6. Time series of discharge (grey lines) and groundwater levels at sites 4, 5, 3, 32, 2 and 1, manually read sites are shown with dashed lines and diamonds at measurement points. Levels above zero at sites 1 and 2 occur as the soils are highly pugged in the riparian zone and water pooled on the surface in places leading to slightly positive water levels being measured, at site 2 in particular.**

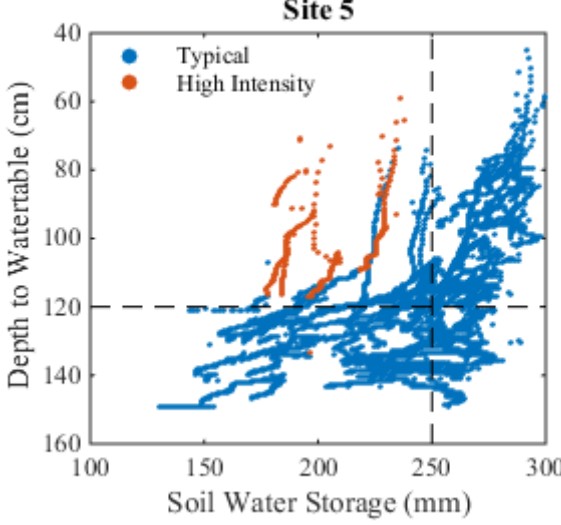

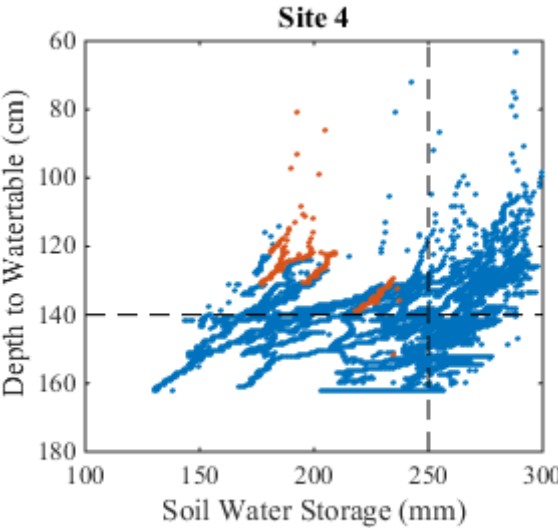

**Figure 7. Soil water storage versus water table level at sites 4 and 5. The red colour distinguishes events with high hourly rainfall intensity, defined as $I_{peak}$>15mm, consistent with Figure 5. The dashed black lines correspond to values of water level and soil water storage discussed in the text. Soil water storage is measured for the top 60cm at the Automatic Weather Station.**

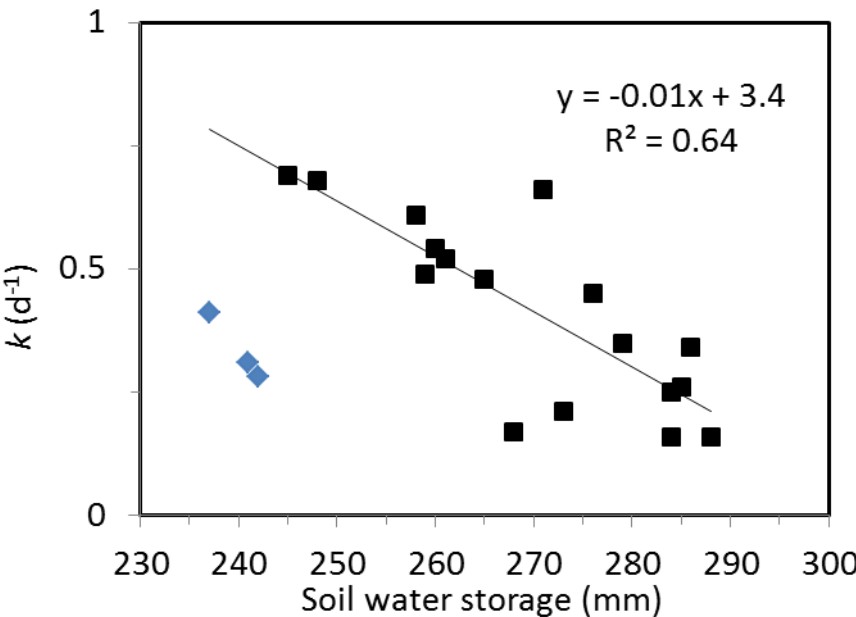

**Figure 8. Recession constant (*k*) and soil water storage the start of the recession for individual events. The choice of events is explained in the text, as are the three blue points.**

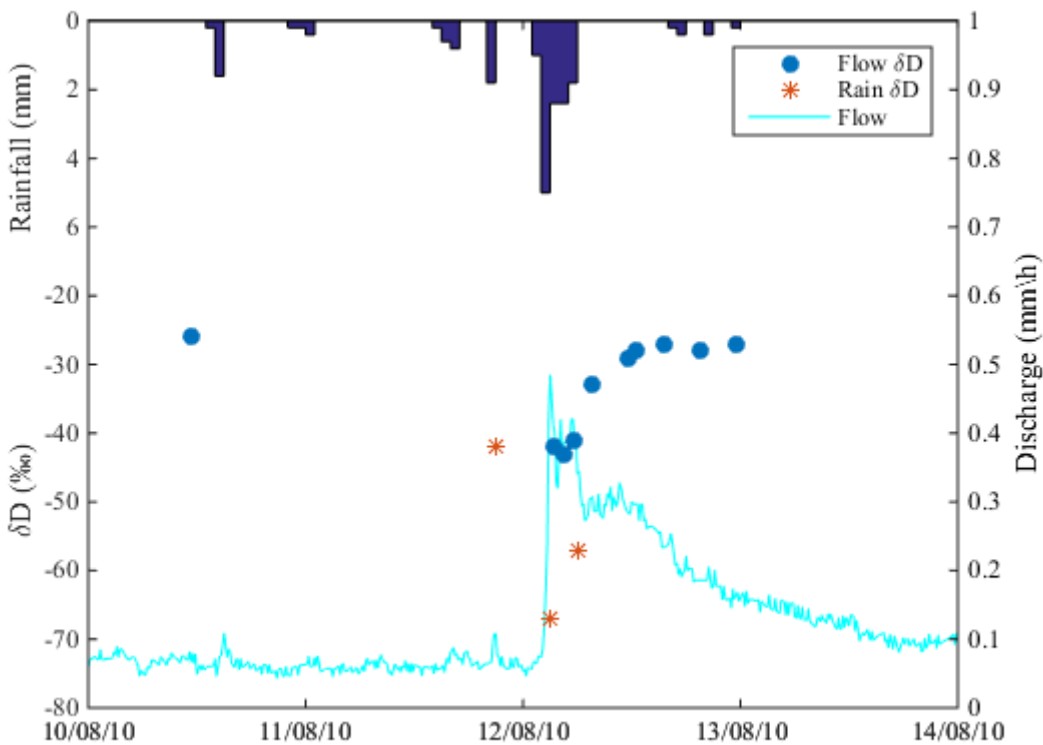

**Figure 9. Time series of total rainfall and discharge, [2]H of rainfall and runoff for the event on 12/8/10**

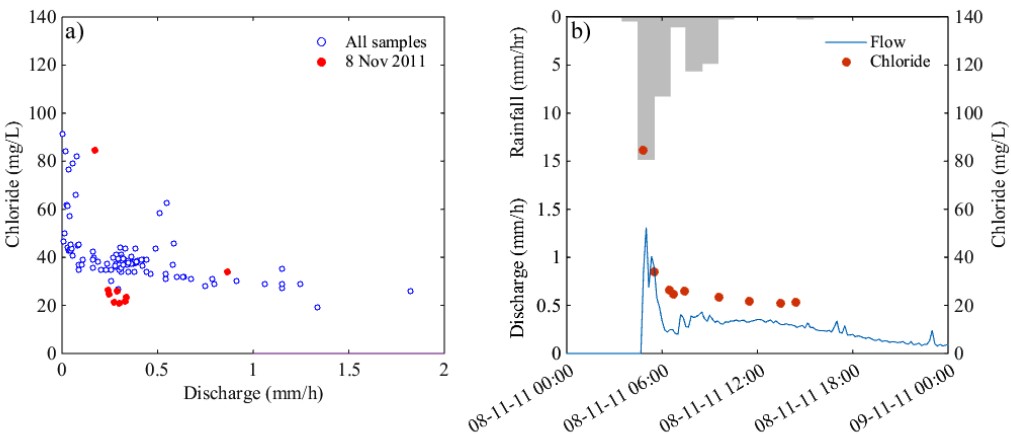

**Figure 10. a) Discharge versus Cl⁻ concentration for all events. The red colour identifies samples from event on 8/11/11, and b) Time series of rainfall, runoff and Cl⁻ concentration for the event on 8/11/11**