# Peer review of "Multiple runoff processes and multiple thresholds control agricultural runoff generation"

_Hydrology and Earth System Sciences, 2016_

## Referee Comment (RC1) · Anonymous Referee #1 · 13 Jul 2016

**General comment**

This is a very interesting work that focuses on the analysis of runoff processes and the controls exerted by different thresholds on the hydrological mechanisms related to runoff generation in an agricultural Australian catchment. The research aims to understand how subsurface connectivity, saturation excess and rainfall intensity play a role in rainfall-runoff response at the seasonal and event time scale. The manuscript is well written, logically organized and with overall clear graphical presentations. Results are generally well supported by data and interpretation are overall sound. I particularly like the conceptual summary of hydrological processes and thresholds reported in Fig 1, and how this figure was referred to in the Introduction and in the discussion. However, I think that there are some confused points that deserve to be clarified and better explained. I have some comments and suggestions that can hopefully help this paper to

have a greater impact on the hydrological community.

Specific comments

5, 24-26. As far as I understand, two different isotope laser analysers have been employed for the analysis of stable isotopes of water. This is a methodologically critical point: based on my experience, two different laser machines, even of the same model and calibrated using the same set of reference standards, could return quite different values of isotopic composition. Using a different sets of standards in different laboratories, as it seems that was the case here, could lead to differences that have the potentials to impact the resulting analysis of hydrograph separation. I think it is important for the paper to run some tests and report some comparison metrics between the measurements performed by the two machines in order to assess, and in case correct, potential deviations.

6, 10-12. It is not clear how the soil water storage has been computed starting from ASI. A specification, perhaps including equations, would be really useful here. This is important because the soil water storage is addressed several times in the rest of the manuscript. Moreover, it's not clear how manual measurements of the saturated are have been carried out. Please explain.

8, 2. The statement that 'any rainfall depth could produce a response' seems to contradict what reported elsewhere in the manuscript (eg, 7, 4-5; 8, 14; 8, 26) about some rainfalls that did not produce runoff. This is confusing and should be clarified.

8, 6-16. Fig. 5b seems to be dense and informative. However, I think that is not straightforward to understand it. The different symbols are hard to distinguish and the scale of runoff coefficients is not very useful to understand their values. I suggest considering to replace it by another graphical way (eg, cumulative distribution + bar plot or multiple panel boxplot). Moreover, I don't understand why ASI and ASI+rain have been plotted against rainfall intensity: the relation is obviously scattered (and so the sentence at 8, 12 is obvious too since rainfall is a stochastic process) because no

relation is expected between these two variables and intensity of rainfall events. But if the authors used this representation to show how the different events plot in reference to these variable this should be clearly stated.

10, 25-26. Here a two-component mixing model is mentioned but no details are given in the Materials and Method section. I'm not suggesting to report the well-known equations (a simple citation to the suggested references 5 and 6 below is enough) but some methodological/conceptual information are needed, eg: which sample(s) has been considered as pre-event for the application of the hydrograph separation technique? Why only deuterium data have been used since both 18-oxygen and deuterium data have been measured? How many samples for isotopes have been collected and which ones were used? How many events have been sampled? More importantly: why has the separation been carried out only for the 12 August event showed in Fig. 9? Or was it also performed for other events? In this case, what are the results? Are they similar so that they corroborate the proposed conceptualization? Or did they provide much different estimates? Can the author report the results of all sampled events in a Table? This would be useful. All this information should be reported and these points well addressed in the revised version of the manuscript.

Another major point related to this is the lack of uncertainty analysis of the estimated fractions of pre-event water and event water (I prefer these terms instead of old and new water) in streamflow. This is particularly critical since these estimates have been used to build some conceptualization (eg, 17% of event water corresponding to 5% of rainfall amount..but what is the uncertainty of that 17%?). And is the result about 5% of rainfall based only on the 12 August event? In that case this is not robust. The traditional method of uncertainty estimation proposed in reference 2 below is suggested.

11, 1-11. It is mentioned that the concentration of major ions is available for the 8 November event but only chloride has been selected and showed (Fig. 10). What is the reason behind this choice? Moreover, where does the estimate of 5% of the rainfall come from, that agrees surprisingly well with the estimate of the 12 August event (10,

26)? From a two-component hydrograph separation based on chloride? On isotopes? Please, explain in detail.

11, 19. The saturation amount at the 5% of the catchment area is not shown and clearly presented, yet it is one of the most interesting results in terms of process interpretation. Please, provide a sound explanation.

12, 29. I do not see such a clear threshold at 250 mm of ASI + rain. . .please, explain better, also in the results, where it derives from.

Some relevant studies that I'm aware of and that are strictly linked to this research have not been cited. I think they should incorporated in the paper, particularly in the Discussion section (except the ones referring to methods, such as 2, 5 and 6):

1. Fu C, Cheng J, Jiang H, Dong L. 2013. Threshold behavior in a fissured granitic catchment in southern China: (1) analysis of field monitoring results. Water Resources Research 49: 1–17. DOI: 10.1002/wrcr.20191

2. Genereux D. 1998. Quantifying uncertainty in tracer-based hydrograph separations. Water Resources Research 34(4): 915–919. DOI: 10.1029/98WR00010

3. Penna, D., van Meerveld, H.J., Oliviero, O., Zuecco, G., Assendelft, R.S., Dalla Fontana, G., Borga, M., 2015. Seasonal changes in runoff generation in a small forested mountain catchment. Hydrological Processes 29, 2027–2042. doi:10.1002/hyp.10347

4. Penna, D., van Meerveld, H.J., Zuecco, G., Dalla Fontana, G., Borga, M., 2016. Hydrological response of an Alpine catchment to rainfall and snowmelt events. Journal of Hydrology 537, 382–397. doi:10.1016/j.jhydrol.2016.03.040

5. Pinder, G.F., Jones, J.F., 1969. Determination of ground-water com-ponent of peak discharge from chemistry of total runoff. Water Resour. Res. 5 (2), 438–445. http://dx.doi.org/10.1029/WR005i002p00438.

6. Sklash MG, Farvolden RN. 1979. Role of groundwater in storm runoff. Journal of Hydrology 43(1–4): 45–65. DOI: 10.1016/0022-1694(79)90164-1

Minor comments and technical corrections

1, 10. The reference to individual research catchments makes the reader think that this paper focuses on the analysis of several catchments but this is not the case. I suggest to remove or reformulate.

2, 12. Here the suggested references 1 and 3 could be added.

3, 3-4. This sentence is not totally clear. Please, explain.

3, 22. Typo.

3, 27. 'certain processes': too vague. Reformulate.

3, 30. Here the suggested references 1 and 4 could be added.

4, 9. It is a bit surprising to know that the study area is a hillslope after reading the Introduction that focuses almost exclusively on processes at the catchment scale!

4, 17. 'reasonably' is too vague. Specify.

5, 26. This can be misunderstood as the uncertainty in the presented results of hydrograph separation. I think it's clearer to use the term 'instrumental precision'.

6, 15. Do the authors mean 'conceptually separate' here, ie they are considering these processes, and not physically computing the fractions of return flow and SOF in streamflow? Please, reformulate for clarity.

7, 6. Better to use 'stream' or 'streamflow' here instead of 'runoff'.

7, 23. For the sake of clarity, indicate which events/panels.

8, 9. How was the quick flow runoff coefficient computed? In section 2.4 it was not defined...unless it's, as I think, the same than 'event runoff coefficient'. In the latter

case, please be terminologically consistent.

8, 28-9, 4. This part could be condensed by pointing out at the Tables.

10, 10. Although known and intuitive, the symbols of this equation should be explained. Moreover, it should be stated that the events falling into this period are 10 (as inferred from Fig. 8).

10, 17. 'Clearly'. I think it would be more cautious to start this sentence stating the results and/or the figures that point at this.

10, 24. 'different signature': ok, but the trend is similar and should be remarked.

10, 25. Here the suggested references 5 and 6 could be added.

11, 20. Please, explain what the 'field observations' are.

11, 24. Here the suggested reference 4 could be added.

13, 7-13. This part is not very relevant to the observed results and could be skipped, in my opinion.

Tables and Figures Table 1. Remove the first column, it's not useful. Don't use abbreviations in the column name.

Fig. 1. The first 'Yes' on the top horizontal arrows should be moved more to the right close to the dashed arrow, in my opinion. And perhaps the second 'Yes' can be removed.

Fig. 2. I suggest the terms lower, mid and upper hillslope (or slope) instead. Remove the notation and the arrow pointing to the wells and put them in a legend. Why has the DEM been cut before the stream...cannot be extended to it?

Fig. 4. Replace 'overview' with 'example'. I also suggest to include a no-flow event.

Fig. 5. Please, see my comment above. Moreover, the difference between 'Soil Moisture Index' of panel b) and 'ASI' of panel c) is not clear and should be explained (or

fixed if they are the same thing).

Fig. 6. Why are there values above zero? Explain or fix.

Fig. 7. Why have only these sites been shown and not also water table at the other locations? This should be explained in the text. Additionally, 'high intensity' is too vague and should quantified, possibly using thresholds presented in Fig. 5.

Fig. 8. Add a mention to the period when these events have been selected. It would be interesting to see these results also for other events.

Fig. 9. The symbol '‰' should be put in parenthesis.

Fig. 10. Please be consistent with the use of terms such as 'discharge' (as here) or 'flow' (as in Fig. 9).

---

## Referee Comment (RC2) · Anonymous Referee #2 · 30 Jul 2016

General Comments: The content of the article is relevant to the hydrological community and meets the focus of the selected journal. It investigates functional relationships between antecedent wetness and rainfall characteristics and the streamflow response of a small agricultural catchment in Australia. In doing so the authors aim at identifying multiple co-existing runoff processes and potential threshold behavior between catchment-states and streamflow response. The dataset, comprising of hydrometric and hydrochemical parameters has potential but needs more quantitative analysis in order to address the outlined themes.

My main suggestions to improve the manuscript are the following:

I acknowledge the idea to use a decision scheme based on properties and mechanisms to structure runoff processes such as in Figure1. However, such a scheme is designed

to result in one dominant runoff process and not in multiple ones such as outlined in the title. (We all know, that processes co-exist in different degrees of intensity). Original versions of such decision schemes are designed for the point or plot scale. While I acknowledge the authors idea to extend it with the concept of connectivity and timescales, I think that this causes a mismatch of scales. At least the authors need to define very clearly what spatial and temporal scale they are considering (and stick to their definition) and what the landscape units are, between which they consider connectivity. I suggest to come up with a separate Figure for connectivity

The method section needs to provide more quantitative information. (e.g. soil profile, total number of Q, GW, NS, monitoring sites, procedure of manual sampling, delineation of the saturated area, lab-analysis devices used. (Using two different devices for analyzing isotopes can cause considerable difficulties in comparing or pooling data). The result section is descriptive and lacks statistical/data analysis to quantify the authors' statements and derive generally applicable results. Some results are based on one or a few selected events only, which is not representative to draw conclusions. I suggest to exploit the entire dataset the authors have at hand and calculate statistics over all events. Some parts of the result section are the authors' interpretation and better fit in to the discussion section. Terms are either not defined in the text (e.g., in the method section) or not used consistently and the term "threshold" is used in circumstances where "exponential relation" is more appropriate.

The conclusions are drawn from one or two individual rainfall events and not logically derived from or supported by the results of this study. I would encourage the authors to refine and strengthen their analysis based on their dataset. I think it is good to discuss the findings in the light of Fig1. but as it is originally developed for point- or plot scale assessments it misses out the spatial (and temporal) heterogeneity across a catchment. – a fundamental aspect when analyzing thresholds and connectivity – and something that I think the authors try to address. For my detailed comments please see the provided pdf documents and summary of comments.

In general, I think this manuscript has potential to be an interesting contribution to the hydrological society why I encourage the authors to work on a revised version.

Please also note the supplement to this comment:
http://www.hydrol-earth-syst-sci-discuss.net/hess-2016-288/hess-2016-288-RC2-supplement.pdf

———————————————————

[Figure]

**Supplement:**

[Figure]

**Multiple runoff processes and multiple thresholds control agricultural runoff generation**

Shabnam Saffarpour[1], Andrew W. Western[1], Russell Adams[1], and Jeffrey J. McDonnell[2,3]

[1]Department of Infrastructure Engineering, The University of Melbourne, Parkville, 3010, Australia

[2]Natl Hydrol Res Ctr, Global Inst Water Secur, University of Saskatchewan, Saskatoon, SK S7N 3H5, Canada }

[3]School of Geosciences, University of Aberdeen, Aberdeen UK}

*Correspondence to*: Andrew Western (a.western@unimelb.edu.au)

**Abstract.** Hydrologic connectivity associated with runoff processes is a critical concept for understanding catchment hydrologic response at the event timescale. However, to date, most attention has focused on single runoff response types in individual research catchments. Here we examine how runoff response and the catchment threshold response to rainfall affect a suite of runoff generation mechanisms in a small agricultural catchment. A 1.37 ha hillslope in the Lang Lang River catchment, Victoria, Australia was instrumented and hourly data of rainfall, runoff, shallow groundwater level and isotope water samples were collected. We analyse 60 rainfall events that produced 38 runoff events over two runoff seasons. Our results show that the catchment hydrologic response was typically controlled by the antecedent soil moisture condition and rainfall characteristics. There was a strong seasonal effect in the antecedent moisture conditions that led to marked seasonal scale changes in runoff response. Analysis of shallow well data revealed that streamflows early in the runoff season were dominated primarily by saturation excess overland flow from the riparian area. As the runoff season progressed, the catchment soil water storage increased and the hillslope connected to the riparian area. The hillslope transferred a significant amount of water to the riparian zone during and following events. Then, during a particularly wet period, this connectivity to the riparian zone, and ultimately to the stream, persisted between events for a period of one month. These findings are supported by isotope results which showed the dominance of pre-event water, and increased contributions of new water early (rising limb and peak) in the event hydrograph for wetter conditions. We conclude that event runoff at this site is a combination of subsurface event flow and saturation excess overland flow. However, during high intensity rainfall events, flashy hillslope flow was observed even though the soil moisture threshold for activation of subsurface flow was not exceeded. We hypothesize that this was due to the activation of infiltration excess overland flow and/or fast lateral flow through preferential pathways on the hillslope and saturation overland flow from the riparian zone.

**Summary of comments: saffarpoupr_et-al_2016_hess_288_ed.pdf**

**Page:1**

**Number: 1  Author: Anonymous  Subject: Strikeout  Date: 2016-07-29 16:11:05**

**Number: 2  Author: Anonymous  Subject: Insert Text  Date: 2016-07-29 16:11:27**
increased

**Number: 3  Author: Anonymous  Subject: Note  Date: 2016-07-29 16:15:23**
Maybe this become clear in the text but how did you distinguish between these runoff source areas/mechanisms?

[Figure]

**1 Introduction**
[Figure]

Thresholds have been an integral part of overland flow theory since the early infiltration excess work of Horton (1933) and saturation excess studies of Dunne and Black (1970a, b). Thresholds in runoff response have also been observed in subsurface stormflow dominated systems (Hewlett and Hibbert, 1967). More recent work has shown these to be a function of

5   catchment wetness status for saturation excess overland flow (Western and Grayson, 1998;Western et al., 2005) and subsurface stormflow (Freer et al., 2002;Tromp-van Meerveld et al., 2007). Hydrological connectivity is now a useful generic concept that links reservoirs to their downstream conduits (Tetzlaff et al., 2010) and a connectivity framework can provide a powerful explanator of catchment flow and transport response (Ali et al., 2013;Detty and McGuire, 2010;Lehmann et al., 2007;McGuire and McDonnell, 2010;Western et al., 1998, 2001). Connectivity and thresholds are intimately related;

10  typically a threshold in some catchment state controls the transition between connected and disconnected states; for example, the observation that subsurface flow becomes connected above some soil water storage and rainfall threshold (Detty and McGuire, 2010;Tromp-van Meerveld and McDonnell, 2006a).

Despite significant progress in understanding the non-linear behaviour of catchments related to soil moisture thresholds, watertable dynamics, connectivity of surface and subsurface pathways and their influence on runoff generation mechanisms,

15  it is not explicitly understood how the non-linear properties of catchments (connectivity and thresholds) work to convert rainfall to runoff nor how such behaviours vary between different types of catchments. It has been argued that interactions between the various processes and thresholds leads to complex non-linear rainfall-runoff behaviour in catchments (Hopp and McDonnell, 2009;Kirchner, 2006;Tetzlaff et al., 2010;Uchida et al., 2005) including: thresholds for initiation of hillslope-to-stream connectivity (Ali et al., 2013;Detty and McGuire, 2010;Fujimoto et al., 2008;Lehmann et al., 2007;McGuire and

20  McDonnell, 2010;Tromp van Meerveld and McDonnell, 2005;Tromp-van Meerveld and McDonnell, 2006a); variable flow hysteresis patterns depending on rainfall amount and antecedent soil moisture conditions (Bowes et al., 2009;Holz, 2010;McGuire and McDonnell, 2010); and flushing of nutrients in agricultural catchments (Bracken and Croke, 2007;Ocampo et al., 2006;Tockner et al., 1999;Withers and Lord, 2002).

While the concept of connectivity has been useful in many of these studies, the studies have concentrated on individual

25  mechanisms. It is less clear how catchments behave when subject to a mixture of runoff mechanisms including infiltration excess and saturation excess overland flow, and subsurface stormflow. Few studies have tried to tease apart the influence of multiple processes in catchments where infiltration excess, saturation excess and subsurface stormflow are all important. Here we do that for an agricultural catchment in south-eastern Australia and show the shifting importance of different processes over time associated with changes in catchment wetness and rainfall intensity. Prior to this, we consider the role

30  of multiple thresholds in catchment states and fluxes as well as the role of thresholds in certain timescales in controlling different modes of hydrologic connectivity and associated rainfall-runoff response.

Figure 1 summarizes the status quo in terms of the combined effects of thresholds and connectivity. It shows the importance of various timescales, fluxes and states, and how these relate to variation in rainfall-runoff processes over time (and between

**Number: 1  Author: Anonymous  Subject: Note  Date: 2016-07-29 17:52:48**

General Comment on Introduction:

The introduction is a lot about Fig., and less describes what others have found about thresholds and runoff mechanisms (only mentioned in a few sentences and (good list of references).
I also think, that not all studies only concluded that one runoff mechanism is dominating in their catchment. So some pervious work has also concluded, that there are multiple processes happening at the same time or vary seasonally

I also think, that it needs to be clearly stated that Fig. 1 (at least I think) is describing dominant processes at the point or plot-scale.
I like the idea of the authors to go beyond that and think about aspects that need to be considered in terms of connectivity between this points but they are not in the diagram (Fig.1). (see my comments for more detail). If talking about connectivity, this term needs to be diefined and also between what (e.g. hillslope stream).

Time scale: I also like the idea to pay more attention to time-scales but it needs to be defined what the authors think of. I guess, in some circumstances they are not talking about time-scale but rather duration.

The authors could be more clear what this work contributed to and say more about how their work is new!

**Number: 2  Author: Anonymous  Subject: Note  Date: 2016-07-29 16:21:53**

There also exists the concept of a continuous nature of connectivity but I agree, that most studies chose the threshold-type of concept.
You could say, that you chose the latter concept which implies ...

**Number: 3  Author: Anonymous  Subject: Note  Date: 2016-07-06 13:17:59**

maybe add Penna et al., 2015

**Number: 4  Author: Anonymous  Subject: Note  Date: 2016-07-29 16:27:59**

not 100% clear. You mean multiple threshold for different runoff generation mechanisms?

**Number: 5  Author: Anonymous  Subject: Note  Date: 2016-07-29 16:56:42**

state here that Fig1 is about dominant runoff mechanisms or that you consider them to co-exist in parallel.

**Number: 6  Author: Anonymous  Subject: Insert Text  Date: 2016-07-29 16:41:52**

on dominant runoff processes.

(see my comment at Fig1)

[Figure]

catchments).
[Figure]
 Of course, questions of instantaneous flux and also of the relative timescales of various processes are often important in determining the existance of connectivity (Tromp van Meerveld and McDonnell, 2005;Western et al., 2005). It would be attractive to think of the problem of runoff response purely in terms of timescales of competing processes following Oldham et al. (2013); however, both flux and time thresholds are important. This arises because there is finite

5  capacity for flow in various parts of the catchment system.

Figure 1 is divided into three areas, the lefthand area (climate and landscape characteristics) provides a series of catchment thresholds that determine runoff processes and connectivity, depending on whether they are exceeded or not. The middle area points to the outcome in terms of runoff generation processes and the righthand area provides example catchments from the literature that exhibit those processes. Some of the thresholds are posed in terms of flux rate compared with a flow

10  capacity (e.g. box 2) and some in terms of a state threshold (box 5). The flux and state thresholds are considered in the context of a process timescale. This is because the threshold needs to be exceeded for a sufficient time for the action of the process to lead to a significant impact.

Consider box 1. Rainfall rates vary across a very wide range to timescales. If the rainfall (or throughfall) intensity exceeds the infiltration threshold for only a very short time, the water that ponds on the surface will continue to infiltrate as it flows

15  toward the stream (runon infiltration) when the intensity reduces and very little or no runoff will result (surface connectivity didn't become established). However if average intensities exceed the infiltration capacity for long enough for ponded water to flow to the catchment outlet, the hillslope will connect to the catchment outlet via surface pathways and produce runoff. The remaining boxes consider thresholds in the context of subsurface flow times. Box 3 considers situations where subsurface saturation exists, allowing lateral subsurface flow paths to be activated. If any of deep infiltration through the

20  impeding layer (Jackson et al., 2014), unfilled bedrock storage (Janzen and McDonnell, 2015) or evaptranspiration cause the saturation and/or lateral flow to disipate before water can move a significant distance downstream, the water will not be effectively redistributed downslope and subsurface connection wont be established (this is Grayson et al.'s (1997) local control). If the saturation persists for long enough lateral subsurface flow will connect to the stream. At the other extreme (box 7), if lateral flow is persistently exceeding the hillslope subsurface flow capacity, surface saturation will exist leading to

25  saturation excess runoff.

Figure 1 goes about here

While Figure 1 suggests catchments are dominated by certain processes it needs to be recognised that many catchment conditions vary over time. For example summer rainfall is often more intense than winter rainfall. Soil water conditions vary seasonally in response to both rainfall and potential evapotranspiration, sometimes leading to switching between

30  characteristic spatial patterns and prevailing responses to rainfall (Grayson et al., 1997;Western et al., 1999). Topographic, soil and vegetation conditions can also vary across a catchment. This all suggests that catchments could exhibit a mix of processes.

Here we use the above framework to understand the behaviour of a catchment that does indeed exhibit a mix of runoff processes. We examine how soil water storage and shallow water table response influence subsurface connectivity and

**Number: 1  Author: Anonymous  Subject: Insert Text  Date: 2016-07-29 16:55:28**

with different physiographic characteristics

**Number: 2  Author: Anonymous  Subject: Note  Date: 2016-07-29 16:57:24**

2006?

**Number: 3  Author: Anonymous  Subject: Note  Date: 2016-07-29 17:00:44**

Do you mean the difference between the dominant runoff process concept and a concept of co-existing of these processes with different degree of importance. Can you rewrite?

**Number: 4  Author: Anonymous  Subject: Strikeout  Date: 2016-07-29 17:02:03**

**Number: 5  Author: Anonymous  Subject: Note  Date: 2016-07-29 17:01:04**

parts?

**Number: 6  Author: Anonymous  Subject: Note  Date: 2016-07-29 17:04:26**

I am not sure if they tell directly about connectivity? Connectivity between what? Hillslope stream, neighbouring sites?

maybe better "lateral flwo"?

**Number: 7  Author: Anonymous  Subject: Insert Text  Date: 2016-07-29 17:02:50**

that depend on climate and landscape characteristics and ...

**Number: 8  Author: Anonymous  Subject: Insert Text  Date: 2016-07-29 17:04:53**

dominant

**Number: 9  Author: Anonymous  Subject: Note  Date: 2016-07-29 17:06:52**

I like this idea but I think you need to introduce the reader more to these time scales.

**Number: 10  Author: Anonymous  Subject: Note  Date: 2016-07-29 17:23:24**

I think these are not time scales but time durations or simple ".. over an event"

**Number: 11  Author: Anonymous  Subject: Note  Date: 2016-07-29 17:24:41**

I understand what you want to say but I think now you need to decide if you consider the diagram in Fig.1 to be on the point- scale (which I think the diagram is). If you want to consider processes such as run-on, than you need to redraw your diagram. Suggestion: try to come up with a similar diagram that considers connectivity between the hillslope and the stream and what processes can occur. For connectivity a few other criteria need to be fulfilled. Things you describe here fit well in this other context.

**Number: 12  Author: Anonymous  Subject: Note  Date: 2016-07-29 17:25:04**

please define

**Number: 13  Author: Anonymous  Subject: Note  Date: 2016-07-29 17:26:47**

On an event-time-scale evapotranspiration might be a minor component. It more effects e.g. antecedent conditions.

**Number: 14  Author: Anonymous  Subject: Note  Date: 2016-07-29 17:28:56**

use different word

**Number: 15  Author: Anonymous  Subject: Note  Date: 2016-07-29 17:31:07**

it is not only about persistence (which implies time), it is about a continuous connection (hydraulically or hydrologically from point A to point B.

[Figure]

catchments). Of course, questions of instantaneous flux and also of the relative timescales of various processes are often important in determining the existance of connectivity (Tromp van Meerveld and McDonnell, 2005;Western et al., 2005). It would be attractive to think of the problem of runoff response purely in terms of timescales of competing processes following Oldham et al. (2013); however, both flux and time thresholds are important. This arises because there is finite

5   capacity for flow in various parts of the catchment system.

Figure 1 is divided into three areas, the lefthand area (climate and landscape characteristics) provides a series of catchment thresholds that determine runoff processes and connectivity, depending on whether they are exceeded or not.  The middle area points to the outcome in terms of runoff generation processes and the righthand area provides example catchments from the literature that exhibit those processes. Some of the thresholds are posed in terms of flux rate compared with a flow

10   capacity (e.g. box 2) and some in terms of a state threshold (box 5).  The flux and state thresholds are considered in the context of a process timescale.  This is because the threshold needs to be exceeded for a sufficient time for the action of the process to lead to a significant impact.

Consider box 1. Rainfall rates vary across a very wide range to timescales.  If the rainfall (or throughfall) intensity exceeds the infiltration threshold for only a very short time, the water that ponds on the surface will continue to infiltrate as it flows

15   toward the stream (runon infiltration) when the intensity reduces and very little or no runoff will result (surface connectivity didn't become established).  However if average intensities exceed the infiltration capacity for long enough for ponded water to flow to the catchment outlet, the hillslope will connect to the catchment outlet via surface pathways and produce runoff. The remaining boxes consider thresholds in the context of subsurface flow times.  Box 3 considers situations where subsurface saturation exists, allowing lateral subsurface flow paths to be activated.  If any of deep infiltration through the

20   impeding layer (Jackson et al., 2014), unfilled bedrock storage (Janzen and McDonnell, 2015) or evaptranspiration cause the saturation and/or lateral flow to dissipate before water can move a significant distance downstream, the water will not be effectively redistributed downslope and subsurface connection wont be established (this is Grayson et al.'s (1997) local control).  If the saturation persists for long enough lateral subsurface flow will connect to the stream.  At the other extreme (box 7), if lateral flow is persistently exceeding the hillslope subsurface flow capacity, surface saturation will exist leading to

25   saturation excess runoff.

Figure 1 goes about here

While Figure 1 suggests catchments are dominated by certain processes it needs to be recognised that many catchment conditions vary over time. For example summer rainfall is often more intense than winter rainfall. Soil water conditions vary seasonally in response to both rainfall and potential evapotranspiration, sometimes leading to switching between

30   characteristic spatial patterns and prevailing responses to rainfall (Grayson et al., 1997;Western et al., 1999). Topographic, soil and vegetation conditions can also vary across a catchment. This all suggests that catchments could exhibit a mix of processes.

Here we use the above framework to understand the behaviour of a catchment that does indeed exhibit a mix of runoff processes. We examine how soil water storage and shallow water table response influence subsurface connectivity and

Please define at the beginning A and B.

**Number: 16  Author: Anonymous  Subject: Note  Date: 2016-07-29 17:33:29**

On a singe hillslope there can be patches of different processes. The question is: Do they for a continous path between A and B?

**Number: 17  Author: Anonymous  Subject: Insert Text  Date: 2016-07-29 17:34:58**

in our study catchment ...

**Number: 18  Author: Anonymous  Subject: Insert Text  Date: 2016-07-29 17:34:17**

and space

**Number: 19  Author: Anonymous  Subject: Note  Date: 2016-07-29 17:35:31**

patters of what (soil moisture?)

**Number: 20  Author: Anonymous  Subject: Insert Text  Date: 2016-07-29 17:36:53**

in Australia

rainfall-runoff response at seasonal and event based time scales. We also examine the relative role of saturation excess and subsurface flow in generating peak runoff rates and event volumes. Finally we examine circumstances under which rainfall intensity plays a role in runoff generation responses. The field site is a small agricultural catchment in the Lang Lang River catchment, Victoria, Australia, which we examine through the lens of hydrometric and isotope and geochmistry

5    measurements. These results are used to propose a conceptual model of the processes and pathways that contribute event runoff as catchment wetness and rainfall intensity vary.

**2 Methods**

**2.1 Study location**

The study site is a 1.37 ha hillslope (RBF) located on a dairy farm at Poowong East, in the Lang Lang River Catchment,

10   Victoria, Australia, 130 km south-east of Melbourne (Figure 2). A general description of the study catchment can be found in Adams et al. (2014). The study period was between September 2009 and December 2011. Elevation ranges from 160 to 210 mAHD and the slope varies from 2% to 50%. Based on field observations and the hydrologic behaviour, the hillslope was divided into four different zones: 1) the riparian area located on the relatively flat convergent lower part of the hillslope (outlined in black on Figure 2) included sites 1, 2, 32 and 3; 2) the lower slope (low slope) area; 3) the mid slope area

15   with sites 4, 5, 6 and 7 ; and 4) the upper slope (upslope) area with sites 10, 11 and 15 (Figure 2).

Figure 2 goes about here

The study area has a humid climate and rainfall is reasonably uniformly distributed across the year with an annual mean (1961-1990) of 1100 mm (Bureau of Meteorology, 2009). Annual areal potential evapotranspiration is 1042 mm (Bureau of Meteorology, 2005). The catchment geology comprises of sandstones and mudstones of the Cretaceous Strezlecki Group

20   (VRO, 2013). Outcrops on the lower stream banks of the catchment (just downstream of the monitored hillslope) show weathered sandstone and mudstone bedrock. Hand augering revealed a soil depth of <= 1.5-1.6 m and the lower parts of the profile included mottled clay and weathered bedrock particles. The soils are acidic and mesotrophic brown dermosols (Isbell, 2002). Soil profile depth decreases moving downslope. These soils typically have a moderate hydraulic conductivity surface horizon (0-40 cm, $K_s \approx 5*10^{-6}$ m s$^{-1}$, about 20 mm hr$^{-1}$). The dominant land use is grazing by dairy cows.

25   ### 2.2 Site instrumentation and hydrometric data monitoring

Rainfall data were recorded using a 0.2 mm tipping-bucket raingauge at an automatic weather station which was installed in 2010 at the top of RBF. A rainfall sampler (Kennedy et al., 1979) collected up to ten sequential rainfall samples, each being equivalent to 6.6 mm of rainfall. The sampler was initially installed close to the AWS, however, due to instances of damage by animals, it was relocated near to the flume in August 2010 until the end of the study period. A trapezoidal flume was

30   installed at the riparian zone outlet and an Odyssey (Dataflow Systems inc. Christchurch, NZ) pressure transducer (PT)

**Page:4**

**Number: 1  Author: Anonymous  Subject: Note  Date: 2016-07-29 17:38:22**

do you mean runoff generation response or dominant runoff mechanism?

**Number: 2  Author: Anonymous  Subject: Note  Date: 2016-07-29 19:06:10**

In general: the individual sections could improve from the order in which facts are stated.
Please provide more quantitative information (e.g. soil profile), total number of Q, GW, NS, monitoring sites
be more specific about which instruments have been used to analyze water samples.
Where did you manually take samples, how often nad when>
How did you determine the saturated area?

**Number: 3  Author: Anonymous  Subject: Note  Date: 2016-07-29 18:04:26**

Is this the catchment name? If so please write out the full name once and put the abbr. behind.

**Number: 4  Author: Anonymous  Subject: Note  Date: 2016-07-29 17:55:19**

groundwater?

**Number: 5  Author: Anonymous  Subject: Note  Date: 2016-07-29 17:55:53**

groundwater, streamflow?

**Number: 6  Author: Anonymous  Subject: Note  Date: 2016-07-29 17:57:12**

please describe the criteria you used to define these zones (range of slope values, TWI, etc.)

**Number: 7  Author: Anonymous  Subject: Note  Date: 2016-07-29 17:57:52**

(1961 - 1990 ?)

**Number: 8  Author: Anonymous  Subject: Strikeout  Date: 2016-07-29 17:58:18**

**Number: 9  Author: Anonymous  Subject: Note  Date: 2016-07-29 17:59:29**

... at all groundwater wells ?

**Number: 10  Author: Anonymous  Subject: Note  Date: 2016-07-29 18:00:14**

please be clear if this is the average soil depth, the range ...

**Number: 11  Author: Anonymous  Subject: Note  Date: 2016-07-29 18:01:29**

What about the other horizons. Is ther an impeding soil layer at what depth?

**Number: 12  Author: Anonymous  Subject: Note  Date: 2016-07-29 18:10:40**

section needs better ordering. E,g, if you start with rainfall, than continue with all other meteo parameters. Consider to start with the
parameters that are most relevant.

**Number: 13  Author: Anonymous  Subject: Strikeout  Date: 2016-07-29 18:03:40**

**Number: 14  Author: Anonymous  Subject: Note  Date: 2016-07-29 18:05:00**

per event, day, season?

**Number: 15  Author: Anonymous  Subject: Note  Date: 2016-07-29 18:05:34**

please provide full name!

[Figure]

rainfall-runoff response at seasonal and event based time scales. We also examine the relative role of saturation excess and subsurface flow in generating peak runoff rates and event volumes. Finally we examine circumstances under which rainfall intensity plays a role in runoff generation responses. The field site is a small agricultural catchment in the Lang Lang River catchment, Victoria, Australia, which we examine through the lens of hydrometric and isotope and geochmistry

5  measurements. These results are used to propose a conceptual model of the processes and pathways that contribute event runoff as catchment wetness and rainfall intensity vary.

**2 Methods**

**2.1 Study location**

The study site is a 1.37 ha hillslope (RBF) located on a dairy farm at Poowong East, in the Lang Lang River Catchment,

10  Victoria, Australia, 130 km south-east of Melbourne (Figure 2). A general description of the study catchment can be found in Adams et al. (2014). The study period was between September 2009 and December 2011. Elevation ranges from 160 to 210 mAHD and the slope varies from 2% to 50%. Based on field observations and the hydrologic behaviour, the hillslope was divided into four different zones: 1) the riparian area located on the relatively flat convergent lower part of the hillslope (outlined in black on Figure 2) included sites 16, 1, 2, 32 and 3; 2) the lower slope (low slope) area; 3) the mid slope area

15  with sites 4, 5, 6 and 7 ; and 4) the upper slope (upslope) area with sites 10, 11 and 15 (Figure 2).

Figure 2 goes about here

The study area has a humid climate and rainfall is reasonably uniformly distributed across the year with an annual mean (1961-1990) of 1100 mm (Bureau of Meteorology, 2009). Annual areal potential evapotranspiration is 1042 mm (Bureau of Meteorology, 2005). The catchment geology comprises of sandstones and mudstones of the Cretaceous Strezlecki Group

20  (VRO, 2013). Outcrops on the lower stream banks of the catchment (just downstream of the monitored hillsope) show weathered sandstone and mudstone bedrock. Hand augering revealed a soil depth of $<= 1.5$-$1.6$ m, and the lower parts of the profile included mottled clay and weathered bedrock particles. The soils are acidic and mesotrophic brown dermosols (Isbell, 2002). Soil profile depth decreases moving downslope. These soils typically have a moderate hydraulic conductivity surface horizon (0-40 cm, $K_s \approx 5*10^{-6}$ m s$^{-1}$, about 20 mm hr$^{-1}$). The dominant land use is grazing by dairy cows.

25  ### 2.2 Site instrumentation and hydrometric data monitoring

Rainfall data were recorded using a  tipping-bucket raingauge at an automatic weather station which was installed in 2010 at the top of RBF. A rainfall sampler (Kennedy et al., 1979) collected up to ten sequential rainfall samples, each being equivalent to 6.6 mm of rainfall. The sampler was initially installed close to the AWS, however, due to instances of damage by animals, it was relocated near to the flume in August 2010 until the end of the study period. A trapezoidal flume was

30  installed at the riparian zone outlet and an Odyssey (Dataflow Systems inc. Christchurch, NZ) pressure transducer (PT)

**Number: 16  Author: Anonymous  Subject: Note  Date: 2016-07-29 18:05:57**

give site number

recorded water levels every 10 minutes, which were used to compute instantaneous flow rates. After August 2011, the PT was replaced with an ISCO (Teledyne ISCO , Lincoln,NE,USA), model 730 bubbler.

An auto sampler (Teledyne ISCO 6712) was installed at the outlet of RBF and was triggered based on the rising stage and programmed to collect up to 24 samples at hourly intervals. Samples were collected from the auto sampler within 48 hours.

5  To reduce the laboratory analysis workload, the flow hydrograph was graphed in the field prior to event sample collection. All samples during the rising limb and the peak were collected and samples were typically selected at an interval of 4 hours during the falling limb. Routine grab sampling was undertaken at weekly intervals during the main runoff season when water was flowing through the RBF flume. This was supplemented by additional grab sampling during visits to collect event samples from the auto sampler.

10  Weather variables (temperature, humidity, wind, rainfall, global radiation) were measured by the automatic weather station. Areal potential evapotranspiration (APET) was also computed using the Morton (1983) wet environment method on a daily basis. APET was strongly seasonal resulting in strongly seasonal soil moisture contents and intermittent streamflow at RBF. Hourly volumetric water content (VWC) was measured at 0-30 cm and 30-60 cm depths using vertically installed 30 cm long Campbell Scientific CS625 probes (Campbell Scientific, 2006) situated close to the AWS. The soil moisture sensor data

15  were corrected for soil temperature which was also measured close to the AWS.

To measure the nature of hydrologic connectivity, runoff mechanisms and flow pathways, shallow (1.5-1.6 m) groundwater wells were installed across the RBF hillslope using 40 mm PVC pipes and backfilled with sand, bentonite, the topsoil and grass. Figure 2 shows these sites of which 1, 2, 3, 16 and 32 were in the riparian zone; 4, 5, 6 and 7 were on the mid slope; and 10, 11 and 15 were on the upper slope. Sites 4, 5 and 6 were equipped with water level loggers from July 2010, and sites

20  3, 7, 16 and 32 were logged from winter 2011. Water levels were logged using Odyssey PT loggers.

**2.3 Water sample analysis**

Sub-samples were taken of stream water from both manual and auto sampler samples, and from all full rainfall sample bottles; these were collected in glass bottles for isotope analysis. Bottles were completely filled. The samples were refrigerated (+ 4°C) until analysis for $\delta^{18}O$ and $\delta^{2}H$, either by the Monash University Earth Sciences laboratory or by

25  Professor Russell Drysdale's isotope laboratory at the University of Melbourne, where a Picarro L2120i cavity ring-down isotope analyser was used to determine isotope ratios. The uncertainty in results was $\delta^{18}O$=0.1‰ and $\delta^{2}H$ =0.4‰. Sub-samples were also taken from each water sample for selected major ion (Na$^{+}$, K$^{+}$, Ca$^{2+}$, Mg$^{2+}$ and Cl$^{-}$) analyses, which were analysed in the NATA-certified, analytical chemistry laboratory of the Water Studies Centre at Monash University using standard methods.

**Number: 1  Author: Anonymous  Subject: Note  Date: 2016-07-29 18:06:58**

with what equation (give a citation)

**Number: 2  Author: Anonymous  Subject: Note  Date: 2016-07-29 18:19:25**

sampling what (I guess streamflow)
So you had one site with water samples?

**Number: 3  Author: Anonymous  Subject: Note  Date: 2016-07-29 18:08:53**

how did you know the hydrograph before the event? Unclear please rewrite.

**Number: 4  Author: Anonymous  Subject: Note  Date: 2016-07-29 18:09:22**

interval for the sampling?

**Number: 5  Author: Anonymous  Subject: Note  Date: 2016-07-29 18:13:23**

how did you measure the lower range when the probe is 30 cm? Pits? If so, describe their installation procedure

**Number: 6  Author: Anonymous  Subject: Note  Date: 2016-07-29 18:11:38**

soil water content?

**Number: 7  Author: Anonymous  Subject: Note  Date: 2016-07-29 18:17:55**

please state clearly how many Q, GW, SM sites you had.

**Number: 8  Author: Anonymous  Subject: Note  Date: 2016-07-29 18:15:36**

... until 20xx

**Number: 9  Author: Anonymous  Subject: Note  Date: 2016-07-29 18:17:18**

I think Odyssey are not pressure transducers!

**Number: 10  Author: Anonymous  Subject: Note  Date: 2016-07-29 18:16:00**

until 20xx (state the duration you were analysing!

**Number: 11  Author: Anonymous  Subject: Note  Date: 2016-07-29 18:20:33**

You need t say more about the manual sampling. Where, how often, at what times in streamflow, in gw?

**Number: 12  Author: Anonymous  Subject: Note  Date: 2016-07-29 18:23:29**

Please state exactly which instruments you used (not the Professor).
If you used different machines but pooled the data for the analysis you have to somehow say something about potential differences (best you state which samples were analyezed where or at least sampels from 2010  ... and 2011 ...

**Number: 13  Author: Anonymous  Subject: Note  Date: 2016-07-29 18:23:58**

How is this determined?

**Number: 14  Author: Anonymous  Subject: Note  Date: 2016-07-29 18:24:50**

which ones and which machines

[Figure]

**2.4 Rainfall and runoff events**

In order to analyse ev
behaviour, it was necessary to identify rainfall and runoff events. Based on an examination of the time series of hourly rainfall in the catchment, rain events were defined as having >= 5 mm total rainfall, and peak hourly rainfall intensity, $I_{peak}$ >= 1.5 mm hr$^{-1}$. Distinct events were separated by > 12 hours without rainfall.

5   The runoff hydrograph was also divided into events. Runoff events began when the hydrograph started to rise from its initial low flow value or moved above a threshold of 0.05 mm hr$^{-1}$ following the commencement of a rainfall event. Events ended either when: 1) the flow returned to its initial value; 2) a new rainfall event started; or, 3) 96 hours after the end of the rainfall event in unusually wet situations where elevated flow continued. For each event, a number of characteristics were determined as shown in Tables 1 and 2.

10   The antecedent soil moisture condition (ASI) was represented using the soil water storage in the top 60 cm of the profile at the AWS at the start of each rainfall event. Saturated area extent was estimated based on manual measurements of the upstream extent of the saturated area (see Figure 2 for the maximum boundary location at site 3) in the field between events, combined with GIS information. The saturated area is topographically constrained in the lateral direction. The proportion of saturated area was estimated using these data and then used to estimate saturation excess runoff generation for the different

15   events. In this study we separate return flow and flow resulting from direct precipitation on the saturated area and use Saturation excess Overland Flow (SOF) to refer to the latter. The event runoff depth (mm) and event runoff coefficient (RC %) were calculated by separating the event hydrograph using the method of (Hewlett and Hibbert, 1967), which has been widely applied (Buttle et al., 2004;Fujimoto et al., 2008;McGuire and McDonnell, 2010). The method assumes that baseflow increases at the rate of 0.55 l s$^{-1}$km$^2$h$^{-1}$ (0.002 mm hr$^{-1}$) from the start of the rising limb.

20   **3 Results**

The following results first provide an overview of the seasonal behaviour and rainfall-runoff events. They then examine whether thresholds in the antecedent conditions and/or event rainfalls exist. Next, links between the hillslope condition and the event runoff are examined using the piezometer and soil moisture data. After that, the recession behaviour of events is examined and linked to hillslope wetness conditions. Finally isotope and major ion data are presented for selected events.

25   **3.1 Overview of runoff behaviour and rainfall-runoff event characteristics**

Figure 3 shows time series of weekly rainfall, APET, soil water storage and runoff. The rainfall, although variable from week-to-week, exhibited little seasonality, while there was strong seasonality in PET. This drove a strong seasonality in soil water storage. An examination of the weekly runoff data shows that there was generally no flow from about October to May due to the seasonal nature of this catchment; however, an exception was that persistent low flow occurred from 26 November

30   2011 to the end of the event on 10 December 2011. During this period ASI was often relatively low but there was frequent and substantial rainfall (>200mm in 30 days). While a strong link between runoff and soil water storage is evident in Figure

**Number: 1   Author: Anonymous   Subject: Note   Date: 2016-07-29 18:41:32**

state the duration of the study period

**Number: 2   Author: Anonymous   Subject: Note   Date: 2016-07-29 19:25:40**

I think stating ASI needs also to state a duration over which ASI is calculated. If it is the VWC at the start of an event, than it is not typically called ASI. Please define!

**Number: 3   Author: Anonymous   Subject: Note   Date: 2016-07-29 18:33:34**

not clear from Fig. 2. Also not clear, if you did this during each events. Was it based on mapping?

Was it satturated area or the extent of the channel?

**Number: 4   Author: Anonymous   Subject: Note   Date: 2016-07-29 18:34:30**

by multiplying the area with ...

**Number: 5   Author: Anonymous   Subject: Note   Date: 2016-07-29 18:35:23**

How can you do that unless you are mapping the catchment. Please describe very clearly.

**Number: 6   Author: Anonymous   Subject: Note   Date: 2016-07-29 18:36:14**

I like the idear of giving definition. Please can you do that in the introduction and for all processes you are referring to!

**Number: 7   Author: Anonymous   Subject: Note   Date: 2016-07-29 18:37:54**

is this approach also applicable in your catchment?

**Number: 8   Author: Anonymous   Subject: Note   Date: 2016-07-30 13:49:39**

Summary of general comments:

Result section is very descriptive and lacks statistical/data analysis. Some results are based on one or a few  selected events only. I suggest to exploit the nice dataset the authors have at ahdn and calcualted statistics over many events to derive generally applicable results.

Results (e.g. thesholds) are "read" form graphs and not calculated from the dataset.

Result section contains parts, that are better suited in the discussion section because they are subject to the authors interpretation and not based on data only.

Terms are either not defined in the text (e.g., in the method section) or not used consistently.

The term "theshold" is used in circumstances where an exponential relation is more appropriate.

As the different runoff mechanisms are prominent in the title, they should be clearer addressed in the results

**Number: 9   Author: Anonymous   Subject: Strikeout   Date: 2016-07-29 18:42:46**

**Number: 10   Author: Anonymous   Subject: Note   Date: 2016-07-29 18:43:45**

define ASI the first time you use it!

**Number: 11   Author: Anonymous   Subject: Note   Date: 2016-07-29 18:44:43**

qunatify?

[Figure]

3, there are exceptions. For example in February 2011, there was a runoff response despite the catchment being near to the lowest soil water storage for the study period.

Figure 3 goes about here

Moving to the event timescale, Table 1 summarises 38 rainfall-runoff events and Table 2 shows a summary of 22 rainfall
5  events that did not produce a runoff response. A further 16 rainfall events occurred over the study period which are not included in the analysis due to missing runoff data. For the 38 runoff events, total event rainfall varied from 7 to 72 mm, $I_{peak}$ ranged from 2 to 31 mm hr$^{-1}$, ASI ranged from 130 to 286 mm and total event runoff varied between 0.23 and 41 mm. For the no-flow events (Table 2), total rainfall varied from 5 to 28 mm, $I_{peak}$ ranged from 2 to 10 mm hr$^{-1}$and ASI ranged from 146 to 238 mm. Figure 4 shows rainfall-runoff responses for selected events at RBF. These graphs are ordered from lowest
10  (27/11/2010) to highest ASI (7/6/2011) for the selected events. Most of the events presented in Figure 4 had zero or very low initial flow.

Table 1 goes about here

Table 2 goes about here

Figure 4 goes about here

15  In Figure 4 most events showed rapid response to rainfall, except for the event on 12/11/2010 which did not produce any runoff and the event on 7/6/2011. The events on 27/11/2010 and 10/12/2011 in particular showed a very flashy response. These events had the highest peak hourly rainfall intensity during the study period and they occurred at the end of the flow season with low ASI (for the characteristics of these events see Table 1). The highest peak runoff rates for the study period were for the events on 27/11/2010 and 10/12/2011, which were 2.4 and 5.6 mm hr$^{-1}$, respectively. In contrast to most events,
20  the runoff response for the event on 27/11/2010 was transient with very rapid recession. For the event on 10/12/2011, a second peak of moderate rainfall intensity (about 10 mm hr$^{-1}$) produced a second runoff peak and there was a more significant recession flow following the rainfall bursts. This was also true for the other events shown in Figure 4, which were typical of responses to lower intensity rainfall during wetter (in terms of soil water) periods.

For events with $I_{peak} < 10$ mm hr$^{-1}$ there was a general increase in response as the ASI increased. The event on 12/11/2010
25  had 184 mm ASI and total rainfall was 28 mm and it did not produce any runoff. This was a typical example of no flow events. Coming into the runoff season, as ASI increased (e.g. 220 mm on 11/5/2011), RBF started to respond gradually, producing small amounts of runoff (e.g. for events on 11/5/2011 and 14/5/2011). When the ASI was > 250 mm for the event on 7/6/2011, it can be clearly seen that RBF responded to this low intensity, small size rainfall event with a delayed and smooth flow hydrograph with continued flow following the event. This also occurred for the next event on 1/8/2010.

30  **3.2 Runoff thresholds**

In Figure 1 we set out a number of thresholds that are important in runoff production mechanisms. We now explore the event data from the perspective of thresholds, concentrating on two key ones: catchment wetness and rainfall intensity. Figure 5 builds on approaches by Detty and McGuire (2010) and Janzen and McDonnell (2015). Figure 5a shows event runoff as a

**Number: 1  Author: Anonymous  Subject: Note  Date: 2016-07-29 18:46:08**

the overall seasonal pattern is the same but details not!

**Number: 2  Author: Anonymous  Subject: Note  Date: 2016-07-29 18:48:51**

which ones?

**Number: 3  Author: Anonymous  Subject: Note  Date: 2016-07-29 18:52:31**

whcih ones! fable them with a,b,c, so you can refer to them

**Number: 4  Author: Anonymous  Subject: Note  Date: 2016-07-29 18:54:06**

give value in ()

**Number: 5  Author: Anonymous  Subject: Note  Date: 2016-07-29 18:57:48**

that does what?

[Figure]

function of event rainfall, with the highest hourly rainfall intensity indicated by colour. Acknowledging that we have excluded rainfall events below 5 mm total rainfall, essentially any rainfall depth could produce a response at the catchment outlet, but there was a wide variation in runoff coefficients (indicated by the scatter). It is also clear that the events with high peak hourly intensity also had relatively large total rainfall accumulations.

5    Figure 5 goes about here

Figure 5b shows the impact of five factors together. The cumulative curve shows the distribution of soil water storage as observed through the study period. Specific events are shown with the ASI identified (left hand end of the grey lines) and the ASI plus rainfall depth (filled markers at the right end of the grey lines). The length of the lines is the rainfall depth. The colour shows the peak hourly rainfall intensity ($I_{peak}$) and the size of the bubbles shows the quick flow runoff coefficient.

10   Squares indicate events that did not produce any runoff or where the peak runoff rate was less than 0.05 mm hr$^{-1}$. There are several trends that can be discerned from Figure 5b. Rainfall events occurred across the full range of catchment wetness and were relatively evenly spread. The larger rainfall events generally occurred when ASI<250 mm and a mix of low and high intensity events occurred for these conditions. All the events on a wet catchment (ASI>250 mm) had low $I_{peak}$ ($\leq$6.2 mm hr$^{-1}$). Events where the ASI plus rainfall was less than 250mm usually did not generate any runoff, although there were some

15   high intensity exceptions and a small number of events with very low runoff coefficients (1-4%) where the ASI plus rainfall was generally between 240 and 250mm. These low runoff coefficient events were at the end of the runoff season.

Figures 5c and 5d look at the role of catchment wetness at the start (5c) and end (5d) of the event, combined with rainfall intensity, $I_{peak}$. The bubble size shows the quick flow runoff coefficient, as before, and crosses indicate rainfall events that did not produce runoff. Colour indicates the runoff volume. The runoff behaviour is separated into groups more clearly in

20   Figure 5d than in 5c. There are essentially three different groups including: 1) events without runoff where ASI+Rain<250mm and $I_{peak}$<10 mm hr$^{-1}$; 2) events that produce runoff when ASI+Rain > 250 mm; and 3) events with ASI+Rain<250 mm and $I_{peak}$>15 mmhr$^{-1}$ that did produce runoff (Tables 1 and 2). Where the ASI plus event rainfall exceeded 250 mm (group 2), some runoff was always produced.

Both of the first and third groups had ASI plus event rainfall less than 250 mm but they behaved differently in that some

25   produced runoff and others did not. In the first group low intensity rainfalls mostly happened in drier periods when ASI varied between 146 and 227 mm. These rainfall events completely infiltrated into the soil and they did not produce runoff (see Table 2 for event characteristics).

The third group, including events on 27/11/2010, 4/2/2011 and 10/12/2011, occurred during dry periods at the end of the flow season when the ASI was < 200 mm. These events were distinguished by having maximum hourly rainfall intensities

30   above ~15 mm hr$^{-1}$ and they did produce runoff. In particular, two of these events on 27/11/2010 and 10/12/2011 had the highest rainfall intensities observed ($I_{peak}$> 30 mm hr$^{-1}$) and they produced the highest peak runoff rates (8.1 mm hr$^{-1}$and 9.1 mm hr$^{-1}$) and hourly runoff totals (2.4 and 5.6 mm) observed during the study period (Figure 3). These runoff peaks were synchronised with the highest rainfall intensities. Antecedent flow for the events on the 27/11/2010 and 4/2/2011 was zero and the hydrograph rose and recessed quickly. For the event on the 10/12/2011, the ASI was 192 mm, the total rainfall was

**Number: 1  Author: Anonymous  Subject: Note  Date: 2016-07-29 18:59:33**

its not the RC but the Total Runoff

**Number: 2  Author: Anonymous  Subject: Note  Date: 2016-07-29 19:00:48**

Fig 5a could be described more. What about the relation with intensity?

**Number: 3  Author: Anonymous  Subject: Note  Date: 2016-07-29 19:18:56**

Fig 5b. needs much better explanation (consider to reduce information to make more clear).

**Number: 4  Author: Anonymous  Subject: Note  Date: 2016-07-29 19:16:44**

I like the figure but it has (too) many information in one plot in order to deliver your message.

readability:
Where can I quantify what the length of the grey line is? Where is zero for each individual line?

**Number: 5  Author: Anonymous  Subject: Note  Date: 2016-07-29 19:17:55**

but your legend suggest that the filled marker is RC?

**Number: 6  Author: Anonymous  Subject: Note  Date: 2016-07-29 19:26:49**

use other expression

**Number: 7  Author: Anonymous  Subject: Note  Date: 2016-07-29 19:22:52**

How much of Fig. 5b is actually describing the general seasonal pattern in rainfall event types?

**Number: 8  Author: Anonymous  Subject: Insert Text  Date: 2016-07-29 19:21:10**

rainfall events that state

**Number: 9  Author: Anonymous  Subject: Note  Date: 2016-07-29 19:27:28**

the role on what?

**Number: 10  Author: Anonymous  Subject: Note  Date: 2016-07-29 19:31:24**

(quantified as ASI and ASI+ Rain)

**Number: 11  Author: Anonymous  Subject: Note  Date: 2016-07-29 19:29:44**

(RC)

**Number: 12  Author: Anonymous  Subject: Note  Date: 2016-07-29 19:35:22**

say also that the y-axis is Ipeak (Fig 5 c , d)

**Number: 13  Author: Anonymous  Subject: Note  Date: 2016-07-29 19:37:09**

are these three groups statistically significant. It "looks" promising, they are!

**Number: 14  Author: Anonymous  Subject: Note  Date: 2016-07-29 19:34:53**

why? Guide the reader to your thoughts.

**Number: 15  Author: Anonymous  Subject: Strikeout  Date: 2016-07-29 19:39:16**

**Number: 16  Author: Anonymous  Subject: Note  Date: 2016-07-29 19:43:49**

[Figure]

function of event rainfall, with the highest hourly rainfall intensity indicated by colour. Acknowledging that we have excluded rainfall events below 5 mm total rainfall, essentially any rainfall depth could produce a response at the catchment outlet, but there was a wide variation in runoff coefficients (indicated by the scatter). It is also clear that the events with high peak hourly intensity also had relatively large total rainfall accumulations.

5  Figure 5 goes about here

Figure 5b shows the impact of five factors together. The cumulative curve shows the distribution of soil water storage as observed through the study period. Specific events are shown with the ASI identified (left hand end of the grey lines) and the ASI plus rainfall depth (filled markers at the right end of the grey lines). The length of the lines is the rainfall depth. The colour shows the peak hourly rainfall intensity ($I_{peak}$) and the size of the bubbles shows the quick flow runoff coefficient.

10  Squares indicate events that did not produce any runoff or where the peak runoff rate was less than 0.05 mm hr$^{-1}$. There are several trends that can be discerned from Figure 5b. Rainfall events occurred across the full range of catchment wetness and were relatively evenly spread. The larger rainfall events generally occurred when ASI<250 mm and a mix of low and high intensity events occurred for these conditions. All the events on a wet catchment (ASI>250 mm) had low $I_{peak}$ (≤6.2 mm hr$^{-1}$). Events where the ASI plus rainfall was less than 250mm usually did not generate any runoff, although there were some

15  high intensity exceptions and a small number of events with very low runoff coefficients (1-4%) where the ASI plus rainfall was generally between 240 and 250mm. These low runoff coefficient events were at the end of the runoff season.

Figures 5c and 5d look at the role of catchment wetness at the start (5c) and end (5d) of the event, combined with rainfall intensity, $I_{peak}$. The bubble size shows the quick flow runoff coefficient, as before, and crosses indicate rainfall events that did not produce runoff. Colour indicates the runoff volume. The runoff behaviour is separated into groups more clearly in

20  Figure 5d than in 5c. There are essentially three different groups including: 1) events without runoff where ASI+Rain<250mm and $I_{peak}$<10 mm hr$^{-1}$; 2) events that produce runoff when ASI+Rain > 250 mm; and 3) events with ASI+Rain<250 mm and $I_{peak}$>15 mmhr$^{-1}$ that did produce runoff (Tables 1 and 2). Where the ASI plus event rainfall exceeded 250 mm (group 2), some runoff was always produced.

Both of the first and third groups had ASI plus event rainfall less than 250 mm but they behaved differently in that some

25  produced runoff and others did not. In the first group low intensity rainfalls mostly happened in drier periods when ASI varied between 146 and 227 mm. These rainfall  completely infiltrated into the soil and they did not produce runoff (see Table 2 for event characteristics).

The third group, including events on 27/11/2010, 4/2/2011 and 10/12/2011, occurred during dry periods at the end of the flow season when the ASI was < 200 mm. These events were distinguished by having maximum hourly rainfall intensities

30  above ~15 mm hr$^{-1}$ and they did produce runoff. In particular, two of these events on 27/11/2010 and 10/12/2011 had the highest rainfall intensities observed ($I_{peak}$> 30 mm hr$^{-1}$) and they produced the highest peak runoff rates (8.1 mm hr$^{-1}$ and 9.1 mm hr$^{-1}$) and hourly runoff totals (2.4 and 5.6 mm) observed during the study period (Figure 3). These runoff peaks were synchronised with the highest rainfall intensities. Antecedent flow for the events on the 27/11/2010 and 4/2/2011 was zero and the hydrograph rose and recessed quickly. For the event on the 10/12/2011, the ASI was 192 mm, the total rainfall was

what do you mean by synchronized (peak timing?)

steaflow?

[Figure]

53 mm, with an initial flow of 0.13 mm hr$^{-1}$ (Table 1). It produced the highest observed peak hourly runoff of 5.6 mm, the runoff duration was 32 hours and total runoff was 41 mm. The highest intensity was observed in the first two hours of the event and 18% of the rainfall had become runoff by the end of those 2 hours. This compared to a maximum surface saturated extent of about 6% of the catchment area. The event on 10/12/2011 marked the end of a particularly rainy period, with more than 200 mm over 30 days. Note that due to equipment being removed after this event, this event was the last recorded flow event at RBF.

**3.3  runoff processes and thresholds**

The above presentation of results from Figure 5d identifies a threshold catchment wetness expressed as antecedent soil water storage plus event rainfall depth of 250 mm above which runoff always occurred and another threshold of hourly rainfall depth exceeding 15 which also led to runoff production. Looking at events in the lower right quarter of Figure 5d also shows that the event runoff coefficient tends to increase as either catchment wetness or peak hourly intensity increases. These results suggest that there are both a wetness dependent and an intensity dependent runoff production mechanisms operating. This section examines the evidence for different runoff mechanisms contributing to event runoff.

**3.3.1 Hillslope wetness-flow response relationships**

Figure 6 shows the runoff time series together with water level time series at several shallow piezometers; sites 4, 5, 3, 32, 2 and 1 (Figure 2). All sites except 4 were located in drainage lines. The sites are organized by elevation from highest to lowest in the catchment. Manually read sites are shown with dashed lines. Figure 6 clearly shows that the duration of saturation increased downslope. Comparing the runoff time series with the piezometer record for sites 1, 2 and 32, it is clear that the water table rose to the surface in the upper parts of the riparian zone during runoff events. Furthermore, the lower half of the riparian zone remained saturated to the surface for long periods during the runoff season. The record for site 3 indicates that the water table at this site did not rise to the surface, even during events.

Figure 6 goes about here

Looking at the flow record, there were periods where significant baseflow persisted between events. These correspond to periods where the water table at site 5 was above about 120 cm and at site 4 was above about 140 cm. Flow became more strongly persistent between rainfall events as the water table at sites 4 and 5 rose further. The water table recessions at sites 4 and 5 correspond with flow recessions when the water table was above 120 cm and 140 cm at sites 4 and 5, respectively. Figure 7 shows the relationship between water table levels on the hillslope (sites 4 and 5) and soil water storage at the weather station. Site 5 in particular shows a strong change in behaviour for soil water storage above 250 mm. Above this threshold much higher water tables were typically observed and those water tables showed relatively rapid recession when shallower than 120 cm. Similar observations were seen at site 4 but the threshold depth was 140 cm. Figure 7 thus explains the linkage between the 250mm ASI plus event rainfall threshold and runoff. When soil water contents exceeded this level, water tables rose into a more permeable zone and lateral subsurface drainage occurred, as evidenced by the recessions. That

**Page:9**

**Number: 1  Author: Anonymous  Subject: Note  Date: 2016-07-29 19:52:02**

the RC calculated for the first 2 hours was 18% (is that what you wanted to say?)

**Number: 2  Author: Anonymous  Subject: Note  Date: 2016-07-29 19:53:30**

You could summarize all findings in one or two sentences at the end. (also applies to other result sections)

**Number: 3  Author: Anonymous  Subject: Strikeout  Date: 2016-07-30 11:59:35**

**Number: 4  Author: Anonymous  Subject: Note  Date: 2016-07-30 10:23:12**

You need to calculate the actual value. In the discussion it is OK to round it to a meanungfull value.

**Number: 5  Author: Anonymous  Subject: Note  Date: 2016-07-30 10:26:57**

This is based on 4 datapoints and you have a gap in data between ca. 10 and 15 mm total rainfall. I think it is valid to state the 15 mm threshold but you need to be critical about it in the text.

**Number: 6  Author: Anonymous  Subject: Note  Date: 2016-07-30 10:31:42**

you need to do a statistical test to prove this (e.g., rank correlation).

**Number: 7  Author: Anonymous  Subject: Note  Date: 2016-07-30 10:36:10**

what do you mean? Where they aligned along transects?

**Number: 8  Author: Anonymous  Subject: Note  Date: 2016-07-30 10:44:56**

please help the reader to better understand this. Saturation would mean to me, that the entire soil profile is saturated and the groundwater level at the soil surface. That's, only the case for well 1 and 2 (maybe for 32).

**Number: 9  Author: Anonymous  Subject: Note  Date: 2016-07-30 10:38:33**

What do you mean. Manually read groundwater levels? If so please state this in the method section. It seemed, you were using Odyssey loggers for all wells.

If you used irregular intervals, please use a line type that has a dot at each time you were manually taking a reading.

**Number: 10  Author: Anonymous  Subject: Note  Date: 2016-07-30 10:52:25**

I think it is hard to say something about recession details then plotting 1.5 years of data (x-axis). If you want to say something about the relationship between certain groundwater levels and streamflow, you should either show selected events or even better use statistical tests that show, that streamflow is statistically higher, when certain gw-levels (e.g., in well 4 and 5) are above a theshold of XY cm. That would nicely fit your topic.

**Number: 11  Author: Anonymous  Subject: Note  Date: 2016-07-30 11:03:55**

Why do you use soil water storage (or are you actually using soil moisture (VWC)?
You compare groundwater on the hillslope with soil moisture in the riparian zone?

Why not plotting streamflow as a function of depth to groundwater or streamflow as a function of soil water storage?

**Number: 12  Author: Anonymous  Subject: Note  Date: 2016-07-30 10:56:12**

Can you remind the reader that site 4 and  5 are on the hillslope

**Number: 13  Author: Anonymous  Subject: Note  Date: 2016-07-30 10:57:14**

It looks lie an exponential relation rather than a threshold.

**Number: 14  Author: Anonymous  Subject: Note  Date: 2016-07-30 11:05:46**

This is descriptive. Please can you  quantify this using statistics using all you events.

[Figure]

53 mm, with an initial flow of 0.13 mm hr$^{-1}$ (Table 1). It produced the highest observed peak hourly runoff of 5.6 mm, the runoff duration was 32 hours and total runoff was 41 mm. The highest intensity was observed in the first two hours of the event and 18% of the rainfall had become runoff by the end of those 2 hours. This compared to a maximum surface saturated extent of about 6% of the catchment area. The event on 10/12/2011 marked the end of a particularly rainy period, with more

5    than 200 mm over 30 days. Note that due to equipment being removed after this event, this event was the last recorded flow event at RBF.

**3.3  runoff processes and thresholds**

The above presentation of results from Figure 5d identifies a threshold catchment wetness expressed as antecedent soil water storage plus event rainfall depth of 250 mm above which runoff always occurred and another threshold of hourly rainfall

10   depth exceeding 15 which also led to runoff production. Looking at events in the lower right quarter of Figure 5d also shows that the event runoff coefficient tends to increase as either catchment wetness or peak hourly intensity increases. These results suggest that there are both a wetness dependent and an intensity dependent runoff production mechanisms operating. This section examines the evidence for different runoff mechanisms contributing to event runoff.

3.3.1 Hillslope wetness-flow response relationships

15   Figure 6 shows the runoff time series together with water level time series at several shallow piezometers; sites 4, 5, 3, 32, 2 and 1 (Figure 2). All sites except 4 were located in drainage lines. The sites are organized by elevation from highest to lowest in the catchment. Manually read sites are shown with dashed lines. Figure 6 clearly shows that the duration of saturation increased downslope. Comparing the runoff time series with the piezometer record for sites 1, 2 and 32, it is clear that the water table rose to the surface in the upper parts of the riparian zone during runoff events. Furthermore, the lower

20   half of the riparian zone remained saturated to the surface for long periods during the runoff season. The record for site 3 indicates that the water table at this site did not rise to the surface, even during events.

Figure 6 goes about here

Looking at the flow record, there were periods where significant baseflow persisted between events. These correspond to periods where the water table at site 5 was above about 120 cm and at site 4 was above about 140 cm. Flow became more

25   strongly persistent between rainfall events as the water table at sites 4 and 5 rose further. The water table recessions at sites 4 and 5 correspond with flow recessions when the water table was above 120 cm and 140 cm at sites 4 and 5, respectively. Figure 7 shows the relationship between water table levels on the hillslope (sites 4 and 5) and soil water storage at the weather station. Site 5 in particular shows a strong change in behaviour for soil water storage above 250 mm. Above this threshold much higher water tables were typically observed and those water tables showed relatively rapid recession when

30   shallower than 120 cm. Similar observations were seen at site 4 but the threshold depth was 140 cm. Figure 7 thus explains the linkage between the 250mm ASI plus event rainfall threshold and runoff. When soil water contents exceeded this level, water tables rose into a more permeable zone and lateral subsurface drainage occurred, as evidenced by the recessions. That

you only gave one Ksat, So you do you know that the lower soil horizon is less permeable (That is a reasonable assumption you could use in the discussion but not in the results section).

subsurface drainage form the hillslopes to the stream?

You base your conclusion on data of two wells?

[Figure]

is, the hillslope was becoming connected to the catchment outlet via subsurface flow. There were a few occasions where the water table responded strongly for soil water storage less than 250 mm. As indicated by the red colour, these corresponded to high intensity rainfall events. It is not clear exactly how the water moved rapidly into the wells in these cases but it could be due to preferential flow through macropores.

5    Figure 7 goes about here

Flow recessions provide information on the drainage characteristics of catchments. Figure 6 shows that the hillslope flow usually ceased between events during the wet period, with no flow during dry periods. However, in August and early September 2010, continuous hillslope flow endured for a month (Figure 6). There was also a marked variation in the recession behaviour during August/September 2010. To explore this, we calculated the recession constant, $K$ (as in

10    $Q = Q_0 e^{-kt}$), and plotted it against soil water storage at the start of the recession for individual events within this period (Figure 8). $K$ decreased as soil water storage increased. Considering this and the transient nature of flow during dry periods, it is clear that the wetter the catchment is, the slower the recessions are. By inference, this  greater (perhaps more spatially extensive) subsurface connectivity is providing flows from the hillslope and maintaining catchment discharge during wetter conditions.

15    Figure 8 goes about here

**3.3.2 Isotope and major ion results**

Clearly, subsurface flow is often important in this catchment, suggesting that the event outflow would be dominated by "old" water; however, the saturated area in the lowest parts of the catchment would also be expected to produce direct flows of "new" water. Figure 9 shows an event from 12 August 2010 during the wettest part of the study period. The antecedent soil

20    water storage at the beginning of this event was 274 mm and total rainfall was 17 mm. Stable isotope data was available for the event and is shown on Figure 9. Over the study period $\delta^2H$ for rainfall varied between -7‰ and -83‰ and $\delta^2H$ for low flows was highly damped. Low flow samples from the RBF flume before and after the event showed a $\delta^2H$ of -27‰ and rainfall for this event was strongly depleted (3 samples prior to and during the event $\delta^2H$ = -42, -67 and -57‰), compared with low flow. The runoff samples showed a very different signature in the rising limb and the peak of the hydrograph

25    (-43‰) in comparison to antecedent low flow (-27‰). Using a two-component mixing model, we estimate that the peak contribution of new water was about 50%, and the new water contribution to the event volume was 17%, corresponding to 5% of the rainfall depth. The timing of the new water contribution matched the main rainfall burst well with relatively rapid return to the typical old water isotopic signature following the cessation of rainfall. These results suggest that precipitation on the saturated area generates direct runoff in amounts that would be expected given that the saturated area is around 5-6%

30    of the catchment area.
Figure 9 goes about here

**Number: 1  Author: Anonymous  Subject: Note  Date: 2016-07-30 11:09:48**

Can you use your isotope data to prove this?

**Number: 2  Author: Anonymous  Subject: Note  Date: 2016-07-30 11:10:10**

... in Fig.xy

**Number: 3  Author: Anonymous  Subject: Note  Date: 2016-07-30 11:11:07**

Please move this to the discussion section. There it is possible to speculate about potential causes.

**Number: 4  Author: Anonymous  Subject: Note  Date: 2016-07-30 11:21:36**

you only measure groundwater levels and not flow.

**Number: 5  Author: Anonymous  Subject: Note  Date: 2016-07-30 11:22:20**

unless you have a trench, you cannot say anything about hillslope flow.

**Number: 6  Author: Anonymous  Subject: Note  Date: 2016-07-30 11:27:46**

... of streamflow (or of the groundwater levels?)
Q would indicate streamflow so you where plotting k of stremaflow as a function of volumetic water content (which is also measured near the outlet of the catchment. What do we learn form this in terms of connectivity between hillslope and streams (see title)?

**Number: 7  Author: Anonymous  Subject: Note  Date: 2016-07-30 11:28:16**

Please quantify!

**Number: 8  Author: Anonymous  Subject: Replace  Date: 2016-07-30 11:29:03**

suggests

**Number: 9  Author: Anonymous  Subject: Note  Date: 2016-07-30 11:36:50**

It is OK to show only one event in the paper but for deriving conclusions you need to analyse many events and give the statistic. You have a good data set, so please use it!

**Number: 10  Author: Anonymous  Subject: Note  Date: 2016-07-30 11:34:10**

in what amounts (please check sentence to be complete)

**Number: 11  Author: Anonymous  Subject: Note  Date: 2016-07-30 11:34:54**

This result is different to what you concluded from the hydrometric data.

[Figure]

Another interesting event is the higher intensity ($I_{peak}$ = 15 mm hr$^{-1}$) event on 8/11/11. Major ion geochemistry data were available for this event. Figure 10a shows the typical relationship between flow and chloride concentration, with samples from this event identified by red. Figure 10b shows the time series of chloride concentration along with the hydrograph. The first and second chloride samples respectively plot above and within the typical scatter of data on Figure 10a, while the

5    remaining samples plot well below the typical chloride concentration. Given the late spring timing of this event, the first sample probably reflects some evapoconcentration of solutes in the riparian area. The flow shows a rapid peak in response to the main rainfall burst followed by a sustained relatively low flow and a recession over the second half of the day suggestive of subsurface flow. The volume of the main peak corresponds to around 5% of the rainfall. The flow after the main peak had a surprisingly low concentration of chloride (the cluster of low concentration red points on Fig 10a), which may suggest that

10   the higher intensity activated either overland or preferential flow paths, limiting soil contact time and leading to this low concentration.

Figure 10 goes about here

**4 Discussion**

**4.1 Runoff mechanisms**

15   The hydrometric data enables us to identify the important runoff mechanisms under different circumstances. The isotope and major ion geochemistry data provide further supporting evidence. The rainfall plus antecedent soil water storage threshold of 250 mm that needs to be exceeded for runoff in most circumstances shows that wetness dependent runoff processes are important, that is either saturation excess or subsurface stormflow. Shallow groundwater data combined with field mapping of surface saturated areas shows that complete profile saturation is limited to about 5% of the catchment area and this

20   saturation is persistent over the winter-spring season. Field observations show this saturated area is highly connected to the catchment outlet and it would be expected to produce SOF. The isotope results for 12 August 2010 enabled the event runoff to be separated into new water and old water contributions. Five percent of the rainfall volume on the catchment appeared in the event runoff, which corresponds to the proportion of surface saturated area in the catchment, supporting the contention of significant saturation excess runoff from this part of the catchment, as observed elsewhere (McGlynn and McDonnell, 2003).

25   While saturation excess runoff undoubtedly occurs, many of the event runoff coefficients were well in excess of 5% and they approach 1 under very wet conditions (Figure 5). The event on 12 August showed substantial old water contribution, logged shallow wells show that the water table did not reach the surface on the steeper hillslope areas, even within the convergent drainage lines under very wet conditions (e.g. sites 5, 7). The recession behaviour of wells on the hillslope suggests subsurface flow is moving from the hillslope under wet conditions and the recession constant analysis shows that this

30   connection becomes stronger as the hillslope wets beyond 250 mm of stored water. This is all consistent with a substantial contribution of subsurface flow to event runoff once the hillslope is sufficiently wet to establish connection to the riparian

**Number: 1  Author: Anonymous  Subject: Note  Date: 2016-07-30 11:41:24**

Also here: I tis OK to show one example but you need to analyse all your events and present statistics of the general behaviour. You have a good dataset!

**Number: 2  Author: Anonymous  Subject: Note  Date: 2016-07-30 11:39:07**

What is a typical Chloride concentration (define!)

**Number: 3  Author: Anonymous  Subject: Note  Date: 2016-07-30 11:40:16**

Please put your interpretation in the discussion section and present only results.

**Number: 4  Author: Anonymous  Subject: Note  Date: 2016-07-30 13:11:34**

The conclusions are drawn from one or two individual rainfall events and not logically derived from the results of this study but more based on general hydrological conceptualization.

The discussion is short and could better tie in the results of this study and critically discuss them.

I think it is good to discuss the findings in the light of Fig1. But Figure 1 is for poin- or plot scale assessments and misses out spatial (and temporal) heterogeneity across a catchment.

Fig. 1, in my opinion, also misses out connectivity and interactions between sites (e.g., run-on form uphill sites). Your idea to bring that in is good but I think it needs a separate scheme to do this because of different scales of consideration.

**Number: 5  Author: Anonymous  Subject: Note  Date: 2016-07-30 12:03:43**

not so clear form the results

**Number: 6  Author: Anonymous  Subject: Note  Date: 2016-07-30 12:05:04**

I think your conclusions from the isotope data are showing a contrast rather support the hydrometric data (see also your title).

**Number: 7  Author: Anonymous  Subject: Note  Date: 2016-07-30 12:08:16**

Isn't it varying between events and seasons?

**Number: 8  Author: Anonymous  Subject: Note  Date: 2016-07-30 12:07:45**

Please give evidence rather than your subjective field impression.

**Number: 9  Author: Anonymous  Subject: Note  Date: 2016-07-30 12:08:46**

one event is not enough to draw conclusions from it!

**Number: 10  Author: Anonymous  Subject: Note  Date: 2016-07-30 12:11:41**

This is rather general, please discuss, what you study contributed on new insights on multiple runoff processes occurring in parallel.

[Figure]

area, as has been inferred in other studies (Buttle et al., 2004;Detty and McGuire, 2010;Hewlett and Hibbert, 1967;Jencso et al., 2009;Penna et al., 2011).

Perhaps more surprisingly, there was a group of events that produced runoff under conditions of relatively low soil water content (ASI + rainfall < 250 mm) but high rainfall intensity. This suggests an intensity dependent runoff process is being

5 triggered when rainfall exceeds some threshold for sufficient time, in this case about 15 mm of rainfall in an hour. The runoff coefficients for the four events exceeding 15 mm hr$^{-1}$ peak hourly intensity are 3, 12, 20 and 68% for peak hourly intensities of 16, 30, 15 and 31 mm hr$^{-1}$ and event rainfall + ASI of 202, 215, 257 and 245 mm respectively. Note that one of these events exceeds both the wetness and intensity thresholds. It is tempting to assume that this evidence shows surface runoff due to infiltration excess runoff, but it is also possible that the high rainfall intensities are efficiently activating

10 macropore networks (Beven and Germann, 1982) and that the flow could be following subsurface pathways.

The hydrograph from the event on 8/11/11 (the event that exceeded both thresholds, Figure 10) shows both a rapid runoff and a delayed runoff response. The concentration of chloride was unusually low during the delayed runoff component compared with all other events with major ion data. This may suggest limited contact with the catchment soils which could occur if macropore flow was important but it is not definitive. Of the two events with peak hourly intensities around 15 mm

15 hr$^{-1}$, one also exceeded the wetness (rainfall + ASI) threshold of 250 mm and the other had a very low runoff coefficient (only 3%) and hence these two events are somewhat equivocal in terms of the importance of intensity. However, the two events with peak hourly rainfall intensities around 30 mm hr$^{-1}$ both produced rapid runoff responses without a significant delayed component (Figure 4) and had runoff coefficients (12 and 68%) well in excess of the surface saturated area (5%) in the catchment, showing clear evidence of the role of an intensity of 30 mm hr$^{-1}$. Unfortunately isotope and major ion data

20 were not available for those events to attempt to determine whether surface or subsurface pathways are important. Overall there is clear evidence for intensity dependent runoff mechanisms, especially for the largest 30 mm hr$^{-1}$ events.

In summary, the process evidence suggests a catchment where subsurface flow leads to a seasonally saturated riparian area that produces saturation excess runoff. This saturation excess is augmented by subsurface stormflow when the catchment wetness (rainfall + ASI) exceeds a 250 mm threshold. This subsurface flow exfiltrates in the riparian area. When hourly

25 rainfalls exceed a threshold of 15-30 mm hr$^{-1}$, an intensity dependent runoff process is activated that also contributes flow from the hillslope area outside the riparian zone. It is not clear whether this is a purely surface runoff process or not.

**4.2 Thresholds and connectivity in runoff production**

We identified two important thresholds in the catchment response. The first is a wetness threshold of event rainfall plus ASI exceeding 250 mm. Under these conditions the water table approaches the surface in the riparian area and water tables rise

30 on the hillslope into what is inferred from relatively rapid hillslope water table recessions to be a more transmissive part of the soil profile (within ~120-140 cm of the surface). We infer that the hillslope becomes connected to the riparian zone under these conditions. Similar catchment wetness thresholds for connectivity and runoff generation have been reported elsewhere (Detty and McGuire, 2010;Penna et al., 2011;Tromp-van Meerveld and McDonnell, 2006b). These have been expressed

Number: 1  Author: Anonymous  Subject: Note  Date: 2016-07-30 12:12:17

Please discuss in more detail, what your study and other studies found.

Number: 2  Author: Anonymous  Subject: Note  Date: 2016-07-30 12:12:48

unfortunately only 4 events

Number: 3  Author: Anonymous  Subject: Note  Date: 2016-07-30 12:14:21

You used hourly data but a different temporal resolution would lead you to a different threshold. Please, discuss this issue!

Number: 4  Author: Anonymous  Subject: Note  Date: 2016-07-30 12:15:03

better move to results

Number: 5  Author: Anonymous  Subject: Note  Date: 2016-07-30 12:16:22

Please use evidence form your result section to better corroborate this.

Number: 6  Author: Anonymous  Subject: Note  Date: 2016-07-30 12:18:25

You cannot conclude this from one event, only.

Number: 7  Author: Anonymous  Subject: Note  Date: 2016-07-30 12:19:11

is it intensity or storage capacity?

Number: 8  Author: Anonymous  Subject: Note  Date: 2016-07-30 12:20:37

A saturated area would immediately produce saturation excess overland flow if hit by a rainfall.

Number: 9  Author: Anonymous  Subject: Note  Date: 2016-07-30 12:22:00

You assume, that "old water" is from the hillslope but you need to sample water there in order to prove this. In regards to your chosen title, I think this would be something I had expected.

Number: 10  Author: Anonymous  Subject: Note  Date: 2016-07-30 12:31:56

I doubt if one threshold (250 mm) is very informative for two very contrasting landscape units (riparian zone and hillslope). I would assume that they have different threshold or 250 mm are more related to the hillslope.

Number: 11  Author: Anonymous  Subject: Note  Date: 2016-07-30 12:32:28

Please give soil properties for this soil horizon in the method section.

Number: 12  Author: Anonymous  Subject: Note  Date: 2016-07-30 12:33:00

please use your data to give evidence

Number: 13  Author: Anonymous  Subject: Note  Date: 2016-07-30 12:35:48

I am not sure, if you analyzed wetness thresholds of connectivity. Please refer to your data to do so. I think it is rather an assumption, that if we see runoff response (of "old water") in the stream we think it is from the hillslope.

[Figure]

either in terms of rainfall depth, antecedent soil water storage conditions, or a combination of these. Similar to our results, in a study of a forested subcatchment with highly permeable soils and a small riparian area (Detty and McGuire, 2010) found a  relationship between a threshold of 316 mm for ASI plus rain and the start of the event flow  and demonstrated that the ASI+rainfall threshold corresponded with a water table height threshold. They also suggested that

5   subsurface flow, transmissivity feedback and preferential flow can be used to explain runoff mechanisms. They did not observe either Hortonian overland flow or SOF even during the largest events.

In other studies, total rainfall has been found to be the main controller of threshold behaviour (Fujimoto et al., 2008;McGuire and McDonnell, 2010). Analysing 147 rain events in the Panola catchment, USA, Tromp-van Meerveld and McDonnell (2006b) defined an event precipitation depth threshold of 55 mm for initiation of connectivity and subsurface runoff. They

10  (Tromp-van Meerveld and McDonnell, 2006a) proposed the "fill and spill" mechanism and suggested that subsurface connection happens through connectivity of transient bedrock perched areas when depression storage of bedrock fills and water spills over micro-topographic relief. Figure 5a shows that total rainfall is a poor predictor of runoff behaviour in our case.

A second threshold associated with high rainfall intensities was also evident. A similar role of intensity has also been
15  observed by Janzen and McDonnell (2015) who found that the Panola hillslope can produce significant runoff from dry antecedent conditions when high intensity rainfall occurs. In general event runoff from Panola is controlled by catchment wetness, similar to our hillslope. These are the only two studies we know of that have reported intensity thresholds in catchments where runoff is normally dominated by wetness thresholds. This may be a consequence of such events being relatively rare in any given catchment (roughly 10% of runoff producing events in our case).

20  **4.3 Runoff processes framework**

We now consider the three runoff processes occurring in the catchment in relation to the framework proposed in Figure 1 and the various flux and timescale thresholds identified therein. Essentially Figure 1 is posing a series of questions that allow us to systematically think through the runoff processes. Above we have identified three groups of events – those that do not produce runoff, those that produce runoff by saturation excess and subsurface stormflow from a wet catchment and
25  those that produce runoff from higher intensity events.

The rainfall events that do not produce runoff are not exceeding infiltration capacity for sufficient time for runoff to flow from the catchment (box 1, "No"). They may or may not produce significant percolation (box 2) but if any of these events do produce percolation to a perched water table, this only results in an ephemeral water table that dissipates before lateral flow can move water down the hillslope (box 3, "No") and they do not saturate the full profile (box 4, "No"). As a consequence
30  no event runoff is produced. These conditions correspond to the local control state of Grayson et al. (1997).

Some high intensity rainfall events on a dryer catchment do exceed the infiltration capacity for sufficiently long periods of time (box 1, "Yes"; hourly intensity of 15-30 mm hr$^{-1}$) and these events produce runoff. The increase in runoff coefficient as intensity increases for a given ASI+rainfall in Figure 5 suggests that infiltration thresholds may also be playing a role for

Number: 1  Author: Anonymous  Subject: Strikeout  Date: 2016-07-30 12:37:06

Number: 2  Author: Anonymous  Subject: Replace  Date: 2016-07-30 12:41:24

NO! Figure 3 in Detty &  McGuire plots streamflow on the y-axis! So they do not say anything about the hillslope contribution but about the entire watersehd runoff.

Number: 3  Author: Anonymous  Subject: Note  Date: 2016-07-30 12:43:54

Please be more speciffic about what they found for what landscape position

Number: 4  Author: Anonymous  Subject: Insert Text  Date: 2016-07-30 12:43:03

 in their catchment

Number: 5  Author: Anonymous  Subject: Note  Date: 2016-07-30 12:44:29

threshold behaviour of what (streamflow, groundwater, soil moisture)?

Number: 6  Author: Anonymous  Subject: Note  Date: 2016-07-30 12:52:36

What about other studies?

Number: 7  Author: Anonymous  Subject: Note  Date: 2016-07-30 12:53:23

threshold to initiate what? Streamflow, groundwater response?

Number: 8  Author: Anonymous  Subject: Note  Date: 2016-07-30 12:54:32

Please be more clear in the section before what your three runoff processes are and why?

Number: 9  Author: Anonymous  Subject: Note  Date: 2016-07-30 12:59:42

I think it is good to discuss your findings in the light of Fig1. But you miss out a spatial difference in runoff generation mechanisms across the catchment and treat all as one! I think this is not insightful enough! The scheme in Figure 1 should be applied to different landscape units in your catchment and could so reveal a mosaic of processes. Next step would be to discuss connectivity between thse mosaic parts. That's something different!

[Figure]

wetter conditions (i.e. box 1 "Yes) but that infiltrated water (the dashed link in Figure 1) also contributes through other mechanisms.

The final group of events are those that produce runoff from a wet catchment at low intensity rainfall. These follow the path box 1 "No", box 2 "Yes" and box 3 "Yes" in Figure 1. On the upper hillslope parts of the catchment the subsurface flow capacity is sufficient that the water table does not reach the surface and water drains either during or shortly after the storm (box 6, "Yes"). Subsurface connectivity develops during the event and subsurface flow dominates. In the riparian area in the lower part of the catchment there is a substantial reduction in lateral subsurface flow capacity due to much lower slopes (and probably lower hydraulic conductivity associated with poorly structured, poorly drained soils) and this area drains very slowly; taking longer than the typical time between events in the wet season (box 7, "Yes"), resulting in saturation excess runoff from the lower catchment. Hence under wet conditions this catchment produces a mix of saturation excess and subsurface storm flow, but from geographically distinct parts of the catchment.

The above illustrates how the framework in Figure1 can be used to understand the role of different thresholds regarding fluxes and timescales in determining runoff mechanisms. Such a framework is likely to be particularly valuable where there is a mix of runoff mechanisms operating for different events or in different parts of the catchment. Our study catchment nicely illustrates such a mixture.

**5 Conclusion**

This study has examined the role of intensity and wetness thresholds in determining runoff responses for an agricultural hillslope in the Lang Lang River catchment, Victoria, Australia. Both intensity dependent and wetness dependent thresholds were identified in the runoff response. During wet conditions, hydrological connectivity has a strong influence on water delivery to the riparian area. Saturation excess runoff from the riparian zone was also important. The results of this study demonstrated that:

1) Runoff generation in most events is dependent on the catchment connectivity and soil moisture conditions. When the sum of the antecedent soil water storage and event rainfall exceeded 250 mm, runoff was typically produced by a mix of saturation excess and subsurface storm flow. Under these conditions, a water table forms in the soil and a saturated area develops in the riparian zone. When the water level rises to within about 1 m of the surface at mid slope sites, rapid subsurface flow paths are activated which connected the mid slope and riparian area, contributing event flow to the hillslope flume.

2) When the catchment became very wet, high water levels persisted at the mid slope sites which remained hydrologically connected to the riparian area and baseflow became persistent between events.

3) High rainfall intensity events produced runoff even when the soil water content plus event rainfall content was below the 250 mm threshold. This could be due to either Hortonian overland flow or fast subsurface preferential flow paths being activated.

**Number: 1  Author: Anonymous  Subject: Note  Date: 2016-07-30 13:01:02**

give evidence why you assume this?

**Number: 2  Author: Anonymous  Subject: Note  Date: 2016-07-30 13:02:05**

I am not is sure if it is only one process over the entire event

**Number: 3  Author: Anonymous  Subject: Note  Date: 2016-07-30 13:03:14**

I am not so sure: I think Fig 1 is for hte point-or plot-scale and needs an additional step (assessing connectivity) be be meaningful on the catchment scale.

**Number: 4  Author: Anonymous  Subject: Note  Date: 2016-07-30 14:05:26**

General Comment:
The conclusions are drawn from one or two individual rainfall events and not logically derived or supported from the results of this study. I would encourage the authors to refine and strengthen their analysis based on more of their dataset. I think it is good to discuss the findings in the light of Fig1. but as it is originally developed for point- or plot scale assessments it misses out the spatial (and temporal) heterogeneity across a catchment. – a fundamental aspect when analyzing thresholds and connectivity.atial variability within the catchment.

**Number: 5  Author: Anonymous  Subject: Note  Date: 2016-07-30 13:13:23**

Soil moisture is measured at one site in the catchment?

**Number: 6  Author: Anonymous  Subject: Note  Date: 2016-07-30 13:13:02**

I think you are limited in how much you can say about connectivity

**Number: 7  Author: Anonymous  Subject: Note  Date: 2016-07-30 13:14:05**

This is your assumption not clear from data

**Number: 8  Author: Anonymous  Subject: Note  Date: 2016-07-30 13:15:06**

I think flow and connectivity cannot be derived form your data. You assume, that this happens.

**Number: 9  Author: Anonymous  Subject: Note  Date: 2016-07-30 13:16:59**

What is the hillslope flume? It appears here for the first time do you men the streamflow flume at the catchmetn outlet?!

[Figure]

We have also advanced a set of threshold conditions or questions (Figure 1) that allow a logical examination of which runoff mechanism are likely to be important in a catchment given, thresholds regarding fluxes and timescales. This framework provides a useful way of thinking through the controls on rainfall-runoff response as conditions change either between events or between different parts of the catchment. It is illustrated using the behaviour of this catchment.  Our study

5  catchment demonstrates a mix of intensity dependent and wetness dependent processes, something which has been rarely reported for humid catchments.

**Acknowledgements**

The project was funded by the Australian Research Council (grant DP0987738). Saffarpour was supported by an International Postgraduate Research Scholarship (IPRS) from the Australian Government and an International Research

10  Scholarship from the University of Melbourne (MIRS). Peter and Wilma Mackay provided access to their farm for the study. Many people helped in the field, especially Rodger Young, Lucas Dowell, Olaf Klimczak and Jacqui Lloyd. The water samples were analysed for $^{18}$O and $\delta^2$H, either by Ian Carwright's Earth Sciences laboratory, Monash University or by Russell Drysdale's isotope laboratory at the University of Melbourne. The chloride data were analysed by Mike Grace's Water Studies Laboratory, Monash University.  Tony Weatherly assisted with soil interpretation at the site.

I think Fig1 is a point or plot scale

[revised manuscript text omitted]

No Comments.

[Figure]

**Table 1. Rainfall-runoff events summary at RBF**

| Grp No
[Figure]
 | Date | Rain dur'n (hr) | Total rainfall (mm) | Peak hourly rainfall intensity (mm/hr) | ASI (mm) | ASI+ rainfall (mm) | Runoff duration (hr) | Total runoff (mm) | Quick flow (mm) | RC (%) | Sat'd area (%) |
|---|---|---|---|---|---|---|---|---|---|---|---|
| 2 | 25/06/2010 | 49 | 22.4 | 1.8 | 237 | 259 | 74 | 7.6 | 1 | 4 | 4.5 |
| 2 | 13/07/2010 | 33 | 23.8 | 5.4 | 237 | 261 | 41 | 3.5 | 0.54 | 2 | |
| 2 | 20/07/2010 | 15 | 9.2 | 4.2 | 272 | 281 | 32 | 3.1 | 0.8 | 9 | 5.3 |
| 2 | 1/08/2010 | 30 | 31.4 | 6.2 | 253 | 284 | 67 | 19.5 | 13.7 | 44 | 5.5 |
| 2 | 5/08/2010 | 52 | 23.6 | 2 | 275 | 299 | 96 | 21.6 | 7 | 30 | 5.5 |
| 2 | 12/08/2010 | 34 | 17.2 | 5 | 274 | 291 | 94 | 12.7 | 4.3 | 25 | 5.5 |
| 2 | 16/08/2010 | 50 | 19.2 | 3.4 | 275 | 294 | 73 | 16.9 | 7.7 | 40 | 5.5 |
| 2 | 18/08/2010 | 34 | 14.6 | 4 | 286 | 301 | 57 | 13.8 | 4.6 | 32 | 5.5 |
| 2 | 24/08/2010 | 4 | 9.6 | 5.2 | 273 | 283 | 24 | 4.1 | 1.6 | 17 | 5.5 |
| 2 | 25/08/2010 | 59 | 12.6 | 2 | 285 | 298 | 73 | 10 | 1.4 | 11 | 5.5 |
| 2 | 31/08/2010 | 22 | 12.4 | 2 | 271 | 283 | 80 | 9.5 | 0.8 | 6 | 5.5 |
| 2 | 5/09/2010 | 47 | 25.2 | 3.4 | 272 | 297 | 74 | 23.1 | 13.5 | 54 | 5.5 |
| 2 | 9/09/2010 | 17 | 8.0 | 2.4 | 264 | 272 | 16 | 1.1 | 0.2 | 3 | 5.3 |
| 2 | 6/10/2010 | 18 | 15.2 | 6.6 | 231 | 246 | 3 | 0.37 | 0.27 | 2 | na |
| 2 | 15/10/2010 | 59 | 34.2 | 2.8 | 228 | 262 | 86 | 11.3 | 3.9 | 11 | na |
| 2 | 23/10/2010 | 11 | 12.8 | 5.2 | 227 | 240 | 8 | 1 | 0.57 | 4 | na |
| 2 | 30/10/2010 | 29 | 32.8 | 8.8 | 202 | 235 | 16 | 1.2 | 0.83 | 3 | na |
| 3 | 27/11/2010 | 19 | 54.4 | 30.4 | 161 | 215 | 31 | 7.6 | 6.5 | 12 | na |
| 2 | 19/12/2010 | 49 | 25.6 | 3.8 | 191 | 217 | 10 | 0.83 | 0.18 | 1 | na |

Number: 1  Author: Anonymous  Subject: Note  Date: 2016-07-29 18:28:12

typo

Number: 2  Author: Anonymous  Subject: Note  Date: 2016-07-29 18:27:42

define in caption!

Number: 3  Author: Anonymous  Subject: Note  Date: 2016-07-29 18:27:19

typo!

Number: 4  Author: Anonymous  Subject: Note  Date: 2016-07-29 18:28:53

Group number ? define?!

Number: 5  Author: Anonymous  Subject: Note  Date: 2016-07-29 18:28:30

NA? define

[Figure]

[Figure]

**Table 1. cont Rainfall-runoff events summary at RBF**

| Grp No. | Date | Rain dur'n (hr) | Total rainfall (mm) | Peak hourly rainfall intensity (mm/hr) | ASI (mm) | ASI+ rainfall (mm) | Runoff duration (hr) | Total runoff (mm) | Quick flow (mm) | RC (%) | Sat'd area (%) |
|---|---|---|---|---|---|---|---|---|---|---|---|
| 3 | 4/02/2011 | 33 | 71.2 | 16.4 | 130 | 201 | 10 | 2.5 | 2 | 3 | na |
| 2 | 11/05/2011 | 122 | 63.6 | 6.6 | 220 | 284 | 47 | 5.6 | 2.2 | 3 | na |
| 2 | 22/05/2011 | 50 | 21.6 | 4.4 | 231 | 253 | 3 | 0.23 | 0.1 | 0.46 | na |
| 2 | 26/05/2011 | 37 | 13.6 | 1.6 | 244 | 258 | 33 | 2.66 | 1 | 7 | na |
| 2 | 7/06/2011 | 25 | 17.6 | 2.6 | 254 | 272 | 43 | 6.7 | 2.9 | 16 | Na |
| 2 | 17/06/2011 | 31 | 15.6 | 2.2 | 248 | 264 | 45 | 3.2 | 0.2 | 1 | 4.6 |
| 2 | 21/06/2011 | 61 | 38.8 | 3.6 | 255 | 294 | 80 | 21.8 | 12.1 | 31 | 4.6 |
| 2 | 5/07/2011 | 10 | 9.8 | 2.4 | 253 | 263 | 26 | 2.9 | 0.8 | 8 | 4.7 |
| 2 | 6/07/2011 | 12 | 16.6 | 6.2 | 267 | 284 | 29 | 8.7 | 4.8 | 29 | 4.7 |
| 2 | 10/07/2011 | 20 | 7.4 | 2.2 | 262 | 269 | 27 | 3.7 | 1.4 | 19 | 4.7 |
| 2 | 28/09/2011 | 74 | 72.0 | 8.8 | 204 | 276 | 96 | 20.1 | 6.1 | 8 | 5.4 |
| 2 | 9/10/2011 | 54 | 23.2 | 2.4 | 232 | 255 | 5 | 0.35 | 0.14 | 1 | na |
| 2 | 28/10/2011 | 19 | 30.2 | 7.8 | 213 | 243 | 6 | 0.92 | 0.6 | 2 | na |
| 3 | 8/11/2011 | 12 | 35.2 | 14.8 | 222 | 257 | 31 | 10 | 7.1 | 20 | 5.2 |
| 2 | 9/11/2011 | 14 | 23.8 | 6.8 | 239 | 263 | 44 | 15.7 | 8.9 | 37 | 5.2 |
| 2 | 18/11/2011 | 30 | 45.5 | 10.2 | 219 | 265 | 82 | 17.7 | 8.9 | 20 | 5.5 |
| 2 | 26/11/2011 | 29 | 36.8 | 8.4 | 216 | 253 | 85 | 27.6 | 16.3 | 44 | 5.5 |
| 2 | 29/11/2011 | 47 | 19.4 | 4 | 233 | 252 | 70 | 11.9 | 0.9 | 5 | 5.5 |
| 3 | 10/12/2011 | 16 | 52.6 | 31 | 192 | 245 | 32 | 41.1 | 35.6 | 68 | 5.5 |

No Comments.

[Figure]

[Figure]

**Table 2. Rainfall events summary at RBF-no flow events**

| Grp No. | Date | Rain dur'n (hr) | Total rainfall (mm) | Peak hourly rainfall intensity (mm/hr) | ASI (mm) | ASI+ rainfall (mm) | Runoff duration (hr) | Total runoff (mm) |
|---|---|---|---|---|---|---|---|---|
| 1 | 12/10/10 | 5 | 5 | 3 | 227 | 232 | 0 | 0 |
| 1 | 12/10/2010 | 5 | 5 | 2.6 | 227 | 232 | 0 | 0 |
| 1 | 12/11/2010 | 32 | 28.2 | 4.0 | 184 | 212 | 0 | 0 |
| 1 | 25/11/2010 | 17 | 13 | 3.6 | 157 | 170 | 0 | 0 |
| 1 | 2/12/2010 | 8 | 7.2 | 2.0 | 191 | 198 | 0 | 0 |
| 1 | 8/12/2010 | 11 | 19.2 | 7.8 | 175 | 194 | 0 | 0 |
| 1 | 9/12/2010 | 6 | 5.2 | 2.4 | 193 | 198 | 0 | 0 |
| 1 | 17/12/2010 | 21 | 15 | 2.4 | 181 | 196 | 0 | 0 |
| 1 | 10/01/2011 | 18 | 10 | 3.2 | 146 | 156 | 0 | 0 |
| 1 | 11/01/2011 | 26 | 10.4 | 2.4 | 152 | 162 | 0 | 0 |
| 1 | 13/01/2011 | 26 | 22.2 | 9.0 | 151 | 173 | 0 | 0 |
| 1 | 25/01/2011 | 9 | 5.4 | 1.6 | 147 | 152 | 0 | 0 |
| 1 | 5/02/2011 | 12 | 8.6 | 1.6 | 182 | 191 | 0 | 0 |
| 1 | 16/02/2011 | 20 | 14.8 | 9.6 | 167 | 182 | 0 | 0 |
| 1 | 26/02/2011 | 13 | 13.4 | 4.2 | 174 | 187 | 0 | 0 |
| 1 | 21/04/2011 | 28 | 8.4 | 2.6 | 213 | 221 | 0 | 0 |
| 1 | 1/05/2011 | 8 | 10.8 | 2.6 | 207 | 218 | 0 | 0 |
| 1 | 8/05/2011 | 14 | 7.2 | 4.8 | 215 | 222 | 0 | 0 |
| 1 | 5/06/2011 | 18 | 9.4 | 3.0 | 238 | 247 | 0 | 0 |
| 1 | 9/09/2011 | 22 | 12.8 | 3.0 | 221 | 234 | 0 | 0 |
| 1 | 19/09/2011 | 18 | 12.2 | 4.0 | 218 | 230 | 0 | 0 |
| 1 | 24/10/2011 | 12 | 15.6 | 4.2 | 211 | 227 | 0 | 0 |

No Comments.

[Figure]

[Figure]

[Figure]

**Figure 1: The role of flux and timescale thresholds in determining runoff processes.**

Number: 1  Author: Anonymous  Subject: Note  Date: 2016-07-29 16:54:07

1) I agree with your decision tree but in fact it results in one dominant runoff mechanisms and your paper shows that these processes co-exist in parallel in different

2) This view neglects the contributions form upslope or neighboring sites.

3) it is also not clear if you consider the hillslope-scale or the point-scale. Typically these schemes are used to characterize the point- or plot-scale.

4) in box #3: do you mean surface topography?  Bedrock topography might be quite important (i.e. large storage to prevent still)

5) box #5 has no "NO" option!

[Figure]

[Figure]

[Figure]

**Figure 2.** The study site location within Australia and a hillshaded DEM, topography and sampling site locations at RBF.

Number: 1  Author: Anonymous  Subject: Note  Date: 2016-07-29 18:14:39

Can you destinguish the groundwater, streamflow, soil moisture, etc. sites with different symbols!

[Figure]

**Figure 3.** **(A) Weekly rainfall and APET time series (data from AWS), (B) soil water storage in top 60 cm of the soil profile and weekly runoff time series at RBF.**

**Number: 1  Author: Anonymous  Subject: Note  Date: 2016-07-29 18:40:42**

important: Is this one site, n average over how many sites?

**Number: 2  Author: Anonymous  Subject: Note  Date: 2016-07-29 18:42:28**

please give full names in captions so that they can stand alone. This applies to all figures and tabes!

[Figure]

[Figure]

Figure 4. Overview of rainfall-runoff event characteristics and runoff behaviour at RBF

**Number: 1  Author: Anonymous  Subject: Note  Date: 2016-07-29 18:49:17**

why green?

**Number: 2  Author: Anonymous  Subject: Note  Date: 2016-07-29 18:56:10**

can you mark these events in Fig.3B!
and
maybe make the rows of these events bold in table 1 and 2

**Number: 3  Author: Anonymous  Subject: Note  Date: 2016-07-29 18:49:51**

include info on the streamflow as well

**Number: 4  Author: Anonymous  Subject: Note  Date: 2016-07-29 18:51:09**

8 selected rainfall events (sorted by their Antecedent Soil moisture Index (ASI), riainfall shown in blue ...

**Number: 5  Author: Anonymous  Subject: Note  Date: 2016-07-29 18:51:55**

make the reader aware, tha the y-axis scale differ between events.

[Figure]

Figure 5. Thresholds of runoff mechanisms at RBF, a) event rainfall versus total event runoff, colours indicate the highest hourly rainfall intensity, the impact of five factors together including: cumulative curve of the distribution of soil water storage as observed through the study period, ASI, ASI+rain, colour shows the peak hourly rainfall intensity ($I_{peak}$) and the size of the bubbles shows the quick flow runoff coefficient, c) ASI versus the peak hourly rainfall intensity ($I_{peak}$) and the size of the bubbles shows the quick flow runoff coefficient and colour shows event total runoff, and d) ASI+rain versus the peak hourly rainfall intensity ($I_{peak}$) and the size of the bubbles shows the quick flow runoff coefficient and colour shows event total runoff.

**Number: 1  Author: Anonymous  Subject: Note  Date: 2016-07-29 19:02:46**

is this cumulative ASI? (be specific)

**Number: 2  Author: Anonymous  Subject: Note  Date: 2016-07-29 19:01:31**

please put (a) in to the upper left corner

**Number: 3  Author: Anonymous  Subject: Note  Date: 2016-07-29 19:28:32**

Rainfall Intensity (I guess)?

**Number: 4  Author: Anonymous  Subject: Note  Date: 2016-07-30 10:35:23**

You are actually not showing thresholds in these figures

[Figure]

[Figure]

[Figure]

**Figure 6. Time series of runoff and water levels at sites 4, 5, 3, 32, 2 and 1, manually read sites are shown with dashed lines.**

Number: 1  Author: Anonymous  Subject: Note  Date: 2016-07-30 10:39:22

groundwater level ...

Number: 2  Author: Anonymous  Subject: Note  Date: 2016-07-30 10:39:08

... runoff (grey line) ...

[Figure]

**Figure 7. Soil water storage versus water table level at sites 4 and 5. Colours distinguish hourly rainfall intensity.**

**Number: 1  Author: Anonymous  Subject: Note  Date: 2016-07-30 10:53:46**

You need to define "typical" and "high intensity" in the method section and in the captions

**Number: 2  Author: Anonymous  Subject: Note  Date: 2016-07-30 10:54:56**

How did you calculate soil water storage? Average across the catchment? Please state int he method section.

**Number: 3  Author: Anonymous  Subject: Note  Date: 2016-07-30 11:20:45**

Please be consistent with using your terms. I guess soil water storage is volumetric water content of the upper 60 cm of the soil profile), or can you define soil water storage in the method section.

[Figure]

[Figure]

**Figure 8. Recession constant ($K$) and soil water storage the start of the recession for individual events**

Number: 1  Author: Anonymous  Subject: Note  Date: 2016-07-30 11:31:05

Important: Which events did you chose? Why not all events? Please give objective reasosn how you selected these events.

[Figure]

[Figure]

**Figure 9. Time series of total rainfall and runoff, $^2$H of rainfall and runoff for the event on 12/8/10**

No Comments.

[Figure]

[Figure]

[Figure]

**Figure 10. a) Runoff versus Cl⁻ concentration for all events. The red colour identifies samples from event on 8/11/11, and b) Time series of rainfall, runoff and Cl⁻ concentration for the event on 8/11/11**

No Comments.

---

## Author Comment (AC1) · 7 Sep 2016

**Authors' response to Anonymous Referee #1 on "Multiple runoff processes and multiple thresholds control agricultural runoff generation" by S. Saffarpour et al.**

We appreciate the reviewer's comments and suggestions, which we found very useful. We have addressed each reviewer comment separately. In the following document the blue font indicates the reviewer's comment and the black font shows our reply.

**General comment**

This is a very interesting work that focuses on the analysis of runoff processes and the controls exerted by different thresholds on the hydrological mechanisms related to runoff generation in an agricultural Australian catchment. The research aims to understand how subsurface connectivity, saturation excess and rainfall intensity play a role in rainfall-runoff response at the seasonal and event time scale. The manuscript is well written, logically organized and with overall clear graphical presentations. Results are generally well supported by data and interpretation are overall sound. I particularly like the conceptual summary of hydrological processes and thresholds reported in Fig 1, and how this figure was referred to in the Introduction and in the discussion. However, I think that there are some confused points that deserve to be clarified and better explained. I have some comments and suggestions that can hopefully help this paper to have a greater impact on the hydrological community.

Thank you for these positive comments.

**Specific comments**

5, 24-26. As far as I understand, two different isotope laser analysers have been employed for the analysis of stable isotopes of water. This is a methodologically critical point: based on my experience, two different laser machines, even of the same model and calibrated using the same set of reference standards, could return quite different values of isotopic composition. Using a different sets of standards in different laboratories, as it seems that was the case here, could lead to differences that have the potentials to impact the resulting analysis of hydrograph separation. I think it is important for the paper to run some tests and report some comparison metrics between the measurements performed by the two machines in order to assess, and in case correct, potential deviations.

It is true that the two laser analyzers were systematically different, which we established by analyzing identical samples to both laboratories and should have discussed in the original paper. We developed and applied a correction between the two laboratories based on these samples. In the analyses presented here we actually only used samples analyzed at the Monash University laboratory and hence this difference between laboratories does not affect our results. We will edit the methods to reflect this.

6, 10-12. It is not clear how the soil water storage has been computed starting from ASI. A specification, perhaps including equations, would be really useful here. This is important because the soil water storage is addressed several times in the rest of the manuscript.

Fractional volumetric water content (VWC) was measured for the 0-30 cm and 30-60 cm layers using vertically installed 30 cm long Campbell Scientific (CS625) probes (Campbell Scientific, 2006), which were recorded hourly by the automatic weather station (AWS) logger. The VWC was temperature corrected using measured soil temperature and the manufacturers recommended temperature correction, as follows (Campbell Scientific, 2006).

 $\begin{aligned} \tau_{\text{corrected}} \left( T_{\text{soil}} \right) &= \tau_{\text{uncorrected}} \\ &+ \left( 20 - T_{\text{soil}} \right) * \left( 0.526 - 0.052 * \tau_{\text{uncorrected}} + 0.00136 * {\tau_{\text{uncorrected}}}^2 \right) \end{aligned}$

where  $\tau_{uncorrectd}$  is the probe output period,  $T_{soil}$  is the soil temperature and  $\tau_{corrected}$  is the corrected probe output. The VWC was then computed from  $\tau_{corrected}$  using (Campbell Scientific, 2006):

 $VWC = -0.0663 - 0.0063 * period + 0.0007 * period^{2}$

Soil water storage over the top 60 cm soil depth was computed by adding the VWC from the 0-30 cm and 30-60 cm layers and then multiplying by the 300 mm soil layer thickness.

Moreover, it's not clear how manual measurements of the saturated are have been carried out. Please explain.

Measurement of the saturated area is explained briefly in section 2.4. We will expand that explanation so that it is clearer. Specifically: The lateral boundary of the saturated area is constrained and field observations suggested it was stable over time, while the upstream boundary moved up and down the riparian zone. The saturated area was estimated by locating the upper boundary through field inspection and then measuring the distance from either well 2 or 3. The saturated area was then estimated from this information combined with the mapping of the riparian zone boundary (see Figure 2). These measurements were made between events.

8, 2. The statement that 'any rainfall depth could produce a response' seems to contradict what reported elsewhere in the manuscript (eg, 7, 4-5; 8, 14; 8, 26) about some rainfalls that did not produce runoff. This is confusing and should be clarified. 8, 6-16.

We will reword this sentence for clarity to "Acknowledging that we have excluded rainfall events below 5 mm total rainfall, essentially any rainfall depth could produce a response at the catchment outlet, provided the catchment is sufficiently wet. There was a wide variation in runoff coefficients (indicated by the scatter)."

Fig. 5b seems to be dense and informative. However, I think that is not straightforward to understand it. The different symbols are hard to distinguish and the scale of runoff coefficients is not very useful to understand their values. I suggest considering to replace it by another graphical way (eg, cumulative distribution + bar plot or multiple panel boxplot). Moreover, I don't understand why ASI and ASI+rain have been plotted against rainfall intensity: the relation is obviously scattered (and so the sentence at 8, 12 is obvious too since rainfall is a stochastic process) because no relation is expected between these two variables and intensity of rainfall events. But if the authors used this representation to show how the different events plot in reference to these variable this should be clearly stated.

Figures 5b-d are used to examine thresholds leading to different rainfall-runoff behaviors. The reason for plotting ASI + rain against rainfall intensity is that runoff shows threshold behavior with respect to these two variables, not because we expect a relationship between the two variables. We will add the thresholds in both catchment wetness and rainfall intensity to these figures to make the interpretation clearer. A revised version of Figure 5 is below. A joint threshold involving intensity and water storage that correctly separates nearly all events into those that produce runoff and those that do not is also shown. This will be discussed in the first paragraph of section 3.3 where we mention interaction between intensity and wetness.

Figure 5. Thresholds of runoff mechanisms at RBF, a) event rainfall versus total event runoff, colours indicate the highest hourly rainfall intensity, b) the impact of five factors together including: cumulative curve of the distribution of soil water

storage as observed through the study period, ASI, ASI+rain, colour shows the peak hourly rainfall intensity ( $I_{peak}$ ) and the size of the bubbles shows the quick flow runoff coefficient, c) ASI versus the peak hourly rainfall intensity ( $I_{peak}$ ) and the size of the bubbles shows the quick flow runoff coefficient and colour shows event total runoff, and d) ASI+rain versus the peak hourly rainfall intensity ( $I_{peak}$ ) and the size of the bubbles shows the quick flow runoff coefficient and colour shows event total runoff, and d) ASI+rain versus the peak hourly rainfall intensity ( $I_{peak}$ ) and the size of the bubbles shows the quick flow runoff coefficient and colour shows event total runoff. The dashed lines in b), c) and d) show thresholds at ASI and ASI+rain = 250mm and  $I_{peak} = 15$ mm/h. The dot-dashed line in d) shows a threshold of  $260 - 3/11*I_{peal}$ .

10, 25-26. Here a two-component mixing model is mentioned but no details are given in the Materials and Method section. I'm not suggesting to report the well-known equations (a simple citation to the suggested references 5 and 6 below is enough) but some methodological/conceptual information are needed, eg: which sample(s) has been considered as pre-event for the application of the hydrograph separation technique? Why only deuterium data have been used since both 18-oxygen and deuterium data have been measured? How many samples for isotopes have been collected and which ones were used? How many events have been sampled? More importantly: why has the separation been carried out only for the 12 August event showed in Fig. 9? Or was it also performed for other events? In this case, what are the results? Are they similar so that they corroborate the proposed conceptualization? Or did they provide much different estimates? Can the author report the results of all sampled events in a Table? This would be useful. All this information should be reported and these points well addressed in the revised version of the manuscript.

Another major point related to this is the lack of uncertainty analysis of the estimated fractions of pre-event water and event water (I prefer these terms instead of old and new water) in streamflow. This is particularly critical since these estimates have been used to build some conceptualization (eg, 17% of event water corresponding to 5% of rainfall amount..but what is the uncertainty of that 17%?). And is the result about 5% of rainfall based only on the 12 August event? In that case this is not robust. The traditional method of uncertainty estimation proposed in reference 2 below is suggested.

We will update the methods so that the above questions are clarified, including citing the well- known one tracer, two component model of hydrograph separation approach (Pinder and Jones, 1969; Sklash and Farvolden, 1979) and clarifying how each end-member was identified and the overall estimation of event and pre-event water volume. We will also report uncertainties following Genereux (1998).

In terms of events considered, we collected multiple stream water samples from five events. Two events were missing rainfall samples and for two events the rainfall isotope signature was quite close to the typical low flow signature. Uncertainty estimates showed standard deviations for pre-event and event fractions being over 50% for these two events. For the event on 12/8/2010, the standard deviations were approximately 12% and 9% for 18O and D respectively. Given that the event on 12/8/2010 was the only one for which we had a clear differentiation between rainfall and streamflow isotope signatures, we have just analysed that event. We will modify the methods to indicate that only a selection of events were sampled for isotopes and that only one analyzable event was sampled.

We have repeated the analysis using the O18 data and changed the method to estimate the rainfall end member slightly so that the mean of the rainfall samples during the event is used (rather than using the

rainfall sample corresponding to the timing of the stream sample). We used a pre-event low-flow sample from ~2days before the analysed event as the pre-event end member. To estimate the overall event water contribution we interpolated the fraction of event water between stream water sampling times and combined this with the flow hydrograph to calculate the overall volume of event water. In the uncertainty analysis the pre-event end member standard deviation was estimated as the low flow sample standard deviation across the study period and the standard deviation of rainfall samples for the event was used for the event water end member standard deviation. The stream sample standard deviation was taken from the analysis precision as reported by the laboratory. This analysis suggests that the percentage of rain becoming runoff is 4.4% (3.4-5.4%) and 3.6% (3.0-4.2%) using 18O and D respectively, where the figures in the brackets are 95% confidence intervals for uncertainty based on the hydrograph separation only.

11, 1-11. It is mentioned that the concentration of major ions is available for the 8 November event but only chloride has been selected and showed (Fig. 10). What is the reason behind this choice? Moreover, where does the estimate of 5% of the rainfall come from, that agrees surprisingly well with the estimate of the 12 August event (10, 26)? From a two-component hydrograph separation based on chloride? On isotopes? Please, explain in detail.

We will add figures (see below) showing Calcium, Magnesium, Sodium and Potassium to a supplement. The reason for choosing Chloride was that we expect less complication due to any ion exchange processes. Sodium and Magnesium show similar behavior to Chloride. Potassium also shows somewhat anomalous behavior but with higher concentrations. Calcium does not show anomalous behavior. The 5% was estimated directly from the hydrograph. Up until 0600 there had been 23.4mm of rainfall and 1.0mm of runoff. This is 4.3%. We will explain this more precisely in the paper.

---

## Author Response (AR1)

**Authors' response to Anonymous Referee #1 on "Multiple runoff processes and multiple thresholds control agricultural runoff generation" by S. Saffarpour et al.**

We appreciate the reviewer's comments and suggestions, which we found very useful. We have addressed each reviewer comment separately. In the following document the blue font indicates the reviewer's comment and the black font shows our reply.

**General comment**

1. **This is a very interesting work that focuses on the analysis of runoff processes and the controls exerted by different thresholds on the hydrological mechanisms related to runoff generation in an agricultural Australian catchment. The research aims to understand how subsurface connectivity, saturation excess and rainfall intensity play a role in rainfall-runoff response at the seasonal and event time scale. The manuscript is well written, logically organized and with overall clear graphical presentations. Results are generally well supported by data and interpretation are overall sound. I particularly like the conceptual summary of hydrological processes and thresholds reported in Fig 1, and how this figure was referred to in the Introduction and in the discussion. However, I think that there are some confused points that deserve to be clarified and better explained. I have some comments and suggestions that can hopefully help this paper to have a greater impact on the hydrological community.**

Thank you for these positive comments.

**Specific comments**

2. **5, 24-26. As far as I understand, two different isotope laser analysers have been employed for the analysis of stable isotopes of water. This is a methodologically critical point: based on my experience, two different laser machines, even of the same model and calibrated using the same set of reference standards, could return quite different values of isotopic composition. Using a different sets of standards in different laboratories, as it seems that was the case here, could lead to differences that have the potentials to impact the resulting analysis of hydrograph separation. I think it is important for the paper to run some tests and report some comparison metrics between the measurements performed by the two machines in order to assess, and in case correct, potential deviations.**

We have added the following to the methods (p8, l3) section to clarify this issue.

Stable isotope ratios were measured either at Monash University using a Finnigan MAT 252 and ThermoFinnigan Delta Advantage Plus mass spectrometers (2010 samples) or at the University of Melbourne, where a Picarro L2120i cavity ring-down isotope analyser was used to determine isotope ratios (2011 samples). The instrumental precision was $\delta^{18}O = \pm0.15‰$ and $\delta^{2}H = \pm1‰$ for the isotope samples analysed at the Monash University and it was $\delta^{18}O=0.1‰$ and $\delta^{2}H =0.4‰$ for samples

analysed at the Melbourne University. The results from the two laser analysers were checked for systematic differences, by analyzing duplicate samples in both laboratories. We developed and applied a correction between the two laboratories based on these samples. With the exception of determining pre-event end member uncertainty (described later) the analyses presented here only use samples analysed at the Monash laboratory and hence this difference between laboratories only affects our hydrograph separation uncertainty estimations.

3. **6, 10-12. It is not clear how the soil water storage has been computed starting from ASI. A specification, perhaps including equations, would be really useful here. This is important because the soil water storage is addressed several times in the rest of the manuscript.**

The following explanation was added at p6, l26.

Soil moisture storage was calculated from the volumetric water content which was measured for the 0-30 cm and 30-60 cm layers and recorded hourly by the automatic weather station (AWS) logger using two vertically installed 30 cm long Campbell Scientific (CS625) soil moisture probes (Campbell Scientific, 2006).. The 0-30cm probe was inserted from the surface.  To install the 30-60cm probe, a 30cm deep hole was dug, with the excavated soil set aside.  The probe was inserted vertically and then the excavated soil was repacked so that the soil was replaced at a similar depth to that from which had been removed.

The CS625 produces a pulse signal and the pulse period was temperature corrected using measured soil temperature and the manufacturers recommended temperature correction, as follows (Campbell Scientific, 2006).

$$\tau_{corrected}(T_{soil}) = \tau_{uncorrected}$$
$$+ (20 - T_{soil}) * (0.526 - 0.052 * \tau_{uncorrected} + 0.00136 * \tau_{uncorrected}^2)$$
(1)

where $\tau_{uncorrected}$ is the probe output period, $T_{soil}$ is the soil temperature and $\tau_{corrected}$ is the corrected probe output. The *VWC* was then computed from $\tau_{corrected}$ using (Campbell Scientific, 2006):

$$VWC = -0.0663 - 0.0063 * period + 0.0007 * period^2$$
(2)

Soil water storage over the top 60 cm soil depth was computed by adding the VWC from the 0-30 cm and 30-60 cm layers and then multiplying by the 300 mm soil depth.  We use this soil water storage at the start of each rainfall event as an index of antecedent soil water (ASI) and assume it represents the catchment wetness condition.

4. **Moreover, it's not clear how manual measurements of the saturated are have been carried out. Please explain.**

Measurement of the saturated area is now explained in section 2.4 (p9, L12) as follows.

The topography of the catchment was surveyed using a differential Global Positioning System (dGPS) and a 1 m horizontal resolution DEM (digital elevation model) was developed by interpolation methods. Instrument and well locations were also determined using dGPS. Then this data was used to produce maps of the study area and including sampling sites using Arc GIS. The lateral boundary of the saturated area was topographically constrained and field observations suggested it was stable over time, while the upstream boundary moved up and down the riparian zone. The saturated area was estimated by locating the upper boundary through field inspection and then measuring the distance from either well 2 or 3 (The saturated area extended towards site 3 during very wet conditions). Figure 2 demonstrates the approximate maximum boundary location of the saturated area at site 3 and the main stream located close to the outlet of the catchment. The saturated area was then estimated from this information combined with the mapping of the riparian zone boundary in Arc GIS.

5. **8, 2. The statement that 'any rainfall depth could produce a response' seems to contradict what reported elsewhere in the manuscript (eg, 7, 4-5; 8, 14; 8, 26) about some rainfalls that did not produce runoff. This is confusing and should be clarified. 8, 6-16.**

We have removed this sentence.

6. **Fig. 5b seems to be dense and informative. However, I think that is not straightforward to understand it. The different symbols are hard to distinguish and the scale of runoff coefficients is not very useful to understand their values. I suggest considering to replace it by another graphical way (eg, cumulative distribution + bar plot or multiple panel boxplot). Moreover, I don't understand why ASI and ASI+rain have been plotted against rainfall intensity: the relation is obviously scattered (and so the sentence at 8, 12 is obvious too since rainfall is a stochastic process) because no relation is expected between these two variables and intensity of rainfall events. But if the authors used this representation to show how the different events plot in reference to these variable this should be clearly stated.**

Figures 5a-d are used to examine various influences and thresholds leading to different rainfall-runoff behaviors. The reason for plotting ASI + rain against rainfall intensity is that runoff shows threshold behavior with respect to these two variables, not because we expect a relationship between the two variables. We have added the thresholds in both catchment wetness and rainfall intensity to these figures to make the interpretation clearer. We have also edited section 3.2 to more carefully guide the reader through the interpretation of these figures. A revised version of Figure 5 is below. A joint threshold involving intensity and water storage that correctly separates nearly all events into those that produce runoff and those that do not is also shown. This is discussed in the first paragraph of section 3.3 where we mention interaction between intensity and wetness.

[Figure]

**Figure 5. Thresholds of runoff mechanisms at RBF, a) event rainfall versus total event runoff, colours indicate the highest hourly rainfall intensity, b) the impact of five factors together including: cumulative curve of the distribution of soil water storage as observed through the study period, ASI, ASI+rain, colour shows the peak hourly rainfall intensity ($I_{peak}$) and the size of the bubbles shows the quick flow runoff coefficient, c) ASI versus the peak hourly rainfall intensity ($I_{peak}$) and the size of the bubbles shows the quick flow runoff coefficient and colour shows event total runoff, and d) ASI+rain versus the peak hourly rainfall intensity ($I_{peak}$) and the size of the bubbles shows the quick flow runoff coefficient and colour shows event total runoff. The dashed lines in b), c) and d) show thresholds at ASI and ASI+rain = 250mm and $I_{peak}$ = 15mm/h. The dot-dashed line in d) shows a threshold of $260 - 3/11 * I_{peal}$.**

7. **10, 25-26. Here a two-component mixing model is mentioned but no details are given in the Materials and Method section. I'm not suggesting to report the well-known equations (a simple citation to the suggested references 5 and 6 below is enough) but some methodological/conceptual information are needed, eg: which sample(s) has been considered as pre-event for the application of the hydrograph separation technique? Why only deuterium data have been used since both 18-oxygen and deuterium data have been measured? How many samples for isotopes have been collected and which ones were used? How many events have been sampled? More importantly: why has the separation been carried out only for the 12 August event showed in Fig. 9? Or was it also performed for other events? In this case,**

**what are the results? Are they similar so that they corroborate the proposed conceptualization? Or did they provide much different estimates? Can the author report the results of all sampled events in a Table? This would be useful. All this information should be reported and these points well addressed in the revised version of the manuscript.**

**Another major point related to this is the lack of uncertainty analysis of the estimated fractions of pre-event water and event water (I prefer these terms instead of old and new water) in streamflow. This is particularly critical since these estimates have been used to build some conceptualization (eg, 17% of event water corresponding to 5% of rainfall amount..but what is the uncertainty of that 17%?). And is the result about 5% of rainfall based only on the 12 August event? In that case this is not robust. The traditional method of uncertainty estimation proposed in reference 2 below is suggested.**

We have added uncertainty analysis, analysed by the 2H and 18O results and provided more detail on the methodology.  The following was added to the hydrograph separation section (p9, L32):

Isotope samples are used here for hydrograph separation into pre-event and event contributions.  We applied the well- known one tracer, two component model of hydrograph separation approach (Pinder and Jones, 1969; Sklash and Farvolden, 1979). We determined uncertainties following Genereux (1998). We undertook the analysis using both the 2H O18 data.  We used the mean of the rainfall samples during the event to estimate the event water end member and low flow samples from the few days to the event for the pre-event end member. In the uncertainty analysis, the standard deviations of the event rainfall samples and of all the low flow samples across the study period were used.  Half the analytic uncertainty was used to represent the standard deviation of the streamflow sample.

We added the number of isotope samples to page 8 line 5:

For isotope analysis, in total we collected 115 samples of rainfall, 28 samples during low flows and multiple stream water samples of isotopes from five events.

We have provided our reasons for presenting only one event (it was the only one for which we could obtain reliable results) and amended the presentation of results in section Isotope and major ion results (p16, L5)

Of the five events sampled, two events were missing rainfall samples, which prevented their analysis. For another two events the rainfall isotope signature was quite close to the typical low flow signature. Uncertainty estimates showed standard deviations for pre-event and event fractions being over 50% for these two events, therefore these events were also excluded. For the event on 12 August 2010, the standard deviations were approximately 12% and 9% for 18O and 2H respectively, which we judged to be acceptable.

We have also significantly edited the relevant results paragraph (p16, L11) to:

Figure 9 shows the event from 12 August 2010 during the wettest part of the study period. The antecedent soil water storage at the beginning of this event was 274 mm and total rainfall was 17 mm.

We used a pre-event low-flow sample from ~2days before the analysed event as the pre-event end member. We estimated the contribution at the time of each stream flow sample. Over the study period $\delta^2H$ ($\delta^{18}O$) for rainfall varied between -7‰ and -83‰ (-1.4‰ and 12.6‰) and isotopic concentrations for low flows were highly damped. Low flow samples from the RBF flume before and after the event showed a $\delta^2H$ ($\delta^{18}O$) of -27‰ (-4.1‰) and rainfall for this event was strongly depleted (3 samples prior to and during the event $\delta^2H$ = -42, -67 and -57‰, $\delta^{18}O$ = -6.5, -9.8 and -8.2‰), compared with low flow. The runoff samples showed a very different isotopic concentration during the rising limb and the peak of the hydrograph ($\delta^2H$ =-43‰, $\delta^{18}O$ = -6.8‰) in comparison to antecedent low flow. This shows that the isotopic concentration moves significantly towards the rainfall sample concentrations. To estimate the overall event water contribution we first separated the hydrograph at each sampling time and then interpolated the fraction of event water between stream water sampling times and combined this with the flow hydrograph to calculate the overall volume of event water. This analysis suggests that the percentage of rain becoming runoff based on $\delta^2H$ and $\delta^{18}O$ is 4.4% (3.4-5.4%) and 3.6% (3.0-4.2%), respectively. The figures in the brackets are 95% confidence intervals for uncertainty based on the hydrograph separation uncertainty only. These results suggest that precipitation on the saturated area generates direct runoff in amounts that are close to what would be expected (i.e. 100% runoff) given that the saturated area is around 5-6% of the catchment area.

8. **11, 1-11. It is mentioned that the concentration of major ions is available for the 8 November event but only chloride has been selected and showed (Fig. 10). What is the reason behind this choice? Moreover, where does the estimate of 5% of the rainfall come from, that agrees surprisingly well with the estimate of the 12 August event (10, 26)? From a two-component hydrograph separation based on chloride? On isotopes? Please, explain in detail.**

We have also examined the behavior of Calcium, Magnesium, Sodium and Potassium and now provide these plots in a supplement. The reason for choosing Chloride was that we expect less complication due to any ion exchange processes. We have amended the discussion of the chemistry by adding the following to p16 L33.

Similar plots are shown for Sodium, Magnesium, Calcium and Potassium in the supplementary material Figure S1. Sodium and Magnesium show similar behavior to Chloride. Potassium also shows somewhat anomalous behaviour but with anomalously high concentrations. The relationship between flow and $K^+$ is also different to $Cl^-$, $Na^+$ and $Mg^{2+}$ in that concentrations increase slightly as flow increase above about 0.2mm/mm, rather than a decline. Calcium does not show anomalous behavior.

We have also explained how we obtained the 5% proportion for the first peak on p17, L7.

Up until the end of the first flow peak (i.e. 0600), there had been 23.4mm of rainfall and 1.0mm of runoff. This runoff volume is 4.3% of the rainfall volume, which is similar to the proportion of event water that became runoff for the 12 August 2010 event discussed above.

[Figure]

9. **11, 19. The saturation amount at the 5% of the catchment area is not shown and clearly presented, yet it is one of the most interesting results in terms of process interpretation. Please, provide a sound explanation.**

The estimation of the saturated area is explained in response to comment 4 above.

10. **12, 29. I do not see such a clear threshold at 250 mm of ASI + rain. . .please, explain better, also in the results, where it derives from.**

As explained in our response to comment 6, we have shown the thresholds on Figure 5 and tried to explain these more clearly.

11. **Some relevant studies that I'm aware of and that are strictly linked to this research have not been cited. I think they should incorporated in the paper, particularly in the Discussion section (except the ones referring to methods, such as 2, 5 and 6):**

   **1. Fu C, Cheng J, Jiang H, Dong L. 2013. Threshold behavior in a fissured granitic catchment in southern China: (1) analysis of field monitoring results. Water Resources Research 49: 1–17. DOI: 10.1002/wrcr.20191**

Added to P 2 L 15 and the reference list

   **2. Genereux D. 1998. Quantifying uncertainty in tracer-based hydrograph separations. Water Resources Research 34(4): 915–919. DOI: 10.1029/98WR00010**

Added to P10 L1 and the reference list

   **3. Penna, D., van Meerveld, H.J., Oliviero, O., Zuecco, G., Assendelft, R.S., Dalla Fontana, G., Borga, M., 2015. Seasonal changes in runoff generation in a small forested mountain catchment. Hydrological Processes 29, 2027–2042. doi:10.1002/hyp.10347**

Added to P 2 L 15 and the reference list

   **4. Penna, D., van Meerveld, H.J., Zuecco, G., Dalla Fontana, G., Borga, M., 2016. Hydrological response of an Alpine catchment to rainfall and snowmelt events. Journal of Hydrology 537, 382–397. doi:10.1016/j.jhydrol.2016.03.040**

Added to P 17 L 27 and the reference list

   **5. Pinder, G.F., Jones, J.F., 1969. Determination of ground-water com-ponent of peak discharge from chemistry of total runoff. Water Resour. Res. 5 (2), 438–445. http://dx.doi.org/10.1029/WR005i002p00438.**

Added to P9 L32 and the reference list

6. Sklash MG, Farvolden RN. 1979. Role of groundwater in storm runoff. Journal of Hydrology 43(1–4): 45–65. DOI: 10.1016/0022-1694(79)90164-1

Added to P9 L 32 and the reference list

**Minor comments and technical corrections**

12. **1, 10. The reference to individual research catchments makes the reader think that this paper focuses on the analysis of several catchments but this is not the case. I suggest to remove or reformulate.**

We have re-written this as "However, to date, most attention has focused on single runoff response types"

13. **2, 12. Here the suggested references 1 and 3 could be added.**

Added

14. **3, 3-4. This sentence is not totally clear. Please, explain.**

We have expanded this explanation (now at P3 L28) as follows.

It would be attractive to think of the problem of runoff response purely in terms of timescales of competing processes following Oldham et al. (2013), who used a generalise Damköhler to represent the competing effects of transport and reaction timescales on the loss of material along a flow path. When the reaction timescale is small compared with the transport timescale, the reactant is consumed before reaching the exit of the flow path it is moving along. A complication here is that, both flux and time thresholds are important. This arises because there is finite capacity for flow in various parts of the catchment system.

15. **3, 22. Typo.**

Fixed

16. **3, 27. 'certain processes': too vague. Reformulate.**

Reformulated to: "While Figure 1 suggests catchment rainfall-runoff response is dominated by speicific processes (e.g. saturation excess runoff) it needs to be recognised that many catchment conditions vary over time and space."

17. **3, 30. Here the suggested references 1 and 4 could be added.**

Added.

**18. 4, 9. It is a bit surprising to know that the study area is a hillslope after reading the Introduction that focuses almost exclusively on processes at the catchment scale!**

It is really a small catchment. We have changed our terminology throughout the paper.

**19. 4, 17. 'reasonably' is too vague. Specify.**

The sentence has been revised to (p5 l30):

The study area has a humid climate and rainfall is uniformly distributed across the year.

**20. 5, 26. This can be misunderstood as the uncertainty in the presented results of hydrograph separation. I think it's clearer to use the term 'instrumental precision'.**

This has been revised to: "The instrumental precision was …."

**21. 6, 15. Do the authors mean 'conceptually separate' here, ie they are considering these processes, and not physically computing the fractions of return flow and SOF in stream- flow? Please, reformulate for clarity.**

Done

**22. 7, 6. Better to use 'stream' or 'streamflow' here instead of 'runoff'.**

Done

**23. 7, 23. For the sake of clarity, indicate which events/panels.**

Done

**24. 8, 9. How was the quick flow runoff coefficient computed? In section 2.4 it was not defined. . .unless it's, as I think, the same than 'event runoff coefficient'. In the latter case, please be terminologically consistent.**

We will edit the paper so that we use event runoff coefficient consistently.

**25. 8, 28-9, 4. This part could be condensed by pointing out at the Tables.**

Done

**26. 10, 10. Although known and intuitive, the symbols of this equation should be explained. Moreover, it should be stated that the events falling into this period are 10 (as inferred from Fig. 8).**

We have revised this to (p15 L16):

To explore this, we calculated the recession constant, $k$ (as in $Q_t = Q_0 e^{-kt}$, where $Q_t$ is flow at time $t$ during the recession period, $Q_0$ is the flow at the beginning of the recession and $k$ is the recession constant), and plotted it against soil water storage at the start of the recession for individual events within this period (Figure 8). This period contained 10 events. $K$ decreased as ….

**27. 10, 17. 'Clearly'. I think it would be more cautious to start this sentence stating the results and/or the figures that point at this.**

We have amended this to begin as follows (p16 L2):

"The hydrometric results presented in Figures 6, 7 and 8 suggest that subsurface flow is important in this catchment.  Given this, we would expect the hydrograph to be dominated by "old" or pre-event water; however, the saturated area….."

**28. 10, 24. 'different signature': ok, but the trend is similar and should be remarked.**

We have reworded this to (p16 L17):

"The runoff samples showed a very different isotopic concentration during the rising limb and the peak of the hydrograph (δ2H =-43‰, δ18O = -6.8‰) in comparison to antecedent low flow. This shows that the isotopic concentration moves significantly towards the rainfall sample concentrations."

**29. 10, 25. Here the suggested references 5 and 6 could be added.**

The Hydrograph separation is now explained in the methods and these references are cited.

**30. 11, 20.  Please, explain what the 'field observations' are.**

This has been explained as follows (p17 l19):

Field observations of surface saturation extending to the flume and well hydrograph measurements of the water table at the surface showed that the saturated area was highly connected to the catchment outlet and suggest that it would be expected to produce saturation excess runoff.

**31. 11, 24. Here the suggested reference 4 could be added.**

Done

**32. 13, 7-13. This part is not very relevant to the observed results and could be skipped, in my opinion.**

We will delete this paragraph.

**33. Tables and Figures Table 1. Remove the first column, it's not useful. Don't use abbreviations in the column name.**

Column 1 is needed in Table 1 as it distinguishes between group 2 and group 3 events. We have removed the abreviations

34. **Fig. 1. The first 'Yes' on the top horizontal arrows should be moved more to the right close to the dashed arrow, in my opinion. And perhaps the second 'Yes' can be removed.**

We have amended the figure as suggested

35. **Fig. 2. I suggest the terms lower, mid and upper hillslope (or slope) instead. Remove the notation and the arrow pointing to the wells and put them in a legend. Why has the DEM been cut before the stream. . .cannot be extended to it?**

We will amend the figure as suggested

36. **Fig. 4. Replace 'overview' with 'example'. I also suggest to include a no-flow event.**

Done. A zero flow event has been included.

37. **Fig. 5. Please, see my comment above. Moreover, the difference between 'Soil Moisture Index' of panel b) and 'ASI' of panel c) is not clear and should be explained (or fixed if they are the same thing).**

**(I think 'soil moisture index' and 'ASI' are the same thing so we should change the title of X axis in panel b to 'ASI'.)**

Yes the soil moisture index in Figure 5 and ASI are the same thing. We have edited the figure so we use consistent labelling and checked that we are also consistent throughout the text.

38. **Fig. 6. Why are there values above zero? Explain or fix.**

The soils are highly pugged in this area and water pooled on the surface in places leading to slightly positive water levels being recorded at site 2. We have explained this in the figure caption as follows.

Figure 6. Time series of discharge (grey lines) and groundwater levels at sites 4, 5, 3, 32, 2 and 1, manually read sites are shown with dashed lines. Levels above zero at sites 1 and 2 occur as the soils are highly pugged in the riparian zone and water pooled on the surface in places leading to slightly positive water levels being measured at site 2 in particular.

39. **Fig. 7. Why have only these sites been shown and not also water table at the other locations? This should be explained in the text. Additionally, 'high intensity' is too vague and should quantified, possibly using thresholds presented in Fig. 5.**

We have explained on p14 l30 that these sites were chosen because they show significant dynamics and we had logged records available. Other sites with loggers had limited dynamics or short records.

We define high intensity in the caption (>15mm/h) and have amended the figure so this definition is consistent with the threshold in Figure 5.

> **40. Fig. 8. Add a mention to the period when these events have been selected. It would be interesting to see these results also for other events.**

The following sentence is added to the caption of the Figure 8:

These events happened in the very wet period during August/September 2010. We will consider adding other events.

> **41. Fig. 9. The symbol '‰should be put in parenthesis. ´**

Done

> **42. Fig. 10. Please be consistent with the use of terms such as 'discharge' (as here) or 'flow' (as in Fig. 9).**

We have edited the figures as suggested. We use discharge where instantaneous measurements are plotted and runoff where total volumes over some period are plotted. We have edited the paper for consistency.

*Authors' response to Anonymous Referee #2 on "Multiple runoff processes and multiple thresholds control agricultural runoff generation" by S. Saffarpour et al.*

We appreciate these useful reviewer comments and suggestions. We have addressed each comment separately. The following document has been structured as 1) the blue font indicates the reviewer's comment and 2) the black font shows the authors' reply.

**General Comments:** The content of the article is relevant to the hydrological community and meets the focus of the selected journal. It investigates functional relationships between antecedent wetness and rainfall characteristics and the streamflow response of a small agricultural catchment in Australia. In doing so the authors aim at identifying multiple co-existing runoff processes and potential threshold behavior between catchment-states and streamflow response. The dataset, comprising of hydrometric and hydrochemical parameters has potential but needs more quantitative analysis in order to address the outlined themes.

We thank the reviewer for outlining our research and highlighting its importance for the hydrological community.

My main suggestions to improve the manuscript are the following:

43. I acknowledge the idea to use a decision scheme based on properties and mechanisms to structure runoff processes such as in Figure1. However, such a scheme is designed to result in one dominant runoff process and not in multiple ones such as outlined in the title. (We all know, that processes co-exist in different degrees of intensity). Original versions of such decision schemes are designed for the point or plot scale. While I acknowledge the authors idea to extend it with the concept of connectivity and time scales, I think that this causes a mismatch of scales. At least the authors need to define very clearly what spatial and temporal scale they are considering (and stick to their definition) and what the landscape units are, between which they consider connectivity. I suggest to come up with a separate Figure for connectivity

This figure is intended to address hillslope scale processes and phenomena (i.e. connection to the stream) at the event timescale. We have edited the discussion of the figure to reflect this. We have changed figure 1 to indicate when surface and subsurface connectivity exists (indicated by red arrows).

44. The method section needs to provide more quantitative information. (e.g. soil profile, total number of Q, GW, NS, monitoring sites, procedure of manual sampling, delineation of the saturated area, lab-analysis devices used. (Using two different devices for analyzing isotopes can cause considerable difficulties in comparing or pooling data).

We have add these details to the methods section. As discussed in our response to reviewer 1, the two isotope analyzers were compared. Differences were controlled for and have limited impact here as the

final analysis only used samples analysed by a single machine, with the exception of the uncertainty analysis.

45. **The result section is descriptive and lacks statistical/data analysis to quantify the authors' statements and derive generally applicable results. Some results are based on one or a few selected events only, which is not representative to draw conclusions. I suggest to exploit the entire dataset the authors have at hand and calculate statistics over all events.**

We have exploited the full data set already in terms of the hydrometric analysis. The ion and isotopic analysis addresses issues for specific types of events and we use all available data that can be analysed with reasonable uncertainty, as discussed in our various responses to reviewer 1. We have clarified this in the revised paper. We will provide some further statistical summaries of the various categories of events including statistical summaries of the rainfall characteristics and runoff responses.

46. **Some parts of the result section are the authors' interpretation and better fit in to the discussion section. Terms are either not defined in the text (e.g., in the method section) or not used consistently and the term "threshold" is used in circumstances where "exponential relation" is more appropriate.**

We have defined and clarified terms in response to specific comments from this reviewer. We disagree with the suggestion that "exponential relation" is more appropriate. Runoff responses below the identified thresholds are zero, not just small as would be the case with an exponential relationship. Our use of the term threshold is also consistent with the literature.

47. **The conclusions are drawn from one or two individual rainfall events and not logically derived from or supported by the results of this study. I would encourage the authors to refine and strengthen their analysis based on their dataset. I think it is good to discuss the findings in the light of Fig1. but as it is originally developed for point- or plot scale assessments it misses out the spatial (and temporal) heterogeneity across a catchment. – a fundamental aspect when analyzing thresholds and connectivity – and something that I think the authors try to address. For my detailed comments please see the provided pdf documents and summary of comments.**

We disagree with the reviewers contention that we base our conclusions on single events. The hydrometric results consider 60 rainfall events and 38 runoff events. We acknowledge that some of our later exploration of specific phenomena is based on single events but we have analysed all the suitable data available to us and those results are providing additional lines of evidence in support of the hydrometric analysis. Figure 1 is derived with the hillslope scale in mind. We have edited our presentation to make this clearer and to capture the issues of heterogeneity mentioned.

48. **In general, I think this manuscript has potential to be an interesting contribution to the hydrological society why I encourage the authors to work on a revised version.**

**Page:1**

**Number: 1 Strikeout "increased"**

Done

**Number: 2 Insert text increased**

Done

**Number: 3 Maybe this become clear in the text but how did you distinguish between these runoff source areas/mechanisms?**

We clarify this as follows:

Based on a combination of various hydrometric analyses and some isotope and major ion data, we conclude that event runoff at this site is typically a combination of subsurface event flow and saturation excess overland flow.

**Page:2**

**Number: 1**

**General Comment on Introduction:**

**The introduction is a lot about Fig., and less describes what others have found about thresholds and runoff mechanisms (only mentioned in a few sentences and (good list of references).**

**I also think, that not all studies only concluded that one runoff mechanism is dominating in their catchment. So some pervious work has also concluded, that there are multiple processes happening at the same time or vary seasonally**

We agree with the reviewer comment. We have revised this sentence to (P3, L1):

"While the concept of connectivity has been useful in many of these studies, some studies have concentrated on individual mechanisms. It is less clear how catchments behave when …."

**I also think, that it needs to be clearly stated that Fig. 1 (at least I think) is describing dominant processes at the point or plot-scale.**

We have clarified our intention as follows (P3 L14).

To aid systematic consideration of the variation in runoff processes between catchments and over time, we develop a summary of the status quo in terms of the combined effects of thresholds and connectivity on runoff processes at the hillslope to zero-order catchment scale (Figure 1). Often we think of dominant runoff processes and as Figure 1 is easiest to interpret in that context the following discussion takes that view initially.  However, there is often a mix of runoff processes either spatially due to heterogeneity or for different events and Figure 1 can also be used to interpret such a mixture, which we will return to later.  This paper aims to tease the influence of different processes apart by considering 60 rainfall events (resulting in 38 runoff events) in a small agricultural catchment.  We then

investigate runoff processes in an agricultural catchment in south-eastern Australia in detail and show the shifting importance of different processes over time associated with changes in catchment wetness and rainfall intensity and apply the runoff process framework. We consider the role of thresholds in different catchment states and fluxes as well as the role of thresholds in certain timescales in controlling different modes of hydrologic connectivity and associated rainfall-runoff response. The variety of potential thresholds leads to a variety of runoff processes, as illustrated in the following discussion.

**I like the idea of the authors to go beyond that and think about aspects that need to be considered in terms of connectivity between this points but they are not in the diagram (Fig.1). (see my comments for more detail). If talking about connectivity, this term needs to be diefined and also between what (e.g. hillslope stream).**

We have defined connectivity on p2, l10 as follows:

In this paper we are interested in connectivity in terms of the movement of water from hillslopes to streams and we say there is connectivity along a flow pathway when water is moving along that pathway and contributing to stream flow from the catchment.

**Time scale: I also like the idea to pay more attention to time-scales but it needs to be defined what the authors think of. I guess, in some circumstances they are not talking about time-scale but rather duration.**

Yes at times we mean duration. We have edited the discussion of Figure 1 to clarify this.

**The authors could be more clear what this work contributed to and say more about how their work is new!**

We have stated the contribution as we view it more explicitly at the end of the introduction as follows.

Thus the contributions of this paper revolve around the introduction of the framework in Figure 1 and demonstration of its application using a catchment where multiple runoff processes are important. The paper also contributes to further understanding of the role of thresholds and connectivity in determining flow pathways and runoff processes in agricultural catchments. Most connectivity studies in the past have focused on forested catchments.

**Number: 2**
**There also exists the concept of a continuous nature of connectivity but I agree, that most studies chose the threshold-type of concept. You could say, that you chose the latter concept which implies ...**
In this study the latter concept was used to examine runoff mechanisms in the study area. This concept enabled us to discuss how various runoff mechanisms produced and progressed during the runoff season.

**Number: 3**
**maybe add Penna et al., 2015**

Added.

**not 100% clear. You mean multiple threshold for different runoff generation mechanisms?**

We have clarified this as follows.

Prior to this, we consider the role of thresholds in different catchment states and fluxes as well as the role of thresholds in certain timescales in controlling different modes of hydrologic connectivity and associated rainfall-runoff response.  The variety of potential thresholds leads to a variety of runoff processes, as illustrated in the following discussion

**Number: 5**
**state here that Fig1 is about dominant runoff mechanisms or that you consider them to co-exist in parallel.**

We have added the following to p3 L16

Often we think of dominant runoff processes and Figure 1 is easiest to interpret in that context; however, there is often a mix of runoff processes and this paper aims to tease those apart in a small agricultural catchment and Figure 1 can also be used to interpret such a mixture.

**Number: 6 Insert Text**
**on dominant runoff processes.**
**(see my comment at Fig1)**

We have inserted "on runoff processes" and added some further discussion of both using Figure 1 to interpret dominant runoff processes and also mixtures of runoff processes.

**Page:3**
**Number: 1 Insert Text**
**with different physiographic characteristics**

Added

**Number: 2**
**2006?**

2005 is  relevant but we have also added 2006b

**Number: 3**
**Do you mean the difference between the dominant runoff process concept and a concept of co-existing of these processes with different degree of importance. Can you rewrite?**

We have reworded the text here as follow, which hopefully make it clearer.

It would be attractive to think of the problem of runoff response purely in terms of timescales of competing processes following Oldham et al. (2013), who used a generalise Damköhler to represent the competing effects of transport and reaction timescales on the loss of material along a flow path.  When the reaction timescale is small compared with the transport timescale, the reactant is consumed before reaching the exit of the flow path it is moving along. A complication here is that, both flux and time thresholds are important. This arises because there is finite capacity for flow in various parts of the catchment system.

**Number: 4 Strikeout**

Done

**Number: 5**
**parts?**

Added

**Number: 6**
**I am not sure if they tell directly about connectivity? Connectivity between what? Hillslope stream, neighbouring sites? maybe better "lateral flwo"?**

The sentence was amended to:

Figure 1 is divided into three parts, the lefthand area provides a series of catchment thresholds that depend on hydroclimatic and landscape characteristics and influence the type of runoff process and the connectivity between the hillslope and stream, depending on whether they are exceeded or not.

**Number: 7**
**that depend on climate and landscape characteristics and ...**

Numbers 4-7.  Amended to "Figure 1 is divided into three parts, the lefthand area provides a series of catchment thresholds that depend on hydroclimatic and landscape characteristics and influence the type of runoff process and the connectivity between the hillslope and stream, depending on whether they are exceeded or not"

**Number: 8 Insert text**
**Dominant**

Done

**Number: 9**
**I like this idea but I think you need to introduce the reader more to these time scales.**

Specific examples have been added to the following paragraph including:

P4 L26: Thus the duration of high intensity rainfall compared with the overland flow timescale (flow distance / wave celerity) is important.

P4 L33: Thus the ratio of the lateral flow timescale for the hillslope and the evaporative drying (or other loss) timescale is important here.

**I think these are not time scales but time durations or simple ".. over an event"**

We agree some are durations but the lateral flow process is characterized by a timescale determined by hillslope flow distance and wave celerity, which is clarified in the above response.

**I understand what you want to say but I think now you need to decide if you consider the diagram in Fig.1 to be on the point- scale (which I think the diagram is). If you want to consider processes such as run-on, than you need to redraw your diagram. Suggestion: try to come up with a similar diagram that considers connectivity between the hillslope and the stream and what processes can occur. For connectivity a few other criteria need to be fulfilled. Things you describe here fit well in this other context.**

We see figure 1 applying at the hillslope scale and we have explicitly included (surface and subsurface) lateral flow processes in the diagram.  We are not sure what the reviewer thinks should be added.

**please define**

Reworded: "allowing water to flow along lateral subsurface flow paths"

**On an event-time-scale evapotranspiration might be a minor component. It more effects e.g. antecedent conditions.**

Agreed and we have expanded the following discussion to make that clearer.

**use different word**

Changed to "cease"

**it is not only about persistence (which implies time), it is about a continuous connection (hydraulically or hydrologically from point A to point B.**

**Please define at the beginning A and B.**

Amended to: If the saturation persists for long enough lateral subsurface flow to move down the hillslope and into the stream connectivity between the hillslope and the stream will develop.

**On a singe hillslope there can be patches of different processes. The question is: Do they for a continous path between A and B?**

We have added: Provided continuous saturation exists to the stream, this will lead to a surface flow path connecting the hillslope and stream.

**Number: 17 Insert Text**
**in our study catchment ...**

Added

**Number: 18 Insert Text**
**and space**

Added

**Number: 19**
**patterns of what (soil moisture?)**

Yes, clarified in the text

**Number: 20 Insert Text:**
**in Australia**

Added

**Page:4**

**Number: 1**
**do you mean runoff generation response or dominant runoff mechanism?**
We have clarified this as follows: "These results are used to examine various runoff generation mechanisms and flow pathways that are important in the study catchment and to determine how they contribute to produce the catchment runoff as the catchment wetness and rainfall intensity vary.

**Number: 2**
**In general: the individual sections could improve from the order in which facts are stated. Please provide more quantitative information (e.g. soil profile), total number of Q, GW, NS, monitoring sites be more specific about which instruments have been used to analyze water samples. Where did you manually take samples, how often nad when> How did you determine the saturated area?**
We have heavily revised the methods section to address all these comments.  The more substantive ones are addressed in detail below.

**Number: 3**
**Is this the catchment name? If so please write out the full name once and put the abbr. behind.**
Yes, the name of the catchment was rainbow flume (RBF). Added to the text.
**Number: 4**

**groundwater?**

Added.

**Number: 5**
**groundwater, streamflow?**

Amended to : "1) the riparian area located on the relatively flat convergent lower part of the catchment (outlined in red on Figure 2) included shallow groundwater sites 1, 2, 32"

**Number: 6**
**please describe the criteria you used to define these zones (range of slope values, TWI, etc.)**

The text is revised as follows (p6, L2):
Based on field observations, the topography, range of slopes and the groundwater behaviour, the catchment was divided into four different zones

**Number: 7**
**(1961 - 1990 ?)**
Yes that is right.

**Number: 8 Strikeout**
Done.

**Number: 9**
**... at all groundwater wells ?**
The sentence amended to: "Hand augering revealed a soil depth of between 1 and 1.6 m, and the lower parts of the profile included mottled clay and weathered bedrock particles."

**Number: 10**
**please be clear if this is the average soil depth, the range ...**

**Number: 11**
**What about the other horizons. Is ther an impeding soil layer at what depth?**

**Number: 12**
**section needs better ordering. E,g, if you start with rainfall, than continue with all other meteo parameters. Consider to start with the parameters that are most relevant.**

Section 2.2 reordered.

**Number: 13 Strikeout**

Done.

**Number: 14**
**per event, day, season?**
Per event.  This is clarified in the text.

**Number: 15**

**please provide full name!**
Full name was provided and the sentence was transferred to p7, L16.

**Number: 16**
**give site number**
The hillslope flume is now shown more clearly on see Figure 2.

**Page:5**

**Number: 1**
**with what equation (give a citation)**
The following equation was used for calculation of $Q$ from RBF site. A reference to Clemens et al 1984 and rating equation are provided.

**Number: 2**
**sampling what (I guess streamflow)**
**So you had one site with water samples?**
Yes one site

**Number: 3**
**how did you know the hydrograph before the event? Unclear please rewrite.**
We plotted recorded water levels from the RBF flume in the field when the samples were removed from the auto sampler.

**Number: 4**
**interval for the sampling?**
Hourly for auto sampler and weekly for grab samples

**Number: 5**
**how did you measure the lower range when the probe is 30 cm? Pits? If so, describe their installation procedure**

A soil pit was dug in order to install the lower CS615 probe. Details are given in the paper.

**Number: 6**
**soil water content?**
Volumetric soil moisture which was converted to soil water. The method of calculation of soil water storage/content was completely described in the comment 6, 10-12 of reviewer#1.

**Number: 7**
**please state clearly how many Q, GW, SM sites you had.**
Done

**Number: 8**
**... until 20xx**
Done

**Number: 9**

**I think Odyssey are not pressure transducers!**

It is a NZ brand of pressure transducer. A www address was provided in the original paper.

**Number: 10**

**until 20xx (state the duration you were analysing!**
Done

**Number: 11**

**You need t say more about the manual sampling. Where, how often, at what times in streamflow, in gw?**
Done. The following sentence was added to p7, L28:

"Routine grab sampling was also undertaken at weekly intervals during the main runoff season when water was flowing through the RBF flume."

**Number: 12**
**Please state exactly which instruments you used (not the Professor).**
**If you used different machines but pooled the data for the analysis you have to somehow say something about potential differences (best you state which samples were analyezed where or at least sampels from 2010 ... and 2011 ...**
This point has been answered in our response to comment 2 from Reviewer #1.

**Number: 13**
**How is this determined?**
Sample isotopic analysis is now described in detail.

**Number: 14**
**which ones and which machines**
Sample chemical analysis is now described in detail.

**Page:6**
**Number: 1**
**state the duration of the study period**
This was mentioned in the section 2.1 study location. The study period was between September 2009 and December 2011.

**Number: 2**
**I think stating ASI needs also to state a duration over which ASI is calculated. If it is the VWC at the start of an event, than it is not typically called ASI. Please define!**
In this study ASI was the amount of soil water storage/content at the start of a rainfall event. The method of calculation of the soil water storage/content was described in response to comments from reviewer #1. We refer to it as an index because we are using a point to represent catchment conditions.

**Number: 3**

**not clear from Fig. 2. Also not clear, if you did this during each events. Was it based on mapping?**
**Was it satturated area or the extent of the channel?**

This is described in detail in response to reviewer 1 comment4.

**Number: 4**

**by multiplying the area with ...**

We mean we conceptually separate, as described in response to Reviewer 1 comment 21.

**Number: 5**

**How can you do that unless you are mapping the catchment. Please describe very clearly.**

See reviewer 1 comment 4.

**Number: 6**

**I like the idear of giving definition. Please can you do that in the introduction and for all processes you are referring to!**

We have done this in our description of Figure 1 as follows (p4).

Figure 1 we define the specific runoff processes as follows.

- Infiltration excess runoff: runoff that occurs due to the rainfall (or throughfall) rate exceeding the infiltration capacity of the surface and that results in flows of water to the catchment outlet by surface flow pathways.
- Subsurface stormflow: runoff due to infiltration that generates rapid lateral subsurface flow (i.e. a quickflow response) that flows to the catchment outlet through subsurface flow paths for at least part the distance to the outlet. This water may exfiltrate in low convergent parts of the catchment or directly into the stream before reaching the catchment outlet. Often impeading layers and preferential flow paths are implicated.
- Saturation excess runoff: runoff due to rainfall on saturated areas that flows to the catchment outlet by surface flow pathways. The saturated area may be generated by either lateral flow in excess of the capacity of the hillslope to transmit the lateral flow, a drainage impediment at depth coupled with a sufficient excess of infiltration over evapotranspiration and drainage, or a combination of these.

**Number: 7**

**is this approach also applicable in your catchment?**

We acknowledge that the method is arbitrary, just like all hydrometrically based baseflow separation techniques. We did consider other methods such as digital filters but in our view that are just as arbitrary and more complex. Our results are insensitive to the exact rate of rise assumed. We have added a sentence acknowledging this at p9 l28 as follows.

While this method, like other hydrograph based baseflow separateion methods, is arbitrary, the results are insensitive to the assumed rate of rise.

**Number: 8**
**Summary of general comments:**

**Result section is very descriptive and lacks statistical/data analysis. Some results are based on one or a few selected events only. I suggest to exploit the nice dataset the authors have at ahdn and calcualted statistics over many events to derive generally applicable results.**
**Results (e.g. thesholds) are "read" form graphs and not calculated from the dataset.**
We feel most of the results are obvious from the figures, particularly figure 5d and that the pattern in these figures is the most important feature by which to compare groups. We feel that fitting the thresholds by eye is sufficient for the purposes of the paper. Nevertheless, we have added a table summarizing the average conditions for each runoff event group and a paragraph discussing that table and also undertaken statistical testing using the bootstrap technique to examine statistical differences. These are discussed at the end of section 3.2.

**Result section contains parts, that are better suited in the discussion section because they are subject to the authors interpretation and not based on data only.**
We acknowledge some small sections with results interpretation in section 3.3 where we are examining further evidence for runoff processes. These relate to individual aspects of the results, not interpretations of the results collectively. We wish to retain these where they are because we think they make the results easier to digest for the reader, rather than the reader having to retain all the detail in their memory for later interpretation.

**Terms are either not defined in the text (e.g., in the method section) or not used consistently.**
**The term "theshold" is used in circumstances where an exponential relation is more appropriate.**
**As the different runoff mechanisms are prominent in the title, they should be clearer addressed in the results**
We have edited the paper to define and clarify terms in response to specific comments from this reviewer. We disagree with the suggestion that "exponential relation" is more appropriate. Runoff responses below the identified thresholds are zero, not just small as would be the case with an exponential relationship. Our use of the term threshold is also consistent with the literature.
**Number: 9 Strikeout**

Done

**Number: 10**
**define ASI the first time you use it!**
We have defined this at p7  l8 as follows:

We use this soil water storage at the start of each rainfall event as an index of antecedent soil water (ASI) and assume it represents the catchment wetness condition.

**Number: 11**
**qunatify?**

Fixed

**Page:7**

**Number: 1**
**the overall seasonal pattern is the same but details not!**

We have clarified this as follows p10 l17:

While a strong link between runoff and soil water storage is evident at the seasonal scale in Figure 3, there are exceptions at the event scale

**Number: 2**
**which ones?**

Fixed (p10, L23).

**Number: 3**
**whcih ones! fable them with a,b,c, so you can refer to them**
We have labelled the subfigures and amended the text as follows (p10 L27):

All events (except events on 26/11/2011 and 10/12/2011(Figure 4c)) presented in Figure 4 had zero or very low initial flow. For the events on the 26/11/2011 and 10/12/2011, the initial discharge was 0.07 and 0.13 mm hr-1, respectively.

**Number: 4**
**give value in ()**

Done

**Number: 5**
**that does what?**
We have clarified what each of these past studies considered as follows:

Figure 5 builds on approaches by Detty and McGuire (2010), who considered thresholds in ASI and ASI+Rain (i.e. Figure 5c and d), and Janzen and McDonnell (2015), who considered the impact of event rainfall and rainfall intensity on event runoff (i.e. Figure 5a).

**Page:8**
**Number: 1**
**its not the RC but the Total Runoff**

We have removed this sentence

**Number: 2**
**Fig 5a could be described more. What about the relation with intensity?**

We have expanded our description of Figure 5a as follows (p11 l24):

Figure 5a shows event runoff as a function of event rainfall, with the highest hourly rainfall intensity ($I_{peak}$) indicated in colour. This illustrates that there is little relationship between event rainfall and runoff or between rainfall intensity and runoff.  For some quite large events (up to ~25mm), zero runoff can occur. There was also a wide variation in runoff coefficients (indicated by the scatter). It is also clear that the events with high peak hourly intensity also had relatively large total rainfall accumulations.  Overall, Figure 5a shows that event rainfall and intensity do not effectively differentiate rainfall-runoff behaviour when considered by themselves.

**Number: 3**
**Fig 5b. needs much better explanation (consider to reduce information to make more clear).**

Done

**Number: 4**
**I like the figure but it has (too) many information in one plot in order to deliver your message. readability:**
**Where can I quantify what the length of the grey line is? Where is zero for each individual line?**

We have added further description of the figure which should help to make its interpretation clearer.

**Number: 5**
**but your legend suggest that the filled marker is RC?**

It is stated in the text that the bubble size is the runoff coefficient and the color is the total runoff

**Number: 6**
**use other expression**

This was amended to:

**"**There are several trends that can be discerned from Figure 5b. First the rainfall events analysed here occurred across the full range of catchment wetness and were relatively evenly spread, showing that the rainfall events occur over a representative range of catchment antecedent conditions. The larger rainfall events generally occurred in summer when *ASI*<250 mm and a mix of low and high intensity events occurred for these conditions, also in summer. All the events on a wet catchment (*ASI*>250 mm) had low $I_{peak}$ (≤6.2 mm hr$^{-1}$). Events where the *ASI*+Rain was less than 250mm usually did not generate any runoff, although there were some high intensity rainfall events that were exceptions and a small number of events with very low runoff coefficients (1-4%) where the *ASI*+Rain was generally between 230 and 250mm.  These low runoff coefficient events were at the end of the runoff season".

**Number: 7**
**How much of Fig. 5b is actually describing the general seasonal pattern in rainfall event types?**

**Number: 8 Insert Text**
**rainfall events that state**

Comments 3-8:  We have edited the paragraph in response to all these comments.

**Number: 9**
**the role on what?**

Amended to: Figures 5c and 5d examine the impact of catchment wetness, quantified as ASI and ASI+Rain respectively, at the start (5c) and end (5d) of the event on event runoff response, combined with the impact of rainfall intensity, Ipeak. Catchment wetness is plotted on the x-axis and rainfall intensity on the y-axis. Values ASI=250mm (Fig 5c), ASI+Rain =250mm (Fig 5d) and Ipeak=15mm/h are shown by grey dashed lines.

**Number: 10**
**(quantified as ASI and ASI+ Rain)**

Amended to: Figures 5c and 5d examine the impact of catchment wetness, quantified as ASI and ASI+Rain respectively, at the start (5c) and end (5d) of the event on event runoff response, combined with the impact of rainfall intensity, Ipeak. Catchment wetness is plotted on the x-axis and rainfall intensity on the y-axis. Values ASI=250mm (Fig 5c), ASI+Rain =250mm (Fig 5d) and Ipeak=15mm/h are shown by grey dashed lines.

**Number: 11**
**(RC)**

Added.

**Number: 12**
**say also that the y-axis is Ipeak (Fig 5 c , d)**

Comments 9-13 have been addressed in specific edits.

**Number: 13**
**are these three groups statistically significant. It "looks" promising, they are!**

Yes they are different - see response to general comment on statistical analysis.

**Number: 14**
**why? Guide the reader to your thoughts.**

We have added the following guidance at p12 l21:

In Figure 5d, three different groups of events can be identified.  Looking along the x-axis, there is a threshold at ASI+Rain=250mm that separates events with a significant runoff response from those without.  Looking along the y-axis direction, it can be seen that the ASI+Rain threshold is not successful at distinguishing runoff when Ipeak is high and there is also a threshold in runoff response at Ipeak =15mm/h.  Thus the three groups are

**Number: 15 Strikeout**

done

**what do you mean by synchronized (peak timing?)**
Yes. Reworded to:

These runoff peaks happened at the same time as the highest recorded rainfall intensities.

**Number: 17**
**steaflow?**

fixed

**Page:9**
**Number: 1**
**the RC calculated for the first 2 hours was 18% (is that what you wanted to say?)**
Yes correct. We have reworded to (p13 l9):

The highest intensity was observed in the first two hours of the event and the runoff coefficient calculated for the first 2 hours of the rainfall event was 18%. . The RC calculated for the duration this event was 68%.

**Number: 2**
**You could summarize all findings in one or two sentences at the end. (also applies to other result sections)**

Done

Overall these results demonstrate a range of different rainfall runoff responses. The responses depended on both the catchment wetness as quantified by ASI+Rain and on the peak hourly rainfall intensity, Ipeak , with thresholds of 250mm and 15mm/h being identified. Runoff was produced whenever thresholds in either of these were exceeded.

**Number: 3 Strikeout**

Done

**Number: 4**
**You need to calculate the actual value. In the discussion it is OK to round it to a meanungfull value.**

The paragraph amended to:

"The above presentation of results from Figure 5d identifies a threshold catchment wetness expressed as antecedent soil water storage plus event rainfall depth of 250 mm above which runoff always occurred and another threshold of hourly rainfall depth exceeding 15 mm which also led to runoff production. It should be noted that there is some uncertainty in the $I_{peak}$ threshold as the most intense

event (for *ASI*+Rain<250mm) that failed to produce runoff was 10mm/h, while the least intense event that did produce runoff was 14.8mm/h. Looking at events in the lower right quarter of Figure 5d also shows that the event runoff coefficient tends to increase as either catchment wetness or peak hourly intensity increases. In fact, runoff and non-runoff producing events are very well separated by the relationship $I_{peak}$ = 3/11*(*ASI*-260). These results suggest that there are both wetness dependent and intensity dependent runoff production mechanisms operating. This section examines the evidence for different runoff mechanisms contributing to event runoff."

**Number: 5**
**This is based on 4 datapoints and you have a gap in data between ca. 10 and 15 mm total rainfall. I think it is valid to state the 15 mm threshold but you need to be critical about it in the text.**

We have added this caveat:

It should be noted that there is some uncertainty in the Ipeak threshold as the most intense event (for *ASI*+Rain<250mm) that failed to produce runoff was 10mm/h, while the least intense event that did produce runoff was 14.8mm/h.

**Number: 6**
**you need to do a statistical test to prove this (e.g., rank correlation).**

We have shown a new threshold dependent on both intensity and wetness that demonstrates this.

**Number: 7**
**what do you mean? Where they aligned along transects?**
We have clarified as follows (p14 l10):

In figure 6, the sites are ordered by elevation from highest to lowest in the catchment.

**Number: 8**
**please help the reader to better understand this. Saturation would mean to me, that the entire soil profile is saturated and the groundwater level at the soil surface. That's, only the case for well 1 and 2 (maybe for 32).**
We have reworded this to (p14 l12):

Figure 6 clearly shows that the water table became shallower and that the occurrence of profile saturation (water table at the surface) was restricted to the riparian zone.  Within the riparian zone, sites 1, 2 were saturated much of the time and site 32 quickly became saturated during events. The water table remained below the surface at sites 3 (at the upstream end of the riparian area), 4 and 5 (in Mid slope positions).

**Number: 9**
**What do you mean. Manually read groundwater levels? If so please state this in the method section. It seemed, you were using Odyssey loggers for all wells.**

**If you used irregular intervals, please use a line type that has a dot at each time you were manually taking a reading.**

Yes manually read sites are already indicated by a dashed line with markers for measurement points. This is explained in the caption of Figure 6.

**Number: 10**
**I think it is hard to say something about recession details then plotting 1.5 years of data (x-axis). If you want to say something about the relationship between certain groundwater levels and streamflow, you should either show selected events or even better use statistical tests that show, that streamflow is statistically higher, when certain gw-levels (e.g., in well 4 and 5) are above a theshold of XY cm. That would nicely fit your topic.**

We have added a second panel to Figure 6 so that this is clearer.

**Number: 11**
**Why do you use soil water storage (or are you actually using soil moisture (VWC)?**
**You compare groundwater on the hillslope with soil moisture in the riparian zone?**
**Why not plotting streamflow as a function of depth to groundwater or streamflow as a function of soil water storage?**
We have provided additional context at p14 l25 to explain this as follows.

So far we have examined thresholds in catchment wetness (ASI+Rain) associated with a change in runoff behaviour and the linkage between shallow water tables and catchment response.  It is probably that the linkage between catchment wetness and runoff is through the intermediary of shallow subsurface flow controlled by the existence of saturated conditions within a permeable part of the soil horizon.

**Number: 12**
**Can you remind the reader that site 4 and 5 are on the hillslope**
Done -> The water table remained below the surface at sites 3 (at the upstream end of the riparian area), 4 and 5 (in Mid slope positions).

**Number: 13**
**It looks lie an exponential relation rather than a threshold.**

We have reworded this as follows (p14 L31):

Site 5 in particular shows a rapid change in behaviour for soil water storage around 250 mm, which corresponds with the *ASI*+Rain threshold identified above. As soil water storage moves above this level, much higher water tables develop and those water tables showed relatively rapid recession when shallower than 120 cm. Similar observations were seen at site 4 but the corresponding depth was 140 cm.

**Number: 14**
**This is descriptive. Please can you quantify this using statistics using all you events.**

Amended to (p14, L20):

Looking at the discharge record in Figure 6a, there were periods where significant baseflow persisted between events. These correspond to periods where the water table at site 5 was above about 120 cm and at site 4 was above about 140 cm deep. Flow became more strongly persistent between rainfall events as the water table at sites 4 and 5 rose further. The water table recessions at sites 4 and 5 correspond with flow recessions when the water table was above 120 cm and 140 cm at sites 4 and 5, respectively (Figure 6b).

So far we have examined thresholds in catchment wetness (*ASI*+Rain) associated with a change in runoff behaviour and the linkage between shallow water tables and catchment response. It is probably that the linkage between catchment wetness and runoff is through the intermediary of shallow subsurface flow controlled by the existence of saturated conditions within a permeable part of the soil horizon. Figure 7 shows the relationship between water table levels in the catchment (sites 4 and 5) and soil water storage at the weather station. These sites represent planar and convergent mid slope positions respectively. These sites were chosen because they show significant dynamics and we had logged records available. Other sites with loggers had limited dynamics or short records. Site 5 in particular shows a rapid change in behaviour for soil water storage around 250 mm, which corresponds with the *ASI*+Rain threshold identified above. As soil water storage moves above this level, much higher water tables develop and those water tables showed relatively rapid recession when shallower than 120 cm. Similar observations were seen at site 4 but the corresponding depth was 140 cm. Figure 7 thus explains the linkage between the 250mm *ASI*+Rain threshold and runoff. When soil water storage exceeded this level, water tables rose and lateral subsurface drainage occurred, as evidenced by the recessions. The recessions in particular suggest that the water table moved into a more permeable zone on these occasions. This is consistent with soil profiles being characterised by mottled clay at depth which is likely to have lower hydraulic conductivity. This behaviour at these two wells in combination with the soil water and flow data, indicate that the hillslopes were becoming connected to the catchment outlet via subsurface flow from the hillslope to the riparian zone. Our analysis of isotope data below will add to the evidence for this. There were a few occasions where the water table responded strongly for soil water storage less than 250 mm (Figure 7). As indicated by the red colour, these corresponded to high intensity ($I_{peak}$>15mm/h) rainfall events.

**Number: 15**
**you only gave one Ksat, So you do you know that the lower soil horizon is less permeable (That is a reasonable assumption you could use in the discussion but not in the results section).**

We have provided qualitative evidence for this as follows (p15 l3):

The recessions in particular suggest that the water table moved into a more permeable zone on these occasions. This is consistent with soil profiles being characterised by mottled clay at depth which is likely to have lower hydraulic conductivity.

**Number: 16**
**subsurface drainage form the hillslopes to the stream?**

**You base your conclusion on data of two wells?**

16 and 17 were clarified as follows (p15, l5):

This behaviour at these two wells in combination with the soil water and flow data, indicate that the hillslopes were becoming connected to the catchment outlet via subsurface flow from the hillslope to the riparian zone

**Page:10**

**Number: 1**
**Can you use your isotope data to prove this?**
We refer the reader forward to our isotope analysis: Our analysis of isotope data below will add to the evidence for this.

**Number: 2**
**... in Fig.xy**

Done

**Number: 3**
**Please move this to the discussion section. There it is possible to speculate about potential causes.**

Done.

**Number: 4**
**you only measure groundwater levels and not flow.**
We measured groundwater levels and also water level at the catchment flume.
**Number: 5**
**unless you have a trench, you cannot say anything about hillslope flow.**
We measured groundwater levels and also water level at the catchment flume.

**Number: 6**
**... of streamflow (or of the groundwater levels?)**

**Q would indicate streamflow so you where plotting k of stremaflow as a function of volumetic water content (which is also measured near the outlet of the catchment. What do we learn form this in terms of connectivity between hillslope and streams (see title)?**
We have changed this from "hillslope flow" to "catchment flow" for clarity.

Yes we plot k for streamflow against soil storage but the reviewer is incorrect is saying that it is near the outlet, the soil water storage is measured at the top of the hillslope as shown in Figure 2.

**Number: 7**
**Please quantify!**

We added the following to p15 l27:

$k$ decreased as soil water storage increased at a rate of about $0.1d^{-1}$ for each 10mm increase in soil water storage.

**Number: 8 replace**
**Suggests**
Done.

**Number: 9**
**It is OK to show only one event in the paper but for deriving conclusions you need to analyse many events and give the statistic. You have a good data set, so please use it!**

We explain the limitations of the data for reviewer 1 comment 7.

**Number: 10**
**in what amounts (please check sentence to be complete)**

Amended to:

These results suggest that precipitation on the saturated area generates direct runoff in amounts that are close to what would be expected (i.e. 100% runoff) given that the saturated area is around 5-6% of the catchment area.

**Number: 11**
**This result is different to what you concluded from the hydrometric data.**

We don't see the result as being different to the hydrometric data results.  We do not understand the detail of what the reviewer means or why s/he sees it as different.

**Page:11**
**Number: 1**
**Also here: I tis OK to show one example but you need to analyse all your events and present statistics of the general behaviour. You have a good dataset!**

There were only four high intensity events and one with major ion data.

**Number: 2**
**What is a typical Chloride concentration (define!)**

Reworded to (p16 L31):

The first and second chloride samples respectively plot above and within the typical scatter of data on Figure 10a, while the remaining samples plot well below the typical variation in chloride concentration with flow (Figure 10a).

**Number: 3**
**Please put your interpretation in the discussion section and present only results.**

done

**Number: 4**
**The conclusions are drawn from one or two individual rainfall events and not logically derived from the results of this study but more based on general hydrological conceptualization.**
**The discussion is short and could better tie in the results of this study and critically discuss them.**
**I think it is good to discuss the findings in the light of Fig1. But Figure 1 is for poin- or plot scale assessments and misses out spatial (and temporal) heterogeneity across a catchment.**
**Fig. 1, in my opinion, also misses out connectivity and interactions between sites (e.g., run-on form uphill sites). Your idea to bring that in is good but I think it needs a separate scheme to do this because of different scales of consideration.**

As noted earlier, we disagree with the reviewers contention that we base our conclusions on single events. The hydrometric results consider 60 rainfall events and 38 runoff events. We acknowledge that some of our later exploration of specific phenomena is based on single events but we have analysed all the suitable data available to us and those results are providing additional lines of evidence in support of the hydrometric analysis. Figure 1 is derived with the hillslope scale in mind. We have edited our presentation to make this clearer and to capture the issues of heterogeneity mentioned.

**Number: 5**
**not so clear form the results**

It is as clear as we can make it with the data available.

**Number: 6**
**I think your conclusions from the isotope data are showing a contrast rather support the hydrometric data (see also your title).**

It is not clear what the reviewer means by this comment

**Number: 7**
**Isn't it varying between events and seasons?**
We have clarified the reduction in saturated area during the dry season as follows (p17 l16):

Shallow groundwater data combined with field mapping of surface saturated areas shows that complete profile saturation is limited to about 5% of the catchment area and this saturation is persistent with only

a small variation (see table 1) over the winter-spring season but reduces over the dry summer-autumn period.

**Number: 8**
**Please give evidence rather than your subjective field impression.**

We have added a reference to Table 1 which has the detailed evidence.

**Number: 9**
**one event is not enough to draw conclusions from it!**

Agreed more events would be better but we only have one and in the original text we only have one and only claim support not confirmation of the hypothesis.

**Number: 10**
**This is rather general, please discuss, what you study contributed on new insights on multiple runoff processes occurring in parallel.**

That is covered in subsequent paragraphs.

**Page:12**
**Number: 1**
**Please discuss in more detail, what your study and other studies found.**

We have added clearer statements of contributions.

**Number: 2**
**unfortunately only 4 events**

We analysed all available data.

**Number: 3**
**You used hourly data but a different temporal resolution would lead you to a different threshold. Please, discuss this issue!**

Good point.  We have added the following paragraph.

It is worth noting that the 15mm/h threshold will be dependent on the timestep used in calculating the intensity. The choice of timestep needs to consider the travel time from hillslope to stream as runon infiltration processes occurring on the catchment surface will impact on the connectivity to the stream. Thus the most appropriate time to use for averaging intensity would be the average travel time to the stream because it is this time period which is available to infiltrate rainfall that has become runoff at the point scale. Here the choice of hourly rainfall was pragmatic that was the recording timestep of the AWS.

**Number: 4**

**better move to results**

Done

**Number: 5**
**Please use evidence form your result section to better corroborate this.**

Done

**Number: 6**
**You cannot conclude this from one event, only.**

The sentence is clearly discussing multiple events.

**Number: 7**
**is it intensity or storage capacity?**

We have expanded this paragraph so it reads as follows (p18)

In summary, the process evidence relating runoff behaviour with catchment wetness thresholds, together with the data from shallow wells suggests a catchment where subsurface flow leads to a seasonally saturated riparian area that produces saturation excess runoff in immediate response to rainfall. This saturation excess is augmented by subsurface stormflow when the catchment wetness (*ASI*+Rain) exceeds a 250 mm threshold. This subsurface flow exfiltrates in the riparian area. Volume considerations provide evidence that, for many of the monitored events, water must be coming from outside area.  This area is only about 5% of the catchment but quickflow runoff coefficients very regularly exceed this, often (about 1/3 of runoff events) exceeded 20%, and on occasion exceeded 50%. The role of both saturation excess runoff and subsurface stormflow is corroborated to some extend by isotopic data from one event where the hydrograph separation shows a volume of event water similar to the volume of rainfall on the saturated area and by the highly damped isotopic signal in the stream in general.  When hourly rainfalls exceed a threshold of 15-30 mm hr$^{-1}$, an intensity-dependent runoff process is activated that also contributes flow from the hillslope area outside the riparian zone. It is not clear whether this is a purely surface runoff process or not.  One of the key contributions of this work is clear field evidence of the interplay of both wetness and intensity dependent runoff processes in the one catchment.

**Number: 8**
**A saturated area would immediately produce saturation excess overland flow if hit by a rainfall.\**

Agreed.  Noted in the above paragraph

**Number: 9**
**You assume, that "old water" is from the hillslope but you need to sample water there in order to prove this. In regards to your chosen title, I think this would be something I had expected.**

Volume considerations really mean it can't come from elsewere, as explained above.

**Number: 10**
**I doubt if one threshold (250 mm) is very informative for two very contrasting landscape units (riparian zone and hillslope). I would assume that they have different threshold or 250 mm are more related to the hillslope.**

The riparian zone saturated area is dependent on lateral flows of water from the hillslope. Given the two areas are linked through this process they share a common threshold.

**Number: 11**
**Please give soil properties for this soil horizon in the method section.**

Done

**Number: 12**
**please use your data to give evidence**

We have added (p19 l20)

Based on the volume of event runoff (Table 1), the lack of surface saturation on the hillslope and the behaviour of shllow wells (Figure 6), we infer that the hillslope becomes connected to the riparian zone under these conditions. The existence of a large volume of pre-event water in the event on 12 August 2010 (Figure 8) adds further corroboration to this.

**Number: 13**
**I am not sure, if you analyzed wetness thresholds of connectivity. Please refer to your data to do so. I think it is rather an assumption, that if we see runoff response (of "old water") in the stream we think it is from the hillslope.**

See above response – we haven't claimed to measure – we claim to "infer"

**Page:13**
**Number: 1 Strikeout**

Done

**Number: 2**
**NO! Figure 3 in Detty & McGuire plots streamflow on the y-axis! So they do not say anything about the hillslope contribution but about the entire watersehd runoff.**

We have edited this to make our comparison clearer.  Like us they measure streamflow but use other evidence to make inferences about hillslope contributions.

**Number: 3**
**Please be more speciffic about what they found for what landscape position**

Clarified as follows.

Detty and McGuire (2010) found a relationship between a threshold of 316 mm for *ASI*+Rain and the start of the event streamflow response.  They showed that the *ASI*+Rain threshold corresponded with a water table height threshold. Based on this, they suggested that subsurface flow, transmissivity feedback and preferential flow from hillslope to stream could be used to explain runoff mechanisms in their catchment. They did not observe either Hortonian overland flow or SOF even during the largest events.

**Number: 4 insert text**
**in their catchment**

Done

**Number: 5**
**threshold behaviour of what (streamflow, groundwater, soil moisture)?**
Event runoff response – we have clarified in the text.

**Number: 6**
**What about other studies?**

It is not clear what the reviewer means by this comment.

**Number: 7**
**threshold to initiate what? Streamflow, groundwater response?**

Event streamflow response – we clarified in the text.

**Number: 8**
**Please be more clear in the section before what your three runoff processes are and why?**

Done at the end of 4.1

**Number: 9**
**I think it is good to discuss your findings in the light of Fig1. But you miss out a spatial difference in runoff generation mechanisms across the catchment and treat all as one! I think this is not insightful enough! The scheme in Figure 1 should be applied to different landscape units in your catchment and could so reveal a mosaic of processes. Next step would be to discuss connectivity between thse mosaic parts. That's something different!**

As discussed above we intent Figure 1 to apply at the hillslope to catchment scale and we have amended our introductory discussion of the figure to try to make that clearer.

**Page:14**

**Number: 1**
**give evidence why you assume this?**

We have reworded this paragraph as follows (p20, l26)

The shallow well data (Figure 6, wells 1 and 2, to some extent 32) shows that the riparian area in the lower part of the catchment drains very slowly. There is a substantial reduction in slopes (and probably lower hydraulic conductivity associated with poorly structured, poorly drained soils) that suggests a substantial reduction in lateral subsurface flow capacity. The surface topography also suggests subsurface flow would converge in this area. From the perspective of Figure 1, this leads to a situation where water is taking longer than the typical time between events in the wet season to drain (box 7, "Yes"), resulting in persistent surface saturation and saturation excess runoff generation from the lower catchment

**Number: 2**
**I am not is sure if it is only one process over the entire event**

We believe the results show it is, especially the well data.

**Number: 3**
**I am not so sure: I think Fig 1 is for hte point-or plot-scale and needs an additional step (assessing connectivity) be be meaningful on the catchment scale.**

As noted earlier we have amended our introduction of Figure 1 to address this.

**Number: 4**
**General Comment:**

**The conclusions are drawn from one or two individual rainfall events and not logically derived or supported from the results of this study. I would encourage the authors to refine and strengthen their analysis based on more of their dataset. I think it is good to discuss the findings in the light of Fig1. but as it is originally developed for point- or plot scale assessments it misses out the spatial (and temporal) heterogeneity across a catchment. – a fundamental aspect when analyzing thresholds and connectivity.atial variability within the catchment.**

Most of our conclusioins are drawn from the hydrometric data from 60 rainfall events and 38 runoff events. This is complemented by other data. We have used all the available pertinent data here.

**Number: 5**
**Soil moisture is measured at one site in the catchment?**
Yes, it was measured at one site.

**Number: 6**
**I think you are limited in how much you can say about connectivity**

**Number: 7**
**This is your assumption not clear from data**

**Number: 8**

**I think flow and connectivity cannot be derived form your data. You assume, that this happens.**

6-9: We stand by our conclusions and they they are logical inferences from the data. A combination of data (field measurements of flow, depth and chemistry) have been combined with observations to test hypotheses of flow and connectivity.

**Number: 9**
**What is the hillslope flume? It appears here for the first time do you men the streamflow flume at the catchmetn outlet?!**
Yes clarified.

**Page:15**
**Number:1**
**I think Fig1 is a point or plot scale**

As noted above we present this at hillslope scale

**No comment**

**No comment**

**No comment**

**Page:19**
**Number: 1**
**Typo**

Fixed.

**Number: 2**
**define in caption!**
Done.

**Number: 3**
**typo!**

Fixed.

**Number: 4**
**Group number ? define?!**
Done.

**Number: 5**
**NA? define**
All the above have been addressed for Table 1 and 2.

**No comment**

**No comment**

**Page:22**
**Number 1:**
We have addressed points 1-3 below in our revised description of Figure 1 in the text.

**1) I agree with your decision tree but in fact it results in one dominant runoff mechanisms and your paper shows that these processes co-exist in parallel in different**

**2) This view neglects the contributions form upslope or neighboring sites.**

**3) it is also not clear if you consider the hillslope-scale or the point-scale. Typically these schemes are used to characterize the point- or plot-scale.**

**4) in box #3: do you mean surface topography? Bedrock topography might be quite important (i.e. large storage to prevent still)**
We mean bedrock topography and in the text we reference Janzen and McDonnell who examined connectivity impacts of bedrock depressions.

**5) box #5 has no "NO" option!**

We have added a no option leading to "No runoff"

**Page:23**
**Number: 1**
**Can you destinguish the groundwater, streamflow, soil moisture, etc. sites with different symbols!**
A revised version of Figure 2 and its caption is provided.

[Figure]

**Figure 2.** The study site location within Australia and a hillshaded DEM, topography and sampling site locations at RBF. In this figure the black circles show shallow groundwater sites, the red circle shows soil moisture site and the blue triangle demonstrates the hillslope flume.

**Page:24**

**Number: 1**
**important: Is this one site, n average over how many sites?**
Clarified that it is one site in the caption.

**Number: 2**
**please give full names in captions so that they can stand alone. This applies to all figures and tabes!**
The caption of Figure 3 is revised as:

Figure 3. (A) Weekly rainfall and APET (areal potential evapotranspiration) time series (data from automatic weather station (AWS), (B) soil water storage in top 60 cm of the soil profile and weekly runoff time series at RBF.  The vertical bars show the timing of events in Figure 4.

**Page:25**
**Number: 1**
**why green?**

Now blue

**Number: 2**
**can you mark these events in Fig.3B!**
**and maybe make the rows of these events bold in table 1 and 2**

Done

?It is graphed

Figure 4 caption is now:

Figure 4. Characteristics of 8 selected rainfall-runoff events sorted by their antecedent soil moisture index (ASI). Rainfall is shown in blue. It should be noted that the axis scales vary between events. All events (except those on 26/11/2011 and 10/12/2011) had zero or very low initial discharge. For the events on 26/11/2011 and 10/12/2011, the initial discharges were 0.07 and 0.13 mm hr$^{-1}$, respectively.  .

**Number: 5**
**make the reader aware, tha the y-axis scale differ between events.**

Done

ASI is now clearly defined in the methods and in the caption

done

Yes, clarified in caption

**Number: 4**
**You are actually not showing thresholds in these figures**

Now shown

**Page:27**

Done

Done

**Page:28**

Done (Ipeak>115mm/h)

Now explained in detail in the methods and noted in the caption.  It is storage in top 60cm at the AWS.

Now explained in methods

**Page:29**

This is now explained in the discussion of this figure.

[revised manuscript text omitted]